



# Improved representation of soil moisture simulations through incorporation of cosmic-ray neutron count measurements in a large-scale hydrologic model

Eshrat Fatima[1,2], Rohini Kumar[1], Sabine Attinger[1,2], Maren Kaluza[1], Oldrich Rakovec[1,3], Corinna Rebmann[1], Rafael Rosolem[4], Sascha Oswald[2], Luis Samaniego[1,2], Steffen Zacharias[5], and Martin Schrön[5]

[1]Dep. Computational Hydrosystems, UFZ - Helmholtz Centre for Environmental Research GmbH, Leipzig, Germany
[2]Institute of Environmental Science and Geography, University of Potsdam, Potsdam, Germany
[3]Faculty of Environmental Sciences, Czech University of Life Sciences Prague, Praha-Suchdol 16500, Czech Republic
[4]Department of Civil Engineering, University of Bristol, Bristol
[5]Dep. Monitoring and Exploration Technologies, UFZ - Helmholtz Centre for Environmental Research GmbH, Leipzig, Germany

**Correspondence:** Eshrat Fatima (eshrat.fatima@uni-potsdam.de), Rohini Kumar (rohini.kumar@ufz.de), Martin Schrön (martin.schroen@ufz.de)

**Abstract.** Profound knowledge of soil moisture and its variability plays a crucial role in hydrological modeling to support agricultural management, flood and drought monitoring and forecasting, and groundwater recharge estimation. Cosmic-ray neutron sensing (CRNS) have been recognized as a promising tool for soil moisture monitoring due to their hectare-scale footprint and decimeter-scale measurement depth. Different approaches exists that could be the basis for incorporating CRNS

data into distributed hydrologic models, but largely still need to be implemented, thoroughly compared, and tested across different soil and vegetation types. This study establishes a framework to accommodate neutron count measurements and assess the accuracy of soil water content simulated by the mesoscale Hydrological Model (mHM) for the first time. It covers CRNS observations across different vegetation types in Germany ranging from agricultural areas to forest. We include two different approaches to estimate CRNS neutron counts in mHM based on the simulated soil moisture: a method based on the

Desilets equation and another one based on the Cosmic-ray Soil Moisture Interaction Code (COSMIC). Within the Desilets approach, we further test two different averaging methods for the vertically layered soil moisture, namely uniform vs. non-uniform weighting scheme depending on the CRNS penetrating depth. A Monte Carlos simulation with Latin hypercube sampling approach (with N = 100 000) is employed to explore and constrain the (behavioral) mHM parameterizations against observed CRNS neutron counts. Overall, the three methods perform well with Kling-Gupta efficiency > 0.8 and percent

bias < 1% across the majority of investigated sites. We find that the non-uniform weighting scheme in the Desilets method provide the most reliable performance, whereas the more commonly used approach with uniformly weighted average soil moisture overestimates the observed CRNS neutron counts. We then also demonstrate the usefulness of incorporating CRNS measurements into mHM for the simulations of both soil moisture and evapotranspiration and add a broader discussion on the potential and guidelines of incorporating CRNS measurements in large-scale hydrological and land surface models.



## 1 Introduction

Soil moisture is a key terrestrial climate variable because it controls the mass and energy exchange between the Earth's surface, the groundwater, the vegetation, and the atmosphere. Understanding soil moisture levels with changes in temperature is crucial for enhancing the predictability of climate patterns on inter-seasonal and annual time scales, as highlighted in previous studies (Santanello Jr et al., 2011; Seneviratne et al., 2006). Moreover, soil moisture variability also plays a significant role in a wide range of applications, including flood forecasting, weather forecasting, climate modeling, agricultural management, and groundwater recharge (Van Steenbergen and Willems, 2013; Albergel et al., 2010; Jablonowski, 2004; Wahbi et al., 2018; Samaniego et al., 2019; Barbosa et al., 2021). In hydrological modeling, soil moisture is a key variable controlling the partitioning of precipitation into evapotranspiration, infiltration, and runoff (Fuamba et al., 2019; Zhuo et al., 2020). Proper initialization and modeling of soil moisture are crucial for predicting other hydrologic processes (e.g., runoff, evapotranspiration, etc). However, uncertainties in input data and model parameters, and limitations in the representation of subsurface processes, can impede the reliability of soil moisture estimation (Chen et al., 2011).

Obtaining accurate soil moisture measurements at a field scale is challenging due to the limitations of current measurement methods and the complexity of subsurface processes (Dong and Ochsner, 2018). Estimating the average soil moisture at a mesoscale ($\approx$ 1–100 km) is particularly difficult because of the limitations of current measurement techniques in terms of their measurement area or "footprint" and measurement methods bridging the scale gap between point-scale measurements and the areal average required for hydrologic modeling are needed (Chan et al., 2018). One of the promising approaches to infer soil moisture at a field scale is based on the recent development of a cosmic-ray neutron sensing (CRNS) technique (Zreda et al., 2008; Desilets et al., 2010; Köhli et al., 2015; Schrön et al., 2017). The CRNS method utilizes naturally occurring neutrons on the Earth's surface produced by cosmic rays to derive near-surface soil moisture. The rate of neutron moderation in soil, and emitted neutron density above ground, is affected by soil moisture levels: dry soils emit more neutrons while wet soils moderate neutrons more strongly and emit less. This change in neutron emission can be detected with a neutron detector (Zreda et al., 2012; Köhli et al., 2021). A cosmic-ray soil moisture probe is usually calibrated locally using soil samples from its support volume (Franz et al., 2012; Schrön et al., 2017). CRNS has demonstrated a high potential for estimating average soil moisture over areas of several hectares in size and tens of decimeters in-depth (Köhli et al., 2015; Schrön et al., 2017). CRNS data are used in various studies like land surface modeling, understanding vegetation dynamics, catchment hydrology, and supporting the agriculture sector with information on soil types and climates (Franz et al., 2020). Soil moisture measured by CRNS has also been used in water balance studies and has been helpful in estimating infiltration and evapotranspiration (Schreiner-McGraw et al., 2015; Foolad et al., 2017; Wang et al., 2018).

Previous studies, such as Barbosa et al. (2021) and Brunetti et al. (2019) have utilized the HYDRUS-1D model to simulate soil moisture at the field scale, incorporating the so-called COSMIC operator to simulate neutron count rates of a CRNS measurement (Shuttleworth et al., 2013). They compared neutron count rates observed with simulated data to calibrate soil hydraulic parameters inversely, whereas beforehand this was limited to be done via comparison of depth-averaged soil moisture values (Rivera Villarreyes et al., 2014). The potential utility of using CRNS data to calculate volumetric soil water content



(SWC) and improve soil hydraulic parameters within land surface models has also been observed earlier, as highlighted by Rosolem et al. (2014).Then Iwema et al. (2017) used a Land Surface Model to investigate the impact of reducing the scale mismatch between surface energy flux and soil moisture observations of CRNS measurement data, using point-scale soil moisture data from soil layers up to 30 cm depth and simulated data from 2012–2015. Recently, Patil et al. (2021) used a distributed Land Surface Model, Data Assimilation Research Testbed (DART) together with a CRNS time series, and Ensemble Adjustment Kalman Filter to simulate the water and energy balance of the land surface. Both of the latter studies analyzed the water and energy balance of the land surface and the effects of various data assimilation and calibration techniques.

Furthermore, the conceptual rainfall-runoff model Hydrologiska Bryans Vattenbalansavdelning (HBV) was used to study various aspects of water balance and model calibration. For instance, Dimitrova-Petrova et al. (2020) employed CRNS data in a mixed-agricultural landscape to investigate the water balance on the land surface, while Beck et al. (2021) evaluated remote sensing products and groundwater level measurements to temporally calibrate a HBV model. Both studies highlighted the challenge of comparing satellite-derived soil moisture with point-scale in-situ measurements. Baatz et al. (2017) was the first study that utilized spatially distributed hydrological modeling to update soil moisture states across a catchment using CRNS information, the FAO soil map, the BK50 soil map, and other soil data as input information in their Community Land Model (CLM). They found that assimilating data of a CRNS network improved the characterization of SWC on the catchment scale by updating spatially distributed soil hydraulic parameters of a land surface model. Furthermore, Zhao et al. (2021) employed CLM version 3.5 to assess the significance of CRNS data in a land surface model. They conducted simulations based on data from 13 CRNS stations over a two-year period (2017-2018) and employed a simplified Richards equation for water movement calculations. However, the model's limitations include the absence of lateral flows and groundwater representation.

Eventually, the mesoscale Hydrological Model (Samaniego et al., 2010b; Kumar et al., 2013b, mHM;) is known for its spatially distributed hydrologic predictions at a large scale incorporating scale-aware regionalized parameterization technique. Therefore, by including a CRNS neutron count framework, the mHM model becomes a useful tool for improving simulated soil water content and furthering our understanding of the water cycle. This is made possible by the availability of observed CRNS data, which opens up new opportunities for research into novel hypotheses, improving model performance, and developing hydrological modeling methods. All the mentioned studies either compared the simulated and observed soil moisture products or incorporated the first COSMIC version to compare neutron counts directly. As argued by Shuttleworth et al. (2013), the usage of neutron counts is the favorable way to compare simulations with data, since the CRNS sensor intrinsically averages over soil moisture layers while the measurement depth varies with soil moisture and consequently over time. Hence, the direct usage of CRNS neutron counts avoids the question of which modeled SWC layer the observations should be compared to and at what time scales. The COSMIC method is complex because its approach enables a more comprehensive representation of the neutron counting process, which is computationally more demanding than using the analytical Desilets equation (Desilets et al., 2010).

In this study, we incorporate the prediction of neutron counts directly within mHM. Here, we test three approaches, (i) the direct calculation of neutrons from the equal-averaged SWC profiles based on Desilets et al. (2010), (ii) the same with a weighted-average profile SWC based on Schrön et al. (2017), and (iii) the physics-based model COSMIC by Shuttleworth





et al. (2013). We evaluate the simulation of neutron counts at scales of $1.2 \times 1.2\,\mathrm{km}^2$, comparing the results to observed

neutron counts from three different sites including agriculture, deciduous forests, and grasslands. The goal of this study is to investigate the potential of using CRNS probes and measured neutron counts to improve soil moisture predictions through simulations in mHM across different land covers and soil properties and to evaluate the feasibility of incorporating neutron count measurements into the modeling scheme. We employ a (calibration) framework by applying a Monte Carlo experiment to account for parameter uncertainties. We further cross-evaluate our simulations and test the reliability of the CRNS incorporated

soil-moisture scheme in mHM for simulating other variables by utilising time-series of observed evapotranspiration from an eddy covariance station available. Finally, we discuss and provide guidelines (challenges and limitations) for incorporating CRNS measurements in a large-scale hydrologic model.

In summary, the present paper aims to answer the following research questions:

- What is the best approach to simulate CRNS neutron counts in a hydrological model considering the heterogeneity of
vertical soil moisture profiles?

- What is the impact of model calibration with CRNS observations on simulated soil moisture and evapotranspiration?

- Is the mHM at approx. 1 km resolution capable of capturing the dynamics of hectare-scale CRNS measurements at different landcover sites?

## 2 Materials and Methods

### 2.1 Experimental Site Description

For this study, we select four sites with CRNS sensors, namely *Grosses Bruch*, *Hohes Holz*, *Hordorf*, and *Cunnersdorf* in Northern Germany, as provided already within COSMOS EU (Bogena et al., 2022) with particularly long time series and homogeneous land cover (see Tab. 1). The first three sites belong to the TERENO observatory "Harz/Central Germany lowland" (Zacharias et al., 2011) while the fourth site is part of an agricultural research farm operated by the German Weather Service

(DWD). The *Grosses Bruch* site is a meadow/grassland that is usually flooded naturally once or twice a year and is thus prone to producing methane fluxes. The meadows have sandy loam fluvisol-gleysol soil, which is 1.5 meters deep and partially covered with a layer of peat (Wollschläger et al., 2017). Meteorological conditions like soil moisture and temperature at various depths are continuously monitored by a wireless soil moisture monitoring network (Schrön, 2017). *Hohes Holz* is a deciduous forest site and the performance of the CRNS sensor there is highly dependent on dynamic effects such as tree canopy

water or seasonal fluctuations in wet biomass. Water trapped in leaves and litter can present a particular challenge for CRNS measurements, especially at forest stations (Bogena et al., 2013). The mean annual air temperature for each sites ranges from 10.0 to 10.9 °C and the average yearly precipitation ranges from 458 to 535 mm.



**Table 1.** Geographical characteristics of study sites.

| Site | Latitude | Longitude | Altitude | Land Cover | Precipitation | Temperature | Period |
|------|----------|-----------|----------|-----------|---------------|-------------|--------|
| | [°N] | [°E] | [m] | | [mm] | [°C] | |
| *Grosses Bruch* | 52.02 | 11.10 | 80 | Pasture, grassland | 458 | 10.1 | 24/06/2014–31/01/2021 |
| *Hohes Holz* | 52.09 | 11.22 | 217 | Forest, hilltop | 469 | 10.3 | 27/08/2014–31/01/2021 |
| *Hordorf* | 51.99 | 11.17 | 82 | Cropland | 463 | 10.3 | 29/09/2016–31/01/2021 |
| *Cunnersdorf* | 51.36 | 12.55 | 140 | Cropland | 535 | 10.9 | 23/06/2016–31/01/2021 |

**Figure 1.** Study area map of Germany, highlighting the four test sites where observed neutron count rates from CRNS are utilized to evaluate the performance of mHM. The figure utilizes OSM basemap layers from (© OpenStreetMap contributors 2021; distributed under the Open Data Commons Open Database License (ODbL) v1.0) OpenStreetMap contributors (2020).



## 2.2 The mesoscale Hydrological Model (mHM)

mHM is a spatially distributed process-based hydrologic model (www.ufz.de/mhm) representing processes such as canopy
interception, snow accumulation and melting, soil moisture dynamics, infiltration and surface runoff, evaporation, underground
storage, and runoff generation, deep infiltration and baseflow, as well as runoff attenuation and flood routing (Samaniego et al.,
2010a; Kumar et al., 2013a). The mHM model is flexible for hydrological simulations at different spatial scales due to its
novel Multi-scale Parameter Regionalization approach (MPR; Samaniego et al., 2010b); and has demonstrated applicability in
diverse settings (Samaniego et al., 2010a; Kumar et al., 2013a; Rakovec et al., 2016; Samaniego et al., 2017). The MPR's basic
concept is to estimate parameters (e.g., porosity) based on soil properties (e.g., sand and clay content) using transfer functions at
a fine spatial resolution (e.g. 100 m) and upscaling them to modelling resolutions (e.g., 1 km). In MPR, transfer functions (e.g.,
pedo-transfer functions to estimate soil parameters) are combined with morphological inputs (e.g., soil texture properties) and
thus lead to model hydrologic parameters (e.g., porosity or hydraulic conductivity of the soil) (Livneh et al., 2015; Zacharias
and Wessolek, 2007). In mHM, the soil moisture horizons/profile can be divided into several horizons, all of which are sensitive
to root water uptake and evapotranspiration processes. mHM simulates the daily dynamics of soil moisture at different depths
considering the incoming water (e.g., rainfall plus snow melt for the topmost layer and infiltration from above layers for
other layers) and outgoing ET and ex-filtration fluxes. Further details on mHM code can be found at https://mhm-ufz.org and
underlying modelling concepts at Samaniego et al. (2010a); Kumar et al. (2013a).

## 2.3 Model Set-up

The latest version 5.12 of mHM is used in this study (see Samaniego et al., 2023, and https://github.com/mhm-ufz). The model
is executed over six years (2014–2020) with a daily time step, and the spatial resolution of the mHM grid cells is fixed at
0.015625° (approx. $1.2\,km^2$) using the WGS84 Coordinate Systems.

CRNS neutron count rates are calculated based on daily soil moisture values simulated with mHM. The model boundary
conditions such as precipitation and temperature for the mHM model are acquired from the German Weather Service (DWD)
station closest to the test site. The potential evapotranspiration required by mHM is estimated using the Hargreaves-Samani
method (Hargreaves and Samani, 1985). The model setup and parameterization for the soil moisture module use the scheme
optimized by Boeing et al. (2022). A raster dataset describing the distribution of the soils in the model area and a corresponding
lookup table with the attributes depth, soil texture (sand and clay fraction), and bulk density are required as soil input data and
are derived from national digital soil maps provided by the Federal Institute for Geosciences and Natural Resources (BGR,
2020). The data set contains physical and chemical properties for soil at different layers and the available at a resolution of
1:250,000 (BUEK 200; BGR, 2020). mHM uses three dominant land cover classes (forest, permeable, and impervious) that
were retrieved by a GLOBCOVER database ESA (2009). Furthermore, vegetation characteristics like Leaf Area Index (LAI)
and fraction of roots for different vegetation types are prescribed in the model. The mHM soil domain is divided into three
horizons with depths of 0–5 cm, 5–25 cm, and 25–60 cm. The upper two model layers are parameterized using the topsoil
layer properties while for the lower model layer, the subsoil properties are used. More details on the underlying input data for




mHM can be obtained from Boeing et al. (2022). Figure 2 shows the flow diagram depicting the basic methodology of the entire study.



**Figure 2.** Flowchart depicting the methodology employed for calculating CRNS neutron counts through the utilization of the LHS technique for parameterization in mHM. The computation of CRNS neutron count is carried out through three distinct approaches: $N_{(Des,Uni)}, N_{(Des,W)},$ and $N_{(COSMIC)}$.

## 2.4 Conversion of soil moisture to neutron count rate

In this study, we compare observed neutron counts from CRNS data with simulated neutron counts estimated from modeled soil moisture with the goal of optimizing the parameterization of soil water content from mHM shown in Fig. 3. By coupling the approaches from Desilets et al. (2010) and Shuttleworth et al. (2013) each directly with the mHM model, we are able to



account also for the uncertainty in the model predictions and test their feasibility across four distinct sites. We analyzed the soil water content data at different soil layers (0–5 cm, 5-25 cm, and 25-60 cm) simulated by mHM. We compared these simulated values with the measured soil water content obtained through CRNS. Simulations from mHM revealed that the sensitivity to the highest soil water content was observed at 5 cm depth and decreased with increased depth, also indicating that SWC is high responsiveness to precipitation.

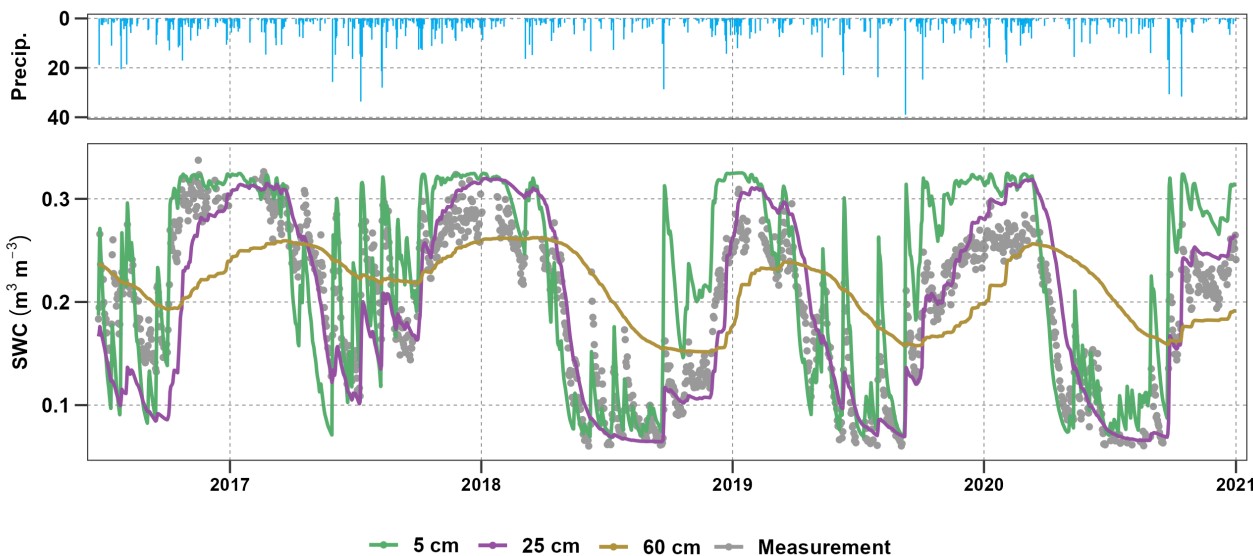

**Figure 3.** Daily time series of soil water content ($cm^3 cm^{-3}$) at the *Cunnersdorf* site. The graph shows a comparison between the measured SWC from CRNS data representing an integral over the first decimeters and the simulated data derived from the mHM for three distinct soil depths, at 0–5 cm (green), 5–25 cm (purple), and 25–60 cm (brown).

### 2.4.1 Desilets based method

In the present study, we utilize the soil moisture information from the mHM model to convert it into neutron counts using the Desilets et al. (2010) empirical-based approach by calculating neutron counts from soil moisture, three constant parameters i.e., $a_0$, $a_1$, $a_2$, and $N_0$, which is further improved by adding lattice water and bulk density following the approaches by Dong et al. (2014) and Hawdon et al. (2014), respectively. Theoretically, the $N_0$ parameter, which represents the neutron count rate level of the particular CRNS probe used for rather dry soil at the local conditions, should be site-specific but does not change over time, as noted by Franz et al. (2013) and Hawdon et al. (2014). In order to obtain accurate measurements of soil moisture using CRNS data in the mHM model, $N_0$ has to be estimated through calibration and is crucial as it directly affects the accuracy of the mHM neutron counts results. This time-constant calibration parameter is specific to each site environment and reference condition. This parameter primarily depends on site-specific environmental factors and reference conditions. This coefficient is specific to the particular CRNS detector and may be impacted by factors such as soil chemistry, vegetation cover, heterogeneity,



and altitude at each observation site Schrön et al. (2021). Therefore, calibration of $N_0$ is necessary for each CRNS data set at a site to ensure realistic model output.

Soil moisture for three vertical mHM soil layers is used to drive both the Desilets method and the COSMIC operator. To improve comparability between measurements and modeling techniques, Schrön et al. (2017) propose the depth-weighted approach. This approach incorporates the contributions of different soil layers by calculating depth-weighted average SWC, $\theta_{avg}$, resulting in a more comprehensive representation of soil moisture dynamics. This gives CRNS neutron count rates, after being corrected for incoming neutron flux, pressure, and air humidity variations, to be

$$N_{\text{Des}} = N_{0,\text{Des}} \left( \frac{a_0}{(\theta_{\text{avg}} + \theta_{\text{lw}})/(\varrho_b/\varrho_w) + a_2} + a_1 \right) \tag{1}$$

Among the four parameters, three of which are coefficients ($a_0 = 0.0808$, $a_1 = 0.372$, and $a_2 = 0.115$) derived from neutron particle physics modeling (Zreda et al., 2008; Desilets et al., 2010), and are considered as constants. The fourth parameter, $N_{0,Des}$ is $N_0$ when using the Desilet's equation, and here is a free calibration parameter. Whereas the parameter $\theta_{\text{lw}}$ is the grid average volumetric water content of the equivalent lattice water content of the CRNS area (cm³ cm⁻³), $\varrho_b$ (g cm⁻³) is the

bulk density of the dry soil, usually determined from soil samples, and $\varrho_w = 1$ g cm⁻³ is the density of water. Organic water equivalent (from SOC) and biomass water equivalent, these variables are frequently unknown, especially biomass. To address this, the free parameter $N_0$ is utilized to account for these unknowns. For lattice water, we assume a linear relationship to clay content (Avery et al., 2016):

$$\theta_{\text{lw}} = \theta_{\text{lw0}} \cdot CC + \theta_{\text{lw1}}, \tag{2}$$

$CC$ denotes the clay fraction in %. The derived quantity lattice water, $\theta_{\text{lw}}$, is regionalized based on $CC$ and varies between 0.0 and 0.1 m³/m³. In order to obtain the average soil moisture for a layered soil moisture profile within mHM, the following averaging equation is employed:

$$\theta_{\text{avg}}(w, \theta) = \frac{\sum_{i=1}^{n} w_i \theta_i}{\sum_{i=1}^{n} w_i} \tag{3}$$

where the volumetric soil water content at a specific layer of mHM in a given profile is denoted by $\theta_i$ (m³ m⁻³). The total

number of layers in all soil sampling profiles is represented by the variable $n$, and the weight assigned to layer $i$ is denoted by $w_i$. In the uniformly weighted approach, all weights equal one:

$$N_{\text{Des,U}} = N_{\text{Des}}(w_i = 1) \quad \forall i. \tag{4}$$

In the weighted-averaging approach, the weights are determined based on Schrön et al. (2017):

$$N_{\text{Des,W}} = N_{\text{Des}}(w_i = w(z_i)), \tag{5}$$

where $w(z) = e^{-2z/D}, \tag{6}$

$$D = \varrho_b^{-1} \left( p_0 + p_1 \left( p_2 + e^{-p_3 r} \right) \frac{p_4 + \theta}{p_5 + \theta} \right). \tag{7}$$





Here, $z_i$ is the depth of the given soil moisture layer $i$, $D$ is the average vertical footprint depth of the neutrons, $p_i$ are numerical parameters presented in Schrön et al. (2017), and $r$ (m) represents the distance from the sensor. It should be noted that the equation for D seems to be valid only if bulk density does not get too small and SWC is not too high Kasner et al. (2022). In our model, we set $r = 1$ m which is sufficient to represent the average depth across the footprint radius within the model grid. The soil moisture profile is converted to a single average neutron count per grid cell using Eqs. 1–5.

### 2.4.2 Cosmic Ray Soil Moisture Interaction Code (COSMIC)

The Cosmic Ray Soil Moisture Interaction Code (COSMIC) is an analytical, physics-based model that is well-suited for data assimilation applications. It includes descriptions of the degradation of incoming high-energy neutron flux with soil depth, the production of fast neutrons at each soil depth, and the scattering of resulting fast flux neutrons before reaching the soil surface. These processes have a parametric dependence on soil chemistry and moisture content. The COSMIC method solves this inverse problem by calculating neutrons based on soil water profiles, which could then be compared with observed neutrons without the need to deal with dynamic sensing depths or weightings.

$$N_{\text{COSMIC}} = N_{0,\text{COSMIC}} \sum A_{\text{high}}(z) X_{\text{eff}}(z) A_{\text{fast}}(z) \,, \tag{8}$$

where $A_{\text{high}}(z) = e^{-\Lambda_{\text{high}}(z)} \,,$

$$A_{\text{fast}}(z) = \frac{2}{\pi} \int\limits_0^{\pi/2} e^{-\Lambda_{\text{fast}}(z)} (\cos\varphi)^{-1} \, \mathrm{d}\varphi \,,$$

$$X_{\text{eff}}(z) = \alpha_{\text{COSMIC}} X_{\text{soil}} + X_{\text{water}} \,.$$

In this model, $A_{\text{high}}$ represents the high-energy neutron attenuation, $A_{\text{fast}}$ represents the fast neutron attenuation, and $X_{\text{eff}}$ represents the production of fast neutrons from high-energy neutrons in the soil-water composite. It takes into account the different mechanisms in both, water and soil, where soil is typically less effective in producing fast neutrons by a factor of $\alpha_{\text{COSMIC}} \approx 0.24$ (g cm$^3$g$^{-1}$), depending on bulk density.

$$X_{\text{soil}}(z) = \Delta z \varrho_{\text{bulk}} \,, \tag{9}$$

$$X_{\text{water}}(z) = \Delta z \varrho_{\text{water}}(\theta_z + \theta_{\text{lw}}) \,, \tag{10}$$

$$\tag{11}$$

The total attenuation lengths of high and fast neutrons in the soil-water composite are described using physically motivated length scales $L_i$.

$$\Lambda_{\text{high}}(z) = \frac{X_{\text{soil}}(z)}{L_1} + \frac{X_{\text{water}}(z)}{L_2} \,, \tag{12}$$

$$\Lambda_{\text{fast}}(z) = \frac{X_{\text{soil}}(z)}{L_3} + \frac{X_{\text{water}}(z)}{L_4} \,. \tag{13}$$



The COSMIC function considers the attenuation of incoming high-energy neutrons ($A_{\text{high}}$) and their interaction with soil to produce effective neutrons ($X_{\text{eff}}$) in addition to the attenuation of isotropically propagating fast neutrons. The parameter $\alpha_{\text{COSMIC}}$ represents the soil's relative efficiency of forming fast neutrons, and length constants $L_1$, $L_2$, $L_3$, and $L_4$ (in gcm$^{-2}$) are related to local soil properties. COSMIC uses several time-invariant, site-independent, and site-specific parameters, including $L_1$ = 162.0 (g cm$^{-2}$), $L_2$ = 129.1 (g cm$^{-2}$), and $L_4$ = 3.16 (g cm$^{-2}$), as reported by Shuttleworth et al. (2013), regardless of location. However, the $L_3$ parameters (g cm$^{-2}$) and $\alpha_{\text{COSMIC}}$ (g cm$^3$g$^{-1}$) vary with soil bulk density $\varrho_{\text{b}}$ which change with depth. According to the model code $L_{30}$ and $L_{31}$ nomenclature are given as per the model code in mHM (https://github.com/mhm-ufz).

$$L_3 = L_{30}\varrho_{\text{b}} - L_{31}. \tag{14}$$

where $\theta_{\text{lw0}}$, $\theta_{\text{lw1}}$ are the parameters of the lattice water and the equation of $L_3$ was taken from the dissertation of Schrön (2017). The regional formulation of the COSMIC method has been revised to include the $\theta_{\text{lw}}$ lattice water content as well.

Besides the addition of lattice water to the code, the original version of COSMIC has also been numerically optimized to substantially increase the computational performance. This includes the calculation of the geometric integral (Eq. 8) based on lookup tables.

## 2.5 Constraining of model parameterization

In this study, we utilize a model calibration technique to identify the most suitable parameter values for the mHM model. The process of model calibration involves modifying the parameter values of the model to achieve a satisfactory standard for an objective function by comparing the predicted output with the observed data (James, 1982). In this study, we use the general concept of the Kling-Gupta efficiency KGE for designing the objective function, which is widely employed in hydrological modeling to assess the efficiency of a model (Gupta et al., 2009). However, we modify the KGE (Eq. 15) by removing the correlation coefficient $\rho$, as it is just a measure of the temporal signature and is largely dominated by seasonality alone. This modified KGE$_{\alpha\beta}$ (Eq. 16) only depends on variability ($\alpha$) and bias ($\beta$) and variants of it have been used also in other studies (see, e.g., Martinez and Gupta, 2010; Mai, 2023). We utilize observed neutron count data from CRNS and estimated neutron count data from the mHM model to calculate various metrics such as the Kling-Gupta efficiency coefficient (KGE$_{\alpha\beta}$), Nash-Sutcliffe efficiency (NSE) by Nash and Sutcliffe (1970), coefficient of determination ($R^2$) by Kvålseth (1985), and percentage bias (PBIAS) by Gupta et al. (1999). The optimal PBIAS value is 0, with lower values indicating more accurate model simulations. Positive values indicate underestimation by the model, while negative values indicate overestimation. This approach allows us to minimize uncertainty in the simulated neutron count data by comparing it to observed data and determining the optimal parameter values for the mHM model. A summary of the individual parameters and their ranges can be found in



Table 2.

$$\text{KGE} = 1 - \sqrt{(\alpha - 1)^2 + (\beta - 1)^2 + (\rho - 1)^2}, \tag{15}$$

$$\text{KGE}_{\alpha\beta} = 1 - \sqrt{(\alpha - 1)^2 + (\beta - 1)^2}, \tag{16}$$

with Variability $\alpha = \sigma_{\text{sim}}/\sigma_{\text{obs}}$, (17)

Bias $\beta = \mu_{\text{sim}}/\mu_{\text{obs}}$, (18)

Correlation $\rho = \rho(\text{sim}, \text{obs})$,

$$\text{NSE} = 1 - \frac{\sum_{i=1}^{n}(y_{\text{sim,i}} - y_{\text{obs,i}})^2}{\sum_{i=1}^{n}((y_{\text{obs,i}} - \overline{y}_{\text{obs,i}})^2}, \tag{19}$$

$$\text{R}^2 = \left( \frac{\sum_{i=1}^{n}(y_{\text{obs,i}} - \overline{y}_{\text{obs,i}})(y_{\text{sim,i}} - \overline{y}_{\text{sim,i}})}{\sum_{i=1}^{n}\sqrt{(y_{\text{obs,i}} - \overline{y}_{\text{obs,i}})^2}\sqrt{(y_{\text{sim,i}} - \overline{y}_{\text{sim,i}})^2}} \right)^2, \tag{20}$$

$$\text{PBIAS} = 100[\%](1 - \beta). \tag{21}$$

**Table 2.** Performance evaluations for the daily neutron counts simulation with observed CRNS dataset.

| Indices | $\text{KGE}_{\alpha\beta}$ | KGE | NSE | $\text{R}^2$ | PBIAS |
|---|---|---|---|---|---|
| Range | $-\infty$ to 1 | $-\infty$ to 1 | $-\infty$ to 1 | 0 to 1 | $-\infty$ to $\infty$ |
| Optimal Value | 1 | 1 | 1 | 1 | 0 |
| Satisfactory Value | $> 0.70$ | $> 0.80$ | $> 0.50$ | $> 0.65$ | $< \pm 5$ |

# 3 Results

## 3.1 Constraining of the parameter distribution N0 / Sensitivity Analysis

The sensitivity and uncertainty analysis performed in this study use a Latin Hypercube Sampling (LHS) approach, resulting in parameter distributions that almost cover the entire prior defined hypercube of the individual parameters. The LHS approach creates a random value between the min and max values of the parameter set. Initial parameter ranges and exploratory model runs are set based on literature values (Boeing et al., 2022; Kumar et al., 2013b). Supplementary Table S1 shows the values for the parameters in all 10 000 simulations and the selected 31 behavioral simulations, with the posterior mean of the top 10 best parameters set. Among the calibrated parameters, the $N_0$ parameters are different in each method since this parameter has not exactly the same physical meaning in the Desilets and the COSMOS models.

In Fig. 4 the $x$-axis in each graph is fixed to the prior range of each individual parameter to facilitate the comparison of sensitive parameters. Most of the high-sensitive parameters show more peaked densities in a narrower range of parameter values, reflecting the significance of variations in model parameter values. The most sensitive parameters during the calibration period are $N_{0,\text{Des}}$, $N_{0,\text{COSMIC}}$, rootFractionCoefficient_pervious, and rootFractionCoefficient_forest of land cover classes are employed: class 1 = forest which consisted of permeable areas covered by coniferous, deciduous, and mixed forests; class 2



= impervious cover with land uses like settlements, industrial parks, roads, airport runways, and railway tracks; and class 3 = permeable cover covered by fallow lands, or those surfaces covered by crops, grass, and orchards. The calibration process notably sharpenes the Probability Density Function (PDF) of these significant parameters by eliminating some of the uncertainty linked to the variance in the prior probability distributions.

The uniform prior distribution range for $N_{0,\text{Des}}$ lies between 600 and 1500 cph, and $N_{0,\text{COSMIC}}$ lies between 100 and 400 cph. For agricultural sites such as *Cunnersdorf* and *Hordorf*, the $N_{0,\text{Des}}$ best estimate parameter results lie between 1000 and 1400 cph. Meanwhile, for *Hohes Holz*, the $N_{0,\text{Des}}$ parameter range lies between 800 and 1000 cph, the lowest value for the calibration parameter $N_{0,\text{Des}}$ is found between 800 and 900 cph due to the highest wet above biomass in the forest area. Similarly, for *Grosses Bruch* sites, the lowest value for the calibration parameter $N_{0,\text{Des}}$ is found between 800 and 900. The prior
parameter distribution $N_{0,\text{Des}}$ 600 and 1500 cph is the same for all four experiments.

The estimated values of $N_{0,\text{Des}}$ and $N_{0,\text{COSMIC}}$ obtained in our study are close to the optimal values, indicating that the model has the potential to generate accurate cosmic-ray soil moisture estimates even under dry conditions. But since at least COSMIC is physically based, a loss of the physical meaning of the parameters in question would be very critical. Table 3 provides detailed information on the mean and 95% confidence interval (CI) of the parameter values of the prior and posterior
simulated results of $N_0$. One of the important addition of this work is incorporating the lattice water account and we used the regionalization equation to calculate the lattice water which depends on the clay content with free parameters in Eqs. 2. The optimized parameter of the mHM shows the variation of the $\theta_{\text{lw}}$ ranges between (0.02–0.04) $\text{cm}^3\text{cm}^{-3}$ for different sites (see Supporting Information Table S3). This behaviour and the way it was defined in Eq. 1 indicate that $\theta_{\text{lw}}$ likely represents not only soil lattice water itself, but rather the total offset of all hydrogen pools in the vicinity of the sensor (see e.g., Schrön et al.,
2017; Iwema et al., 2021).





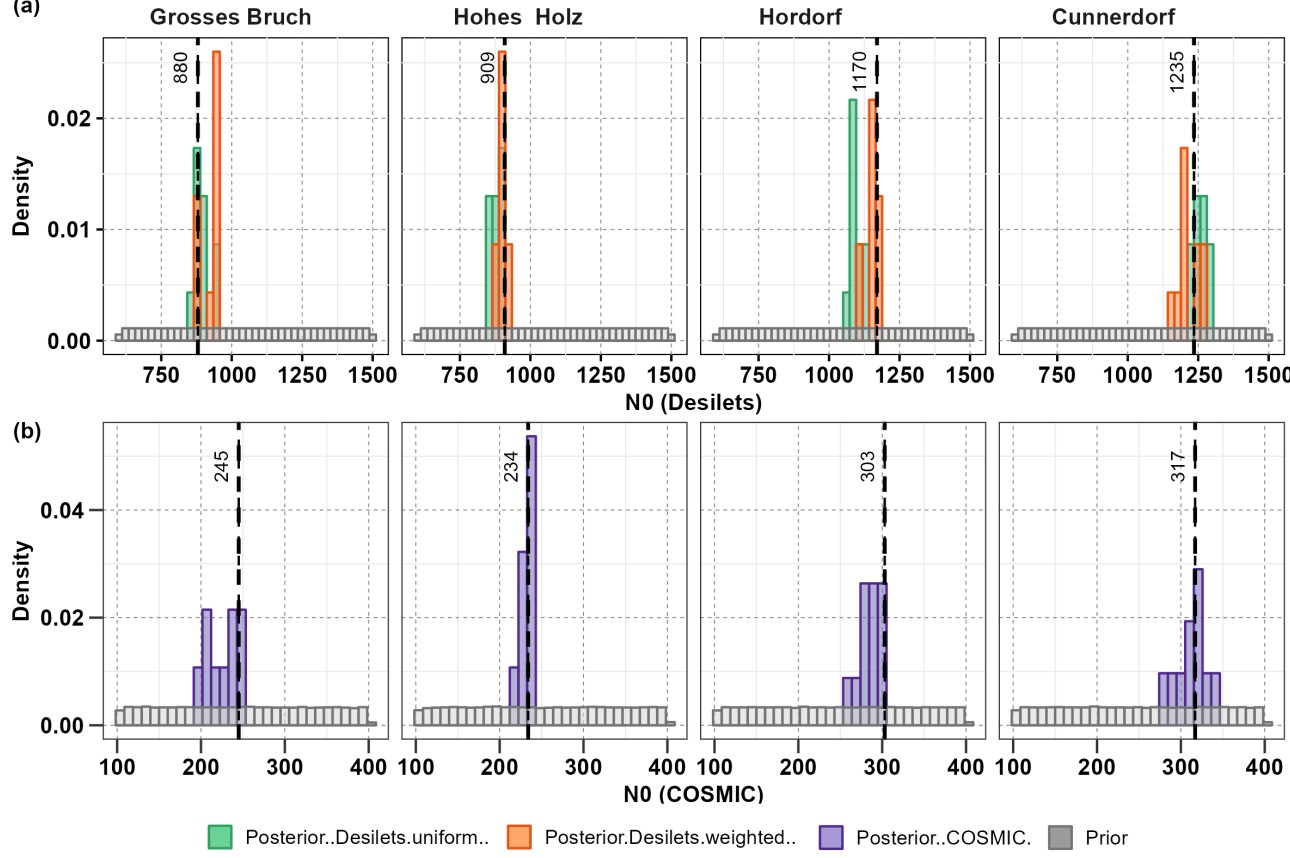

**Figure 4.** Probability Density Function (PDF) of the mHM parameter $N_0$ for two different approaches: (a) the Desilets method, and (b) the COSMIC method. The prior PDF of the original sample, consisting of 100 000 data points, is represented by the grey color. The behavioral PDF, obtained after applying the objective function, is shown for weighted (orange), uniform (green), and COSMIC (purple). The black dashed line represents the one $N_0$ value that best fits the data.





**Table 3.** The four most right columns are the posterior ones of size 100 000 and are the same for all sites. The posterior distributions correspond to the distributions in the behavioral sample obtained after the application of the objective function. The values correspond to the median ($Q_{50}$) and the lower and upper bound of the 95% confidence interval ($Q_{2.5}$ and $Q_{97.5}$, respectively).

| Parameter | CI | Prior | *Grosses Bruch* | *Hohes Holz* | *Hordorf* | *Cunnersdorf* |
|---|---|---|---|---|---|---|
| $N_{0(Des,uniform)}$ | $Q_{50}$ | 1050 | 889 | 882 | 1092 | 1255 |
| | $Q_{2.5}$ | 623 | 866 | 855 | 1069 | 1224 |
| | $Q_{97.5}$ | 1477 | 945 | 908 | 1126 | 1291 |
| | mean | 1050 | 899 | 883 | 1096 | 1257 |
| | sd | 260 | 27 | 21 | 21 | 25 |
| $N_{0(Des, weighted)}$ | $Q_{50}$ | 1050 | 942 | 906 | 1158 | 1209 |
| | $Q_{2.5}$ | 623 | 876 | 871 | 1116 | 1168 |
| | $Q_{97.5}$ | 1477 | 954 | 915 | 1176 | 1262 |
| | mean | 1050 | 925 | 899 | 1152 | 1216 |
| | sd | 260 | 34 | 16 | 21 | 34 |
| $N_{0(COSMIC)}$ | $Q_{50}$ | 250 | 225 | 234 | 287 | 316 |
| | $Q_{2.5}$ | 108 | 202 | 216 | 265 | 281 |
| | $Q_{97.5}$ | 392 | 249 | 241 | 302 | 339 |
| | mean | 250 | 226 | 232 | 285 | 312 |
| | sd | 87 | 18 | 9 | 12 | 19 |

## 3.2 Time series analysis of simulated neutron counts

The study conducts simulations of neutron counts in mHM using soil moisture parameterizations, with results presented in Figs. 5–6 across different sites. The simulated neutron counts were based on the simulated soil moisture content at the modeled soil horizons i.e., 0–5 cm, 5–25 cm, and 25–60 cm. The results of the ensemble runs show that the precision is higher for the behavioral simulation ensembles (represented by dark gray shaded areas) than in the unconstrained simulated data (represented by light gray shaded areas). We select the best 1 % with the highest KGE from 100 000 model runs, and the results are presented in Tab. 4. Furthermore, the behavioral simulation ensembles captured more variations in the COSMIC method compared to the Desilets method after the application of the objective function (i.e., $KGE_{\alpha\beta}$).

The $N_{COSMIC}$ method performs best at the forest site (*Hohes Holz*), whereas at the agricultural sites (*Hordorf* and Cunnersdorf), the $N_{Des,W}$ method performs slightly better. Only for the grassland site (*Grosses Bruch*), the uniform method $N_{Des,U}$ slightly outperforms the other two methods i.e., $N_{Des,W}$ and $N_{COSMIC}$, while overestimating the observations at all the other sites. The better performance of $N_{COSMIC}$ and $N_{Des,W}$ over $N_{Des,U}$ demonstrates the benefits of explicitly resolving individual soil moisture profiles, bulk densities, and lattice water, as opposed to a uniform average across the layers. This perception, however, might depend on site-specific soil profile characteristics and be less prominent if profiles are largely uniform or in-



correctly resembled by the model structure. In general, we observe good model performance for all methods indicated by a correlation coefficient greater than 0.80 and a percent bias (PBIAS) below 2 % across the majority of investigated sites and methods.

These results suggest that the neutron-forward models match the observed neutron counts well. However, the mean ensemble had difficulties reproducing the neutron counts for the *Grosses Bruch* site in all three methods. One of the primary sources of uncertainty at the *Grosses Bruch* site is surface ponding and shallow groundwater, as well as the loamy texture of the soil. Those factors contribute to the formation of permanent water ponds in the area and may introduce uniform or even inverse soil moisture profiles which directly influence the neutron emissions, but cannot be captured by the mHM model. Another factor is the time-variable effect of crowding cows near the station, which may influence the CRNS signal, but is challenging to correct in the CRNS measurement (Schrön et al., 2017).

We incorporate the CRNS parameter set in mHM, and some parameters related to soil moisture and neutron counts are effectively constrained based on the objective function using $KGE_{\alpha\beta}$. However, there is still room for improvement, particularly with regard to the coefficient in root fractions distributed across soil layers. The incorporation of dynamic vegetation in models is important as it can impact the model parameter LAI, which in turn can affect root water uptake and soil water content. Currently, these factors are not considered in the models, leading to a permanent and systematic shift in these variables each year.

The results also highlight the uncertainties associated with model simulations and the sensitivity of the objective function. We find that three soil moisture-related parameters, namely $N_0$, *rotfrcoffpre*, and *rotfrcofforest*, have the most significant impact on the objective function $KGE_{\alpha\beta}$, compared to the other parameters of mHM. The parameter $N_0$ directly affects the neutron count simulations, while the parameters *rotfrcoffpre* and *rotfrcofforest* correspond to the fractions of vegetation roots in different soil layers that directly affect the water availability related stress for the estimation of actual evapotranspiration, and thereby the soil-water dynamics (Samaniego et al., 2010b; Kumar et al., 2013b). The best parameter set values in mHM across all sites and methods are given in (Supporting Information Table S2).



**Figure 5.** Simulated daily time series of black for $N_{(Des,W)}$, red for $N_{sim(Des,Uni)}$ for the four sites. The black lines represent the median of the behavioural simulation ensembles that satisfy the objective function which is LHS10 ensemble members. The light grey shaded areas represent the 95% CI of the simulation ensembles corresponding to different levels of constraining which is LHS1000 ensemble members, and the observation is shown in grey points. Precipitation is shown in blue color on the top.





**Figure 6.** Simulated daily time series of $N_{(\text{COSMIC})}$ for the four sites. The black lines represent the median of the behavioural simulation ensembles that satisfy the objective function which is LHS10 ensemble members. The light grey shaded areas represent the 95% CI of the simulation ensembles corresponding to different levels of constraining which is LHS1000 ensemble members, and the observation is shown in grey points.

## 3.3 Model calibration statistics and evaluation

In addition to $\text{KGE}_{\alpha\beta}$, the four metrics KGE, NSE, R, and PBIAS are used to evaluate further the mHM neutron counts
340 simulated with observed CRNS data. We employ LHS to generate a parameter sample of 100 000 for the three methods, namely $N_{\text{Des,U}}$, $N_{\text{Des,W}}$ and $N_{\text{COSMIC}}$, by uniformly distributing the ranges provided in the (supplementary Table S1). The top 10 parameter sets are found to perform satisfactorily with a KGE range of 0.80 to 0.93, as demonstrated in Table 4. The





calibrated parameter sets obtained from different objective functions are also evaluated and compared using various statistical indices, as shown in Figure 7, with most objective functions performing better than satisfactory based on the criteria in Table 2. The results for the COSMIC method indicate that the main contribution to poorer results during the evaluation period was due to the variability term ($\alpha$). The boxplot displayed in Figure 7 illustrates the threshold achieved by the top 1000, 100, and 10 LHS members, along with the corresponding percentage of the best 10 LHS parameter sets that meet the threshold, as specified in (see Tab. 2). Among the 31 parameters selected to simulate neutron counts, this plot provides an overview of the distribution of results and their variability with respect to the threshold criteria.

**Table 4.** Model Performance using Percent bias (Pbias), coefficient of determination ($R^2$), Nash-Sutcliffe efficiency (NSE), and Kling-Gupta Efficiency (KGE) for the calibration for the period from 2014-2021.

| **Sites** | *Grosses Bruch* | | | *Hohes Holz* | | | *Hordorf* | | | *Cunnersdorf* | | |
|---|---|---|---|---|---|---|---|---|---|---|---|---|
| Methods: | Des,U | Des,W | COSMIC | Des,U | Des,W | COSMIC | Des,U | Des,W | COSMIC | Des,U | Des,W | COSMIC |
| $KGE_{\alpha\beta}$ | **0.97** | 0.96 | 0.93 | 0.96 | **0.99** | **0.99** | **0.99** | **0.99** | 0.98 | 0.93 | **0.99** | **0.99** |
| KGE | **0.85** | 0.83 | 0.81 | 0.81 | 0.80 | **0.85** | **0.87** | **0.87** | 0.86 | 0.88 | **0.93** | 0.90 |
| NSE | **0.69** | 0.60 | 0.29 | 0.54 | 0.60 | **0.70** | **0.75** | **0.75** | 0.73 | 0.57 | **0.85** | 0.79 |
| $R^2$ | **0.73** | 0.69 | 0.66 | 0.66 | 0.65 | **0.73** | **0.76** | **0.76** | 0.75 | **0.86** | **0.86** | 0.81 |
| R | **0.85** | 0.83 | 0.81 | 0.81 | 0.80 | **0.85** | **0.87** | **0.87** | 0.86 | **0.93** | **0.93** | 0.90 |
| PBIAS | **0.7%** | $-1.3\%$ | $-3.3\%$ | 2.0% | **0.3%** | 0.4% | **0.1%** | **0.1%** | 0.3% | 6.0% | 0.7% | **0.5%** |







**Figure 7.** Evaluation of model performance using boxplots constraining of 1000 to the best 10 parameters set at four different sites, using three different methods, namely $N_{(Des,W)}$ in red, $N_{(Des,Uni)}$ in golden, and $N_{(COSMIC)}$ in purple. The figure presents four subplots, where (a) represents Alpha, (b) Beta, (c) KGE$\alpha\beta$, and (d) Kling-Gupta efficiency (KGE) and its components, i.e., the variability term (perfect value: 1), and bias term (perfect value: 1), respectively.

## 3.4 Evapotranspiration evaluation at eddy covariance stations data against mHM simulation

The ensemble model of (10 members) simulations, is further examined with the evapotranspiration (ETa) to cross-evaluate and assess the model's ability to represent other fluxes and states next to neutron counts by using the ETa observational data from eddy covariance measurements Integrated Carbon Observation System (ICOS) at *Hohes Holz* Warm Winter (2022). In terms



of temporal dynamics, the model is able to capture the observed ETa quite well at the study site, as shown in Figure 8. Panel
(c) displays the scatter plot that reveals no systematic over or underestimation of the observed actual evapotranspiration. The
temporal dynamics of the model-simulated evapotranspiration are in good agreement with the observed data from the *Hohes
Holz* forest eddy covariance site, taken from Warm Winter (2022), as illustrated in Figure 8a. The daily correlation between
observed and simulated evapotranspiration is observed high in the growing season at $r = 0.84$, whereas the lowest correlation
is found in the non-growing season at $r = 0.65$ in Figure 8c. The highest deviation in terms of RMSE is observed during
summer when the highest fluxes occur, and the lowest during winter, in which the contribution of ETa is lowest.

In Figure 8b, the prior and posterior parameter distributions of evapotranspiration for *Hohes Holz* are displayed. The prior
distribution represents the 10 000 parameters set utilized for the neutron counts simulation under Latin Hypercube Sampling
(LHS). The results demonstrate that the ensemble model of 10-member simulations (posterior) for neutron counts can also
effectively capture evapotranspiration, exhibiting a root mean square error (RMSE) of 0.76 mmd$^{-1}$ of the growing season and
0.25 mmd$^{-1}$ for non-growing when compared to observed ICOS data and simulated mHM. When compared to the model
simulations with a-priori parameter sets, we notice a substantial improvement in ET simulations (mean RMSE of 0.86 mmd$^{-1}$
to 0.76 mmd$^{-1}$). Furthermore, the RMSE range is also narrower for the posterior simulations compared to the prior ones which
further demonstrated the additional value of incorporating CRNS measurements in improving the consistency of both modeled
soil moisture and evapotranspiration estimates.

The modeled evapotranspiration is highly dependent on soil parameterization because soil water is the main source of
evaporative water. During the growing season (summer), the model exhibited the largest variability in modeled ETa (Figure 8c).
This can be associated among other things with a lack of a dynamic vegetation growth module in mHM, which may not capture
the onset of the vegetation period adequately. This variability could also be attributed to seasonal changes in vapor pressure
difference (VPD) or more localized processes occurring at the forest site (e.g., under-story vegetation), which are currently not
considered in the model. Nevertheless, the overall agreement between modeled and observed ETa is reasonably good; and the
analysis reveals further improvement of model performance in the growing season.



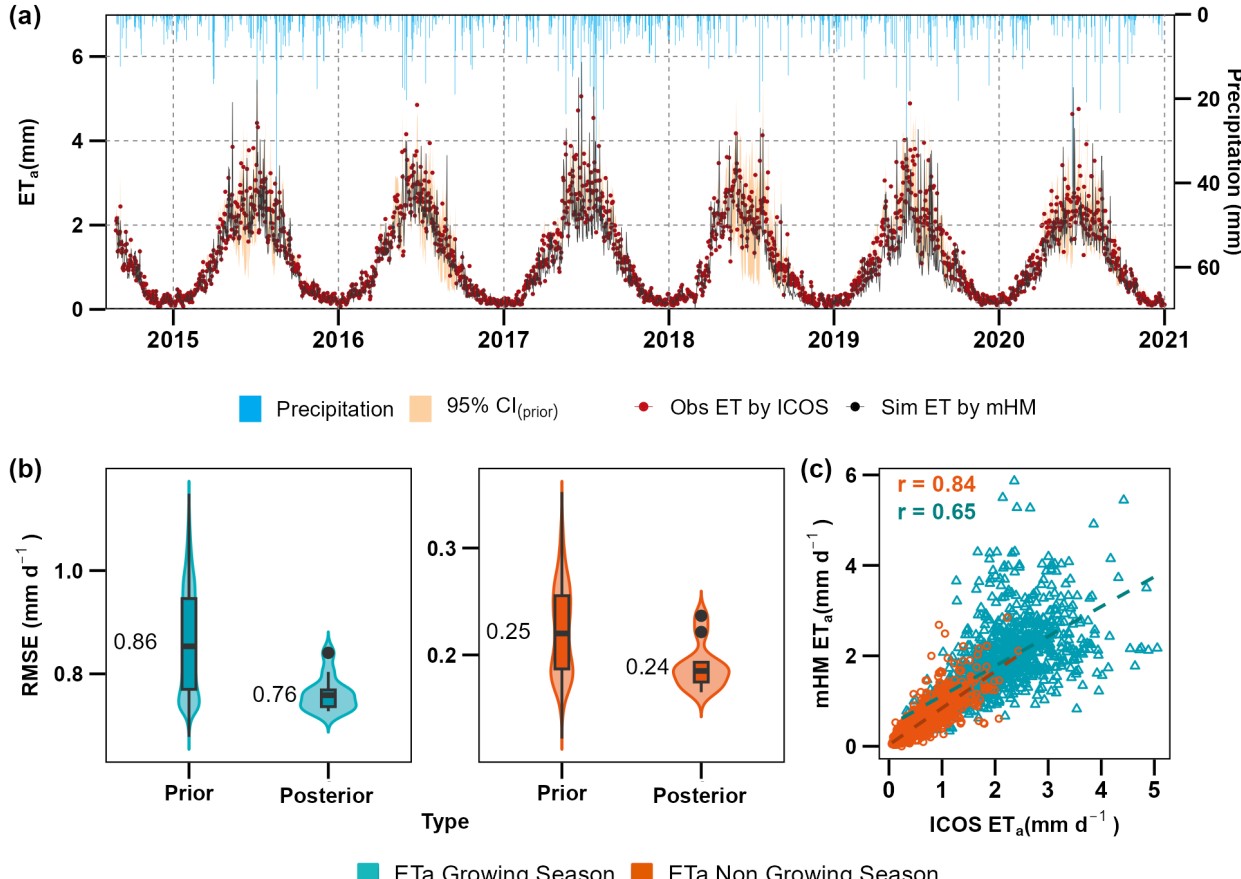

**Figure 8.** (a) Time series of actual evapotranspiration (ETa) from mHM (black), ICOS measurement (red), and the prior range of 100 000 realizations in (orange) color. (b) Boxplot of daily actual evapotranspiration (ETa) differences between the growing and non-growing seasons, comparing two selected prior with 10 000 simulations, the values represent the mean of the statistical metrics and posterior with 10 ensemble member distributions using the root mean square error (RMSE) as the evaluation metric ($\mu$gm$^3$). (c) scatterplots of modeled vs. observed ET on a daily basis from ICOS during the growing season from March to August (green) and non-growing season from September to February (brown) at *Hohes Holz* eddy covariance station in a forest.

## 4 Discussion

This study assessed the suitability of CRNS observations at four sites to enhance soil moisture representation in mHM. The model's depth-dependent simulations allow the estimation of neutron penetration depths, which are typically 5–60 cm and
380 more sensitive to shallow soil moisture Köhli et al. (2015). Our findings contribute to leveraging CRNS data for improved hydrological modeling.



To improve the soil moisture profile representation within mHM it is a major challenge to use a single vertically integrated CRNS measurement. In order to have a fair comparison between the model and observed CRNS data, two conceptually different approaches were integrated into mHM to calculate neutron counts from different SWC horizon depths i.e., an empirical method based on Desilets et al. (2010), and a physics-based method (COSMIC) based on Shuttleworth et al. (2013). Since the empirical method is described by an analytical expression, taking into account the uniform average of the soil moisture layers, it is straightforward to implement and therefore most commonly used (Zreda et al., 2012; Rivera Villarreyes et al., 2011; Andreasen et al., 2017; Bogena et al., 2022). However, the method comes with the risk of missing representation of the vertical profile of soil properties and water content. Therefore, we extended this uniform-averaging scheme by a vertical weighting scheme to mimic the sensitivity of the neutrons to the upper layers. The COSMIC operator also accounts for the full soil moisture profile, but in a more physically behaved manner, following the track and attenuation of the neutrons in and out of the soil column. The mHM model is now able to simulate neutrons directly with all three approaches. The presented results confirmed general consistency with CRNS observations at four sites in Germany (Figs. 5 and 6). Agricultural land presents a valuable opportunity to examine the interaction between soil moisture dynamics, crop growth, irrigation methods, and vegetation dynamics. *Hordorf* and Cunnerdorf are specific agricultural sites where seasonal changes in aboveground biomass are expected to be larger due to crop growth and harvest compared to grassland and forest sites. The study by Schrön et al. (2017) found that the revised weighting strategy for CRNS data improved the accuracy of soil moisture predictions at agriculture sites, but there is still room for improvement in capturing local dynamics through revised parameters in the CRNS model. Our results showed that at the agriculture site, the $N_{0,\mathrm{Des,W}}$ methods in mHM slightly out-performed the other methods.

We also investigated *Hohes Holz*, a forest site, and observed an early simulation of approximately 28 days in the simulation of neutron counts compared to the observations. The early simulation phase could be attributed to the limitation of mHM in simulating the dynamics of detailed vegetation mechanisms Zink et al. (2017). One specific limitation is that the model does not fully account for the fact that trees at the site have access to deeper water sources, which can result in water stress being experienced at later times. Still, we get very good results in terms of KGE, for instance, indicating that these issues are of minor importance and that all three methods in mHM representation of the forest are already performing quite well. Also, the CRNS method may be influenced by temporal biomass variation in the forest (Baatz et al., 2015), but many recent studies have confirmed the good performance of CRNS in forests compared to below-ground soil moisture profiles, indicating that the dynamic vegetation effect is just a minor observational issue (Bogena et al., 2013; Andreasen et al., 2017; Schrön et al., 2017; Boeing et al., 2022; Bogena et al., 2022). It is worth noting that most of the studies on drought analysis look at the anomaly of soil moisture, while our study tries to assess the absolute soil water quantity and the properties that can determine the soil water content. While CRNS and TDR generally agree at this site, the discrepancy shown in our results could be attributed to issues related to process representation in mHM Boeing et al. (2022). Incorporating CRNS data into soil moisture analysis can enhance the accuracy and precision of absolute soil moisture measurements. Future research can focus on exploring the potential relationships between CRNS data and soil moisture anomalies, thus furthering our understanding of the dynamics of drought and assisting in the development of efficient drought monitoring and mitigation strategies.



To cross-evaluate our results, the mHM simulations of evaporation are tested against observational data from eddy covariance measurements ICOS (Warm Winter, 2022; Pohl et al., 2023) at the *Hohes Holz*. Figure 8 shows the scatter including the seasonal correlation coefficient at the forest site. The results indicate low correlations in summer, likely due to mHM's limitations in capturing evapotranspiration values with mHM's static vegetation module. However, the model performs well in
winter, with a high correlation between observed and simulated values of evapotranspiration, the results confirm the findings from Zink et al. (2017), who used mHM to estimate evapotranspiration, groundwater recharge, soil moisture, and runoff with 4 km spatial and daily temporal resolutions (1951–2010). They found deviations in modeled versus observed evapotranspiration, particularly in spring and in cropland areas, however, soil moisture estimations were in good agreement with observed dynamics. The study highlights the importance of considering seasonal variations when analyzing the results. Discrepancies,
such as low correlations in summer, indicate the need for improvements in capturing evapotranspiration dynamics under varying environmental conditions. Refining vegetation dynamics representation could enhance the simulation of evapotranspiration processes. Additionally, the agreement between mHM and observed soil moisture dynamics suggests variable model performance for different hydrological variables, emphasizing the need for a comprehensive assessment of its capabilities across various environmental conditions and spatiotemporal scales.

It is worth highlighting here that the incorporation of neutron data into mHM not only improved the soil moisture estimation in the model, but also the estimation of evapotranspiration. This provides major evidence that CRNS data has the potential to improve hydrological process understanding as a whole.

The *Grosses Bruch* site stands out as a mesophilic grassland site with a nearby water channel, shallow ground water, regular cattle grazing and seasonal flooding (Hermanns et al., 2021). We find a large ensemble-related uncertainty at this site for all
three methods, while the uniformly weighted approach $N_{(\text{Des,U})}$ shows a slightly better performance than the other two methods $N_{(\text{Des,W})}$ and $N_{(\text{COSMIC})}$ (see Table. 4). The behaviour may result of a missing representation of locally significant hydrological components, such as dynamic biomass, snow, shallow ground water, or nearby surface ponding (Schrön et al., 2017). Moreover, in the middle of September, many cows had been present at this site, which could have led to non-negligible variation of the neutron signal and thus to a non-meaningful expression of correlation-related measures Schrön et al. (2017). Döpper et al.
(2022) mentioned thigh impact of grazing on the plant traits and soil properties at this site. Additionally, the use of one grid cell measurements by mHM in our study may have limited the accuracy of our results, as the depth of measurement may not be representative of the entire soil profile. Notably, neutron counts were found to provide a more accurate representation of soil water content during June, July, August, and September, when levels tend to be lower. Further exploration of neutron counts may yield additional improvements to model performance.

Overall, the three methods ($N_{\text{Des,U}}$, $N_{\text{Des,W}}$, and $N_{\text{COSMIC}}$) in mHM were able to consistently simulate the neutron count variability throughout the available data period. The simulated time series tended to slightly underestimate the CRNS neutron count rate, particularly during the dry season. This effect could be explained by the known limitations of the equations under very dry conditions, while recent approaches exist (Köhli et al., 2021) that could lead to further improvement in future studies. Nevertheless, the results generally confirmed the better performance of the $N_{\text{Des,W}}$ than $N_{\text{Des,U}}$, because of its more realistic
representation of neutron propagation with depth. After optimizing the soil hydraulic properties based on CRNS data, the



integrated signal was reproduced very well (Fig. 5). We also included offset hydrogen pools in the form of lattice water to the $N_0$ calibration function, which was important for more accurate soil moisture estimates, confirming initial suggestions by Bogena et al. (2013). Moreover, a strong correlation between biomass and the $N_0$ parameter was reported in several studies (Franz et al., 2013; Hawdon et al., 2014; Baatz et al., 2014, 2015). In our study, we pass the $N_0$ parameter as a calibration

parameter set in mHM. In using the CRNS soil moisture measurement the drier locations show larger deviations than the wetter locations (Iwema et al., 2015). The possibility of using simulated high-resolution soil moisture profiles instead of a few measurements at different soil depths could further increase the accuracy of the model predictions (Brunetti et al., 2019).

According to Beck et al. (2021), model calibration provides more overall benefits than data assimilation. Furthermore, model calibration can be advantageous for regions with both sparse and dense rain gauge networks, whereas data assimilation is more

beneficial for regions with sparse rain gauge networks. In this paper, the Latin Hypercube Sampling (LHS) method McKay and Conover (1979) is adopted to generate input variable samples, which is a stratified sampling method that reduces the number of simulations required compared to the conventional Monte Carlo method Iman and Helton (1988). LHS divides the range of each input into N intervals and selects one representative value from each interval to ensure full coverage of the input variables range and representation of all possible values in the simulation. Previous studies by Smith et al. (2019) and Liu

et al. (2022) addresses the challenges of using the original KGE in Markov chain Monte Carlo (MCMC) methods, offering insights for accurate parameter estimation and posterior distribution exploration. To address this issue, it is recommended to use adaptations to the LHS method instead of directly using the original KGE to improve the exploration of the posterior distributions. Our approach can estimate the posterior distributions of model parameters based on the objective function $KGE_{\alpha\beta}$ by taking the variance and bias.

This paper provides a framework to incorporate CRNS data into the mHM to assess the accuracy of soil water content on different land cover types, including agricultural land, deciduous forest, and grassland. The integration of methods from Desilets et al. (2010) and Shuttleworth et al. (2013) in mHM, using climatic data and soil physical parameters, the parameterization of soil moisture and evapotranspiration can effectively improved (see Fig. 9). This framework lies in its ability to utilize observed neutron counts data, allowing for a comprehensive assessment of the model's performance and enhancing its reliability in

hydrological modeling.



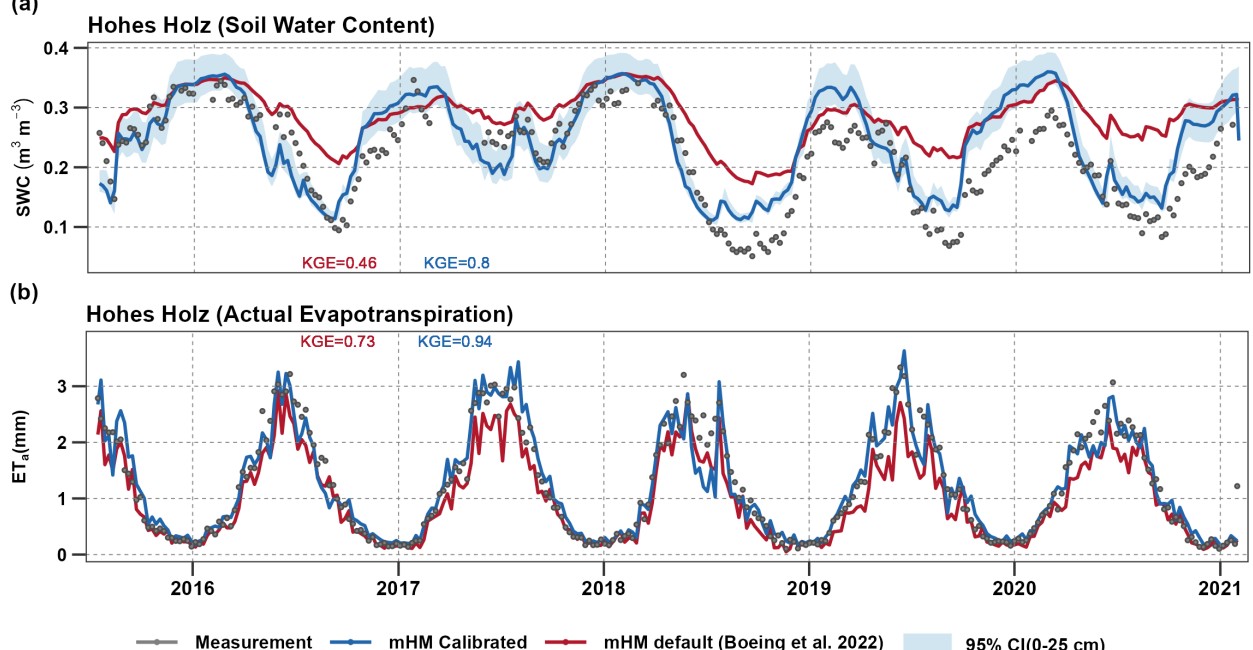

**Figure 9.** (a) Comparison of weekly soil water content time series for 2014-2021 at a depth of 0-25 cm for *Hohes Holz* CRNS-based soil moisture data (grey line) and simulated data from mHM using two different sets of parameters. The red line represents the default parameters used in the German drought monitoring system as described by Boeing et al. (2022), while the blue line represents the best-calibrated value of our study with the improved Kling-Gupta Efficiency (KGE). (b) Comparison of weekly observed actual evapotranspiration (grey dots) and simulated actual evapotranspiration using the default mHM parameters by Boeing et al. (2022) (red line) and the calibrated simulation (blue dots) over the *Hohes Holz* site.

The findings of this study suggest that, with certain cautionary notes, our method has the potential to better characterize drought compared to the approach used in the German drought monitoring system. Our assimilation of neutron counts in mHM resulted in more accurate predictions of soil water content compared to the approach described by Boeing et al. (2022) that is currently employed in the German drought monitor. However, the fact that our method still overestimated the severity of 480 the drought in 2018 and 2019 is noticed, indicating that there may be other processes not accounted for by mHM that require further investigation.

# 5 Conclusion and future outlook

This study evaluates the potential of the mHM a large-scale hydrological model for simulating neutron counts at the $0.01562° \times 0.01562°$ grid scale across different land cover sites for the period 2014–2021. Two empirical and one physical model ap-



proaches are evaluated for deriving neutrons from the soil moisture profile. Neutron measurement data from four sites in Germany are integrated, and the influence on hydrological model parameters, as well as simulated soil moisture and evapotranspiration are analyzed. The parameter sample of $100\,000$ realizations for neutron counts was taken, which are analyzed regarding their uncertainty caused by the parameter estimation. The parameter sets are filtered based on the KGE of observed vs simulated neutron counts. The best 1% member ensemble simulations are evaluated with neutron counts, evapotranspiration, and soil moisture observations.

The evaluation of neutron counts at four different sites yield a KGE value of $> 0.8$, indicating a satisfactory representation of the neutron observations. The 1% ensemble parameter set is found more representative of $100\,000$ realizations, suggesting a reliable model performance. The performance of the neutron counting methods varies across different land cover types. The $N_{\mathrm{Des,W}}$ method generally demonstrates good performance, particularly at the agricultural sites. While the $N_{\mathrm{COSMIC}}$ method performs slightly better at forest site and the $N_{\mathrm{Des,U}}$ method shows slightly better results at the grassland site.

However, there is still room for improvement in some areas. Specifically, working with grassland sites presented challenges, particularly with the $N_{\mathrm{COSMIC}}$ method. On the one hand it is a physics-based approach incorporating a comprehensive representation of the neutron counting process, but on the other hand, it relies on the detailed representation of the site characteristics in the hydrological model. This complexity could introduce additional uncertainties and limitations in the model, potentially affecting its performance, especially when the site is more complex than it has been modeled. The study suggests that the observed discrepancies between model and observations may be attributed to the representation of dynamic biomass, snow, surface ponding, and shallow groundwater dynamics, which are present at the grassland site, for instance. Addressing these features could further enhance the model's accuracy.

The evaluation with evapotranspiration from eddy covariance at *Hohes Holz* stations indicates deficiencies in mHM to deal with forest systems, but also great potential for CRNS measurements to improve the water partitioning as a whole. Especially in the growing season (March-August), deviations of the modeled and observed ETa indicate room for better representation of mixed soils and dynamic vegetation modules at the local scale within mHM. The calibration on neutron counts not only improved the soil moisture performance of the model, but also helped to set the modeled evapotranspiration straight.

In conclusion, the incorporation of neutron counts estimation into mHM by accounting for vertical soil moisture profiles improves the model's accuracy and provides a more realistic representation of soil moisture dynamics as well as evapotranspiration, particularly at the forest site. This research presents a direction for future studies to explore. The next step in this research is to evaluate the ability of this CRNS module in mHM for estimating soil moisture through a large-scale soil moisture monitoring initiative, e.g. by utilizing more stationary CRNS networks or the novel rail-based CRNS data from Altdorff et al. (2023). Improving the model predictions will contribute to reducing the uncertainties associated with drought and flood management strategies and informed agricultural decisions.

*Code availability.* Simulation data is attached as supplemental material. The mesoscale Hydrological Model mHM (version 5.12) is an open-source and can be freely accessed from GitLab: https://git.ufz.de/mhm/mhm/-/tree/v5.12.0?ref_type=tags.





*Data availability.* We kindly acknowledge the German Weather Service (DWD) for providing the meteorological datasets. The terrain eleva-
tion data was collected from USGS EROS Archive - Digital Elevation - Global Multi-resolution Terrain Elevation Data 2010 (GMTED2010),
available at https://www.usgs.gov/centers/eros/science/usgs-eros-archive-digital-elevation-global-multi-resolution-terrain-elevation. Grid-
ded soil characteristics are based on the BUEK200 database obtained from the German Federal Institute for Geosciences and
Natural Resources (BGR, see online at https://geoportal.bgr.de/mapapps/resources/apps/geoportal/index.html?lang=en#/datasets/portal/
154997F4-3C14-4A53-B217-8A7C7509E05F). The geological dataset was downloaded from Institute for Biogeochemistry and Marine
Chemistry, KlimaCampus, Universitt Hamburg (https://www.geo.uni-hamburg.de/en/geologie/forschung/aquatische-geochemie/glim.html).
Leaf Area Index (LAI) dataset was downloaded from the Global Land Cover Facility (GLCF), available at http://iridl.ldeo.columbia.edu/
SOURCES/.UMD/.GLCF/.GIMMS/.NDVIg/.global/index.html. The land cover dataset was downloaded from the European Space Agency
(ESA), available at http://due.esrin.esa.int/page_globcover.php. The ET data were obtained from https://zenodo.org/record/7561854.

*Competing interests.* RK and LS are members of the editorial board of Hydrology and Earth System Sciences.

*Acknowledgements.* The authors thank all the site owners for maintaining the local sensors, particularly to F. Böttcher (DWD) and E.
Thiel (SKWP). The study has been made possible by the Terrestrial Environmental Observatories (TERENO), an infrastructural fund of the
Helmholtz Association. The High-Performance Computing Cluster EVE has contributed to the computation of the scientific findings. Eshrat
Fatima is grateful for the financial support of the German Academic Exchange Service (DAAD) through the Graduate School Scholarship
Program under Reference Number 91788160.



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
