# Peer review of "Improved representation of soil moisture processes through incorporation of cosmic-ray neutron count measurements in a large-scale hydrologic model"

_EGUsphere, 2023_

## Referee Comment (RC1)

The authors have investigated how cosmic-ray neutron soil moisture data can help improve the simulation of both soil moisture and evapotranspiration with a hydrological model. They used three relevant methods to incorporate neutron counts into large-scale hydrological modelling and compared their performance. Using cosmic-ray neutron soil moisture data to calibrate the hydrological model improved the simulation of both soil moisture and evapotranspiration.

I have found the manuscript interesting and mostly well written. The significance of the work is clear to me. I do have a few suggestions to make the contribution of the presented research stronger. These include both suggestions to improve readability and some suggestions to solidify the outcomes with some more elaborate explanations of the methodologies and a few small additional analyses.

**Major comments:**

1. Please, consider shortening the paragraphs of lines 32-48, 49-60, and 61-72 of the Introduction on pages 2 and 3, to help the reader understand the story line better. It is now a broader literature review than might help the reader to get the key message. Some references that are highly relevant can (and in many cases do already) enter the story in the Results and Discussion.

2. Introduction, page 3, L83: "… neutron counts at scales of 1.2 x 1.2 km2" and Conclusion and future outlook, page 26, lines 483-484 "… for simulating neutron counts at the 0.01562o x 0.01562o grid …". Please, clarify in which way the neutron count simulation is evaluated at this scale. Scale mismatch between model grid cell size, different model inputs, and different model calibration/validation/data assimilation data should be an important aspect of this study. Please, include a discussion on the impact of scale mismatches in the manuscript. Clarification can be done in the Introduction and/or Materials and Methods and Discussion.

3. If the model produces other output than soil moisture and evapotranspiration, meaning other water fluxes, can the authors discuss how the estimation of these fluxes changes under calibration with CRNS-data? If observations are available, please include these in comparison, or at least mention such analyses as recommendation for next research steps. It is important to verify that other model outputs do not deteriorate, or better, actually improve simultaneously with evapotranspiration simulation.

4. A comparison with in-situ soil moisture observations is now briefly discussed in the Discussion, page 23, line 411. I suggest that the authors move this forward and make it more prominent by showing a comparison in a figure and expand the discussion. If insitu soil moisture data were available at the other sites, these should be discussed too. If such data are not available, please mention this explicitly. Given the grid cell size of >1 km, satellite remote sensed soil moisture data is relevant too. Please discuss the relevance of CRNS data compared to satellite data at this modelling scale. To my opinion, this issue should be discussed. Implementing actual calibration and/or validation/ data assimilation with point scale soil moisture data and satellite remote sensing soil moisture data, I think should be a recommendation in the final chapter of this manuscript and should be considered by the researchers as interesting future work.

5.  Results section, page 12, lines 269-275, to the reader it is now not crystal clear which parameters were calibrated? Just the neutron related parameters or also other mHM parameters? Please clarify this textually and include a manuscript main text table (or other mechanism) to make this instantly clear.

6.  Please, consider creating either one section Results and Discussion, or move bits of preliminary discussion (p 15, lines 312-315, 318-324, 325-330, 370-376) from the Results section to the Discussion section.

**Other, specific comments:**

7.  Title: "Improved representation of soil moisture simulations..."

I doubt if the word "representation" in relation to "soil moisture simulations" is well chosen. 'representation of soil moisture processes' or 'representation of soil moisture measurements' sounds logic, but here it seems as if the representation of soil moisture simulations is improved. Please think if this is really what you mean and if so, please consider if will be understood by the wider audience.

8.  Abstract: P1,L12-14: "A Monte Carlo simulation with Latin hypercube sampling approach …" Please, consider removing this sentence or writing it in more understandable wording for the audience. It is now hard to see the exact relevance of the technical details given, like 'N = 100 000'. What does such number tell the audience?

9.  Abstract, P1,L15-17: "We find that the non-uniform weighting scheme in the Desilets method provide**s** the most reliable performance, whereas the more commonly used approach with uniformly weighted average soil moisture overestimates the observed CRNS neutron counts"

    How did COSMIC perform compared to the two Desilets methods?

10. Introduction, page 2, L49-41: Please, improve textually by building a logical bridge between the paragraph of lines 32-48 and of lines 49-69. As is, HYDRUS-1D is

introduced suddenly and in a way that makes is seem as a very key model, without being clear why so.

11. Introduction, page 3, L73-74: The word "Eventually" and the wording with which the mHM model is introduces, at the start of this sentence and paragraph, make it seems as if the mHM model is a key hydrological model, that is the logical end-point of a discovery process and that is the standard that every reader should instantly know. It might be a well-known model, but it is one of many. Please, to help the reader understand the position of the the mHM-model, rewrite this to a more neutral wording.

12. Introduction, page 3, L83: "*The COSMIC method is complex ...*". What is meant by complex here? Please, clarify for the reader.

13. Materials and Methods, page 4, L105: "four sites". Why does this number differ from that on line 90 of the introduction (page 4), which says "three"?

14. Materials and Methods, page 4, line 111: "… producing methane fluxes". How is this relevant to the research presented? If it is relevant, it should be mentioned here and maybe discussed later on.

15. Materials and Methods, page 5, Table 1: Please say in the caption that precipitation and temperature are yearly averages and in the table itself, say '[mm/year]' for Precipitation.

16. Materials and Methods, page 6-7: The first reference to figure 2 is now on line 151. I think that by referring the reader earlier (from line 138 onwards), it will be easier for them to understand the methodology, with this key figure in hand.

17. Materials and Methods, page 7, figure 2: In this figure, a 'Neutrons' module now appears in the upper part (modules of mesoscale hydrologic model mHM) and below, where the different neutron models are mentioned. Is this how the modelling actually works? Is there one neutron module in the mHM and then, the outcomes (neutron counts) of these are fed to the neutron models? Please adjust the figure and/or make very clear in the manuscript text how the different bits are actually connected. In addition, please clarify if the short arrows between the left and right bits connect 'Spatial Data' to 'Model Setup' and 'Model Setup' to 'Performance Matrix' or if the connections are actually 'Spatial Data' to 'mesoscale hydrologic model' and 'CRNS-methods output' to 'Performance Matrix'

18. Materials and Methods, page 8, lines 158-159: "We compared these simulated values with the measured soil water content obtained through CRNS" This suggests soil moisture values were compared. Is this true or were actually neutron counts computed from mHM soil moisture simulation compared to neutron counts?

19. Materials and Methods, page 8, 2.4.1: Please, restructure this paragraph, such that parameter names are mentioned after this equation, to improve the readability.

20. Materials and Methods, page 9, lines 185-186: "Organic water equivalent …". Please rephrase this sentence.

21. Materials and Methods, page 10, line 204: "… does not get too small and SWC is not too high". Please, quantify.

22. Materials and Methods, page 11, line 234: COSMIC parameter alpha is mentioned here, but was also mentioned on line 221. This seems confusing. Please, check and improve/clarify.

23. Materials and Methods, page 11, lines 237-238: The parameters within the formula on line 237 seem not to match the parameters on line 238.

24. Materials and Methods, page 11, lines 248-249: "However, we modify KGE (Eq. 15) by removing the correlation coefficient rho, as it is just a measure of temporal signature and is largely dominated by seasonality alone". Why should seasonality not be included? Why is the correlation coefficient not relevant? Please, clarify this better for the reader.

25. Results, p12, line 273: "… in all 10 000 simulations": why was this number chosen? How do we know it is sufficient, insufficient, or too large? If only the N-parameters from the neutron models were calibrated, this seems like a large number.
26. Results, p12, line 270: "… parameter distributions that *almost* cover the entire prior …". Why the word almost here, what is meant with it? Why is it significant to mention 'almost'?
27. Results, p12, lines 277-278: "… Most of the high-sensitive parameters show more peaked densities in a narrower range of parameter values, reflecting the significance of variations in model parameter values …". Please, explain exactly why the statement is true.
28. Results, p13, lines 291-292: "…, indicating that the model has the potential to generate accurate cosmic-ray soil moisture estimates even under dry conditions." Please, explain why 'even under dry conditions'? Is high performance under these conditions a surprise? If so, why?
29. Results, p13, line 293: "…, a loss of the physical meaning of the parameters in question would be very critical". Why would this be critical?
30. Results, p13, line 295: "One of the important addition**s** of this work …". Was incorporating lattice water count added by this study for the first time?
31. Results, page 14, figure 4: Please, try to make this figure easily readable in greyscale, this would help readers who print to read the paper carefully.
32. Results, page 14, line 308: "Furthermore, the behavrioral simulation ensembles captured more variations in de COSMIC method compared to the Desilets method after the application of the objective function (i.e. KGEalpha,beta)". Do you know why? Please discuss here if Results and Discussion are combined.
33. Results, page 16, lines 318-322. How are you sure the that surface ponding and shallow groundwater and other mentioned factors are a major cause of uncertainty? Was an uncertainty analysis performed? Please, if so, discuss these briefly. If not, on which observations is this discussion based?
34. Results, page 16, lines 325-330: Please, provide references to support the discussion in this paragraph.
35. Results, page 17, figure 5: The figure could be interpreted more easily and quicker if the choice of colours stated in the caption (red and black) for the different Desilets daily neutron counts, are put in the figure legend.
36. Results, page 18, line 340: An LHS sample 100 000 seems a lot for just the N-parameters from the three neutron models. Why was this sample size chosen?
37. Results, page 19, table 4: A figure could help the reader to get a clear overview of these results quickly. Please, consider a parallel coordinates plot or something alike.
38. Results, page 20, figure 7: Please, add horizontal axis title.
39. Results, page 20, lines 351-353: Were eddy-covariance measurements available at the other sites? If so, the same analysis should be done and presented for those sites, for complete insights from this research.
40. Results, page 21, lines 354-355: 'Panel © displays the scatter plot that reveals no systematic over or underestimation of the observed actual evapotranspiratiom": The dashed line in the figure does not show the 1:1-line. How then does the scatter plot reveal no over or underestimation?
41. Results, page 21, lines 357-360. Given pieces of discussion occur in the current Results chapter, please discuss the differences in correlations between observed and simulated evapotranspiration between different seasons.

42. Results, page 21, lines 370-376; This paragraph seems to relate to the paragraph and results I mentioned in my previous comment. If this is correct, please restructure the text so this becomes clearer.

43. Results, page 22, figure 8: If the two RMSE boxplots are combined into a single one with a single vertical axis domain, could this help the comparison?

44. Discussion, page 23, lines 389-390: "Therefore, we extended this uniform-averaging scheme by a vertical weighting scheme to mimic the sensitivity of the neutrons to the upper layer" Was this a contribution done through the work in this research or should previous work be referenced here?

45. Discussion, page 23, lines 407-408: "…, indicating that the dynamic vegetation effect is just a minor observational issue (…)". The abundant vegetation does affect the CRNS measurement precision. How does that affect the calibration process and further analysis of this study?

46. Discussion, page 24, lines 420-421: "… the results confirm the findings from Zink et al. (2017)." Please expand a small bit on this reference. Which type of soil moisture data did they use?

47. Discussion, page 24, line 435-436: "… while the weighted approach N(Des,U) shows a slightly better performance that the other two methods …" How significant was the difference, i.e. What is meant with 'slightly'?

48. Discussion, page 25, line 451: "We also included offset hydrogen pools in the form of lattice water to the N0 calibration function, …" Was soil organic matter included? If now, why not? Another factor, vegetation (including intercepted water), was this corrected for in this study? If not why not? If so, what did the results indicate? How substantial was the effect of vegetation at the different sites?

49. Conclusion and future outlook, page 27, lines 500-503: Different sources of uncertainty regarding the neutron modelling are mentioned here. I wondered, given modelling tools are available to give an estimation of the size of the contributions from the different factors on neutron intensity, were such estimations made within this study? If so, what did they tell?

50. Conclusion and future outlook, page 27, lines 509-511: "… provides a more realistic representation of soil moisture dynamics as well as evapotranspiration, particularly at the forest site". If I have understood the manuscript correctly, evapotranspiration was evaluated at one site only. The sentence here in the conclusions chapter seems to suggest a broad result for evapotranspiration. Please, rewrite to make this explicit.

51. *Please check for textual imperfections throughout the manuscript. Three examples from the abstract, introduction, and results:*

    - P1, L3: "… due to their hectare scale footprint and …" -> "… due to its hectare scale footprint and … "
    - P2, L21: "the mass" -> water mass? Carbon mass? Both or more?
    - P13, L285: The words 'uniform prior distribution range for' should be repeated before "N0,cosmic", or rephrase in another way

---

## Referee Comment (RC2)

**Overview**

The paper describes the simulation neutron count rates using a gridded hydrological model, which estimates soil water content in different depth layers. The simulated counts are compared with CRNS neutron counts to optimise the model (mHM) calibration. This approach addresses the issue of the variable measurement depth of CRNS. In simulating neutron counts, published models are used such as COSMIC, or the CRNS neutron count to soil moisture calibration function is inverted. However, these relationships are further fitted through optimisation of the $N_0$ parameter, and different published schemes for vertical soil moisture content weighting are tested. Other mHM parameters are also optimised, but how this is done and in what order, is not explicitly described. Which parameters are optimised is only shown in the supplementary information, and the results and discussion do not cover these parameters.

The motivation of this work is valid, but not very clearly expressed in a long introduction which should be more focused. Only at line 81 is the issue of measurement depth introduced, whilst this is a key part of this work. Other specific issues in the presentation (see below), make the reader question the validity of the work, and I doubt that the study could be reproduced from the description. There is further concern whether the methods are suitable to meet the objectives of the paper, for example there simpler ways to evaluate the depth weighting, without complicating factors such as a gridded model representation of a site (which could be better modelled as a single point).

Some of the Discussion is also unfocussed, and reverts to discussing other work, without bringing much insight into this study. There is no discussion on the optimisation of hydraulic parameters or PTFs, nor how this might improve model performance at other locations e.g. how can the results be transferred to grids without CRNS observations?

The manuscript needs to be revised to be better focused, clearer to the reader, and provide sufficient detail in the Method to be reproducible. The results and discussion need to cover the other model parameters optimised, whereas currently only CRNS parameters and a rooting parameter are mentioned, as though they are the only influences on the results. Also, there is much claim of improvements in modelled soil moisture, when most of the plots only show neutron count comparisons. It is really unclear as to how much improvement comes from better model parameterisation of the soils, and how much is due to changing the relationship between neutron counts and soil moisture – I even wonder if the observations are being calibrated to fit the model? How do the newly fitted values of $N_0$ compare with those fitted by site soil sampling calibrations?

How did you optimise $N_0$ before you optimised other mHM parameters? A schematic of your methodology may help, especially to show the order of optimisation – before optimisation, neither the simulated counts are true (or best estimate), nor the modelled soil moisture (SM) layers. Does your method iterate so these parameter sets both improve together?

**Specific Points**

Line 52 by this point or earlier you should say why there could be an issue with depth averaging.

Line 70 delete 'eventually'

Line 75 still no intro as to the motivation to do this - rather than use the derived SM!

Line 86 ….still not clear what is the objective of this study?

Line 89 If the objective is a technical comparison of methods, then why do this at grid scale, not a point scale, actually at the CRNS station? This may have complicating factors e.g. mixed land cover, soils and topography modelled across a grid cell.

Line 99 there seems to be an implicit assumption that working with simulated counts is better than using CRNS derived SM - did you test this?

Lines 102-3 does this mean different land cover in grid or between grid cells?

Line 108 homogeneous at the CRNS hectare scale, but not at 1km scale! - see photo of Grosses Burch. Please be more specific – and how might in-grid heterogeneity affect optimisation?

Line 115 Are these seasonal biomass fluctuations included in the CRNS count simulation?

Line 159-60 'the sensitivity to the highest…' re-phrase, this is not clear.

Line 170 delete 'time constant'

Lines 183-87 it may make sense to use grid averages for mHM, but this complicates the evaluation of the methods – how representative is the CRNS station of the wider grid properties? If the station is not representative of the grid, then inappropriate parameter changes are forced to match soil moisture or CRNS counts to soil properties that do not match the CRNS site.

Line 187 How can varying $N_0$ properly account for biomass changes? Biomass is dynamic in time, whilst $N_0$ is constant in time?

Line 212 change 'neutrons' to neutron flux or count rate.

Eq. 9 is bulk density here the same as Eq. 1? (a different undefined symbol is used here and in Eq. 10)

Line 238 these symbols are not in Eq.14 ? - what are $L\_30$ & $L\_31$?

Line 257 Table 2 shows model performance measures, not model parameters and their ranges - please add table of model parameters optimised. (it is in supplementary)

Line 269 …presentation of results is confusing, as it does not show hydraulic parameters (only refers to Table S1). Plots and tables only show CRNS parameters.

Fig.4 – add units (…are these counts per hour?)

Line 301 This gives the impression that what has been done here is to optimise neutron count match, by varying some model parameters, especially $N\_0$, and some root depth parameters ….there could be an issue that whilst counts may agree well, systematic bias in SM could be compensated for by varying $N\_0$; i.e. better efficiency in simulating neutron counts may not necessarily lead to better efficiency in SM modelling.

Section 3.2, Fig. 5 & Fig.6 – should show these also as CRNS SM plots - as the non-linear counts to SM hides the magnitude of discrepancies.

Line 410 – this study does not assess the absolute soil water quantity (most results are presented as neutron counts).

Lines 412-13 'Incorporating CRNS data...' this statement is untrue and does not makes sense.

Line 430 this is misleading as it implies data assimilation, whereas neutron data is not incorporated, it is simulated by mHM and then compared with observed counts.

Line 450 'After optimizing the soil hydraulic properties... ' this is not properly described in the method – is this done after CRNS parameters are optimised or at the same time in some form of iteration? How did you do this?

Fig.9 – surely belongs in Results?

---

## Author Comment (AC1)

* * *
*Reviewer comments are presented in italics*, while the authors' responses are in blue.

**1 Referee Comments and Responses**

**Anonymous Referee 1:**

*The authors have investigated how cosmic-ray neutron soil moisture data can help improve the simulation of both soil moisture and evapotranspiration with a hydrological model. They used three relevant methods to incorporate neutron counts into large-scale hydrological modelling and compared their performance. Using cosmic-ray neutron soil moisture data to calibrate the hydrological model improved the simulation of both soil moisture and evapotranspiration.*

   *I have found the manuscript interesting and mostly well written. The significance of the work is clear to me. I do have a few suggestions to make the contribution of the presented research stronger. These include both suggestions to improve readability and some suggestions to solidify the outcomes with some more elaborate explanations of the methodologies and a few small additional analyses.*

   We appreciate the reviewer comments and detailed suggestions. We greatly value your efforts to improve the paper structure. We will make the recommended revisions to the introduction and methodology section. We are especially grateful for the referee positive feedback. In this document, we present our comprehensive responses and outline our strategy for addressing the reviewer's comments in a future revision of this manuscript.

**Major comments:**

1. *Please, consider shortening the paragraphs of lines 32-48, 49-60, and 61-72 of the Introduction on pages 2 and 3, to help the reader understand the story line better. It is now a broader literature review that might help the reader to get the key message. Some references that are highly relevant can (and in many cases do already) enter the story in the Results and Discussion.*

   Thank you for your suggestion to improve the Introduction. We will shorten paragraphs as recommended and consider moving relevant references to the Results and Discussion section for improved clarity.

2. *Introduction, page 3, L83: "... neutron counts at scales of 1.2 x 1.2 $km^2$" and Conclusion and future outlook, page 26, lines 483-484 "... for simulating neutron counts at the $0.01562^o$ x $0.01562^o$ grid ...". Please, clarify in which way the neutron count simulations evaluated at this scale. Scale mismatch between model grid cell size, different model inputs, and different model calibration/validation data assimilation data should be an important aspect of this study. Please, include a discussion on the impact of scale mismatches in the manuscript. Clarification can be done in the Introduction and/or Materials and Methods and Discussion.*

   We are grateful for the reviewer's feedback. We will explain the mHM model setups, including spatial resolution, in more detail.
   L1 and L2: $0.01562°$ x $0.01562°$ is eq.$\sim$ 1.2 x 1.2 $km^2$. Level 1 (L1) describes the spatial resolution, as which dominant hydrological processes are modelled and Level 2 (L2) describes the resolution of the meteorological forcing data.
   L0: $0.001953125°$ × $0.001953125°$. Level 0 (L0) describes the subgrid variability of relevant basin characteristics, which includes information on the soil as well as land use, topography and geology.

3. *If the model produces other output than soil moisture and evapotranspiration, meaning other water fluxes, can the authors discuss how the estimation of these fluxes changes under calibration with CRNS-data? If observations are available, please include these in comparison, or at least mention such analyses as recommendations for next research*

*steps. It is important to verify that other model outputs do not deteriorate, or better, actually improve simultaneously with evapotranspiration simulation.*

For this study, we had access to eddy-covariance measurement data only at the Hohes Holz site, which allowed us to perform the cross-validation of evapotranspiration simulations at this site. In the discussion section, we will add how the correlations between observed and simulated evapotranspiration vary in different seasons. The correlation coefficients (r) for each season are as follows: autumn [SON] (r = 0.79), spring [MAM] (r = 0.77), summer [JJA] (r = 0.42), and winter [DJF] (r = 0.87). It is worth noting that winter shows the highest correlation between observed and simulated ET, while summer exhibits the lowest correlation. The most significant deviation in terms of RMSE is evident during the summer, when evapotranspiration is highest, while the smallest difference is in winter when evapotranspiration has less impact. The model slightly overestimates evapotranspiration in summer and spring, possibly because of the absence of a dynamic vegetation growth module in the mHM, also discussed for evapotranspiration in Zink et al. (2017).

4. *A comparison with in-situ soil moisture observations is now briefly discussed in the Discussion, page 23, line 411. I suggest that the authors move this forward and make it more prominent by showing a comparison in a figure and expand the discussion. If in-situ soil moisture data were available at the other sites, these should be discussed too. If such data are not available, please mention this explicitly. Given the grid cell size of >1 km, satellite remote sensed soil moisture data is relevant too. Please discuss the relevance of CRNS data compared to satellite data at this modelling scale. To my opinion, this issue should be discussed. Implementing actual calibration and/or validation/ data assimilation with point scale soil moisture data and satellite remote sensing soil moisture data, I think should be a recommendation in the final chapter of this manuscript and should be considered by the researchers as interesting future work.*

Thank you for your feedback. In the COSMOS-Europe data paper (Bogena et al., 2022; Boeing et al., 2022), which is a key reference in soil moisture studies across Europe, soil moisture data from 66 CRNS stations deployed across Europe (referred to as COSMOS-Europe) is presented. This paper also includes the study sites that we focused on. In our paper, our primary focus was to establish a framework to invert soil hydraulic parameterization in mHM/COSMIC by directly comparing modelled neutron counts with measured ones. The on-site intensity of epithermal neutrons is directly linked to the soil moisture within the vertical and horizontal CRNS footprint.

Neutron count measurements capture soil moisture variability, as they are closely inter-linked Zreda et al. (2008); Desilets et al. (2010); Shuttleworth et al. (2013). Comparing modeled soil moisture (SM) with observations presents challenges due to scale mismatches, both in spatial extent and vertical depth. Neutron count measurements effectively overcome these challenges, making them a more suitable choice for comparison. The great advantage of CRNS over satellite data is that CRNS not only covers the few top centimeters of the soil as satellite measurements ('surface soil moisture'), but provides information on a vertical integral of soil moisture for about 15-50 cm. Furthermore, CRNS time series have a much higher temporal resolution than current satellite data. We will add this suggestion to the concluding section of the manuscript, emphasizing the significance of such future work for researchers in this field. Due to the reasons mentioned above including scale mismatch between model simulations and observations of soil moisture, we will put less emphasis on soil moisture simulation comparisons in the revised manuscript. This will also help concentrating the main focus of the study towards the neutron count simulations.

5. *Results section, page 12, lines 269-275, to the reader it is now not crystal clear which parameters were calibrated? Just the neutron related parameters or also other mHM parameters? Please clarify this textually and include a manuscript main text table (or other mechanism) to make this instantly clear.*

Thank you for your question. We will provide a comprehensive explanation of the parameter sets used in the methods section to ensure clarity and transparency. In our study, we used a total of 29 parameters for the Desilets method and 31 parameters for the COSMIC method, which include snow, soil moisture, and neutrons modules. For clarity, we have included box plots showing the calibrated range of all parameters in Figure S3 and Table S1 in the supplementary materials.

6. *Please, consider creating either one section Results and Discussion, or move bits of preliminary discussion (p 15, lines 312-315, 318-324, 325-330, 370-376) from the Results section to the Discussion section.*

Thank you for your feedback. We will reorganize the manuscript by moving the mentioned preliminary discussion segments (p 15, lines 312-315, 318-324, 325-330, 370-376) from the Results section to the Discussion section as requested.

**Other, specific comments:**

7. *Title: "Improved representation of soil moisture simulations..." I doubt if the word "representation" in relation to "soil moisture simulations" is well chosen. 'representation of soil moisture processes' or 'representation of soil moisture measurements' sounds logic, but here it seems as if the representation of soil moisture simulations is improved. Please think if this is really what you mean and if so, please consider if will be understood by the wider audience.*

Thank you for your feedback, regarding the title of our manuscript we would like to move to a revised title as: "Improved representation of soil moisture processes ...".

8. *Abstract: P1,L12-14: "A Monte Carlo simulation with Latin hypercube sampling approach . . . " Please, consider removing this sentence or writing it in more understandable wording for the audience. It is now hard to see the exact relevance of the technical details given, like 'N = 100 000'. What does such a number tell the audience?*

Thank you. We will revise the texts to make it more clear: "We use a Monte Carlo simulation method, specifically the Latin hypercube sampling approach with a large sample size (N = 100 000)". Furthermore, to avoid confusion with 'N' representing neutron counts, we will switch to the notation 'S' for the sample size to improve the clarity of the text.

9. *Abstract, P1, L15-17: "We find that the non-uniform weighting scheme in the Desilets method provides the most reliable performance, whereas the more commonly used approach with uniformly weighted average soil moisture overestimates the observed CRNS neutron counts". How did COSMIC perform compared to the two Desilets methods?*

In Table 4, we present a comprehensive evaluation of model performance based on three different approaches. Although we did not elaborate on COSMIC in the abstract, we will address the performance of COSMIC compared to the two Desilets methods in the abstract as suggested.

10. *Introduction, page 2, L49-41: Please, improve textually by building a logical bridge between the paragraph of lines 32-48 and of lines 49-69. As is, HYDRUS-1D is introduced suddenly and in a way that makes is seem as a very key model, without being clear why so.*

Thank you for the suggestion. We will revise the introduction to improve the flow of text between lines 32-48 and lines 49-69, as well explaining the relevance and significance of HYDRUS-1D in the context of our study.

11. *Introduction, page 3, L73-74: The word "Eventually" and the wording with which the mHM model is introduced, at the start of this sentence and paragraph, make it seems as if the mHM model is a key hydrological model, that is the logical end-point of a discovery process and that is the standard that every reader should instantly know. It might be a well-known model, but it is one of many. Please, to help the reader understand the position of the mHM model, rewrite this to a more neutral wording.*

Thank you. We will make the suggested change in the revised manuscript.

12. *Introduction, page 3, L83: "The COSMIC method is complex . . . ". What is meant by complex here? Please, clarify for the reader.*

We will revise the sentence in the introduction to avoid using the term 'complex' and provide a more descriptive explanation instead i.e., "The COSMIC method enables a comprehensive representation of the neutron generation process, which is computationally more demanding than using the analytical Desilets equation ".

13. *Materials and Methods, page 4, L105: "four sites". Why does this number differ from that on line 90 of the introduction (page 4), which says "three"?*

Here the three different sites mentioned are according to the respective landcover states: i.e, agriculture, deciduous forests, and grasslands. Four sites are mentioned in terms of CRNS locations from where we utilized the measured neutron counts data.

14. *Materials and Methods, page 4, line 111: "... producing methane fluxes". How is this relevant to the research presented? If it is relevant, it should be mentioned here and maybe discussed later on.*

The idea to mention the methane flux was related to provide few site-specific environmental conditions of the Grosses Bruch site. But we agree with the reviewer, that mentioning them at this point probably rather confuses and does not illuminate much to the study. We will take these parts out in the revision of the manuscript.

15. *Materials and Methods, page 5, Table 1: Please say in the caption that precipitation and temperature are yearly averages and in the table itself, say '[mm/year]' for Precipitation.*

We will make the suggested change in the revised manuscript.

16. *Materials and Methods, page 6-7: The first reference to figure 2 is now on line 151. I think that by referring the reader earlier (from line 138 onwards), it will be easier for them to understand the methodology, with this key figure in hand.*

Thank you for your suggestion. We will make the recommended change by referring to Figure 2 (Flowchart) earlier in section 2.3 on Model setup.

17. *Materials and Methods, page 7, figure 2: In this figure, a 'Neutrons' module now appears in the upper part (modules of mesoscale hydrologic model mHM) and below, where the different neutron models are mentioned. Is this how the modelling actually works? Is there one neutron module in the mHM and then, the outcomes (neutron counts) of these are fed to the neutron models? Please adjust the figure and/or make very clear in the manuscript text how the different bits are actually connected. In addition, please clarify if the short arrows between the left and right bits connect 'Spatial Data' to 'Model Setup' and 'Model Setup' to 'Performance Matrix' or if the connections are actually 'Spatial Data' to 'mesoscale hydrologic model' and 'CRNS-methods output' to 'Performance Matrix'*

In our study, we considered all mHM calibration parameters related to snow, soil moisture, and neutrons modules, leading to a total of 29 parameters employed for the Desilets method and 31 parameters for the COSMIC method. The simulation of soil water content considered these three mHM modules to estimate neutron counts. To comprehensively cover the parameter ranges, we sampled 100 000 (prior) parameter sets. Finally, we focused on the top 10 best performing (posterior) parameter sets based on the objective function, $KGE_{\alpha\beta}$, for further analysis and evaluation. We intend to enhance the clarity of the model setup section in the revised manuscript by providing additional descriptions, ensuring a clear understanding for readers.

18. *Materials and Methods, page 8, lines 158-159: "We compared these simulated values with the measured soil water content obtained through CRNS" This suggests soil moisture values were compared. Is this true or were actually neutron counts computed from mHM soil moisture simulation compared to neutron counts?*

In the revised manuscript, we will clearly state that we are comparing neutron counts, not soil moisture. Our main objective is to optimize the parameterization of soil hydraulic properties in mHM/Cosmic based on the comparison between measurement and modelled neutron counts. We will adjust the text in the revised manuscript to ensure that this objective is clearly reflected, thereby eliminating any potential confusion.

19. *Materials and Methods, page 8, 2.4.1: Please, restructure this paragraph, such that parameter names are mentioned after this equation, to improve the readability.*

Thank you for the feedback, we will change the Desilets method section as suggested in the revised manuscript.

20. *Materials and Methods, page 9, lines 185-186: "Organic water equivalent ...". Please rephrase this sentence.*

We will rephrase the sentence in the Materials and Methods section as: "regarding the variables of soil organic carbon (SOC) and biomass, it is important to note that these variables are often not readily available, especially when it comes to biomass data."

21. *Materials and Methods, page 10, line 204: "... does not get too small and SWC is not too high". Please, quantify.*

We will revise the text in the manuscript to specify that the lower limit for bulk density (BD) was defined as $1.0$ g/cm$^3$, addressing the reviewer's concern.

22. *Materials and Methods, page 11, line 234: COSMIC parameter alpha is mentioned here, but was also mentioned on line 221. This seems confusing. Please, check and improve/clarify.*

Thank you for pointing this out. It was a typo, and we will remove the mention of $\alpha_{COSMIC}$ from line 234.

23. *Materials and Methods, page 11, lines 237-238: The parameters within the formula on line 237 seem not to match the parameters on line 238.*

We agree, there was a missing equation after the one presented in line 237. We will make the necessary revisions in the manuscript by including the missing equation(s) after line 237. Thanks for spotting this.

24. *Materials and Methods, page 11, lines 248-249: "However, we modify KGE (Eq. 15) by removing the correlation coefficient rho, as it is just a measure of temporal signature and is largely dominated by seasonality alone". Why should seasonality not be included? Why is the correlation coefficient not relevant? Please, clarify this better for the reader.*

Thank you for your comment. Gupta et al. (2009) proposed the KGE as a weighted combination of the three components (bias, variability, and correlation terms), given that our simulation already exhibited satisfactory correlation due to strong seasonality, we opted not to consider it in our assessment (objective function), as it accounted for 33% of the total weighting in the overall KGE score.
Seasonality is an inherent characteristic in the northern hemisphere where precipitation minus evaporation is mostly driven by evapotranspiration. Even if a random parameter is selected correlation will always be higher because the meteorological forcing is the precipitation - evaporation is seasonal. The study by Cinkus et al. (2022) examined the limitations of commonly used hydrological performance criteria, particularly the Kling-Gupta Efficiency (KGE) and its variants, in model calibration and evaluation. In the revised version we will better explain why it is necessary to exclude the correlation from the KGE.

25. *Results, p12, line 273: "... in all 10 000 simulations": why was this number chosen? How do we know it is sufficient, insufficient, or too large? If only the N-parameters from the neutron models were calibrated, this seems like a large number.*

Thank you for your comment. The choice of 100 000 simulations was determined to ensure reasonably good coverage of the parameter sets within their prescribed range, given the relatively high number of parameters involved in our study. We sampled 29 parameters for the Desilets method and 31 parameters for the COSMIC method that are not only related to neutron count module ($N_0$) but also to other snow, soil, and vegetation processes that affects the soil water dynamics in mHM.

26. *Results, p12, line 270: "... parameter distributions that almost cover the entire prior ...". Why the word almost here, what is meant with it? Why is it significant to mention 'almost'?*

In response to this question, we will extend the relevant section in the manuscript with additional explanations. We used the wording 'almost' to recognize that we couldn't be completely sure we sampled every possible parameter set, we meant that this large sample size was chosen to comprehensively explore the parameter sets and capture a wide range of possible parameter combinations.

27. *Results, p12, lines 277-278: "... Most of the high-sensitive parameters show more peaked densities in a narrower range of parameter values, reflecting the significance of variations in model parameter values ...". Please, explain exactly why the statement is true.*

Thank you for mentioning this aspect. We will include the explanation in the manuscript as mentioned. Our study aimed to determine optimal $N_0$ values by refining the parameter range for $N_0$ using the three approach ($N_{\text{Des,U}}$, $N_{\text{Des,W}}$, and $N_{\text{COSMIC}}$), the parameter set range set for $N_0$ ranges between (600–1500) for Desilets method and (100–400) for COSMIC method. Through a calibration process, we adjusted these parameters to align more closely with observed data. From the iteration of 100 000 parameter sets, we selected the top 10 sets that yielded a narrower range of $N_0$ values, providing the best fit to the observed data. By 'more peaked densities,' we mean that following calibration from the posterior distribution, the figure (Fig. 4) displays the x-axis in gray, representing the original parameter range (600–1 500) prior distribution for Desilets method and (100–400) for COSMIC method. Meanwhile, the colored sections in orange, green, and purple indicate the parameter values of the top-performing sets for each study site.

28. *Results, p13, lines 291-292: "..., indicating that the model has the potential to generate accurate cosmic-ray soil moisture estimates even under dry conditions." Please, explain why 'even under dry conditions'? Is high performance under these conditions a surprise? If so, why?*

We mention that "even under dry conditions" emphasizes the model (mHM) performs well under dry conditions, we highlight the model ability to simulate a wide range of moisture conditions. In contrast, some hydrological models, such as HBV and PREVAH (PREecipitation Runof EVApotranspiration Hydrological response unit model; Viviroli et al., 2009), have shown weaker performance in simulating soil moisture, particularly during dry conditions, as demonstrated by Orth et al. (2015), with slightly better agreement with observations during wet conditions.

29. *Results, p13, line 293: "..., a loss of the physical meaning of the parameters in question would be very critical". Why would this be critical?*

Thank you for pointing that out, we agree that the sentence does not contribute to the clarity of the text and we will remove it.

30. *Results, p13, line 295: "One of the important additions of this work ...". Was incorporating lattice water count added by this study for the first time?*

Yes, the addition of lattice water to the neutron counts module of mHM is a novelty of our study.

31. *Results, page 14, figure 4: Please, try to make this figure easily readable in greyscale, this would help readers who print to read the paper carefully.*

Thank you for this suggestion. We will accordingly update the figure in the revised manuscript.

32. *Results, page 14, line 308: "Furthermore, the behavioral simulation ensembles captured more variations in the COSMIC method compared to the Desilets method after the application of the objective function (i.e. KGEalpha,beta)". Do you know why? Please discuss here if Results and Discussion are combined.*

Thank you for your question. The broader confidence interval, indicating a greater range of variations, implies a higher degree of uncertainty in the COSMIC method ($N_{\text{COSMIC}}$). The COSMIC approach explicitly accounts for water content snow, vegetation interception, and root-zone soil processes that may likely lead to a better representation of observed neutron count variation compared to Desilets that empirically represent such processes.

33. *Results, page 16, lines 318-322. How are you sure that surface ponding and shallow groundwater and other mentioned factors are a major cause of uncertainty? Was an uncertainty analysis performed? Please, if so, discuss these briefly. If not, on which observations is this discussion based?*

No, a formal uncertainty analysis was not performed. Our discussion regarding these factors, particularly for the Grosses Bruch site, is based on prior observations of field data explained in Schrön et al. (2017). Ponding in the wet season is a common phenomenon on this site and these effects are explicitly not considered in the mHM model; and therefore we identify and mention them for future model development.

34. *Results, page 16, lines 325-330: Please, provide references to support the discussion in this paragraph.*

    Thank you for pointing this out, we will add following references to support the statement (Massoud et al., 2019 and Zink et al., 2017).

35. *Results, page 17, figure 5: The figure could be interpreted more easily and quicker if the choice of colours stated in the caption (red and black) for the different Desilets daily neutron counts, are put in the figure legend.*

    Thank you for the hint and suggestion to improve the clarity of figure 5. We will make the necessary adjustments to the figure legend, with the name $N_{(Des,W)}$, and $N_{sim(Des,Uni)}$.

36. *Results, page 18, line 340: An LHS sample 100 000 seems a lot for just the N-parameters from the three neutron models. Why was this sample size chosen?*

    The choice of 100 000 simulations was determined to ensure thorough coverage of the parameter sets, given the relatively large number of parameters involved in our study. 29 parameters for the Desilets method and 31 parameters for the COSMIC method, we aimed to comprehensively explore the possible combinations of these parameter sets values. These 100 000 simulations enable us to fully capture the distribution of parameter values.

37. *Results, page 19, table 4: A figure could help the reader to get a clear overview of these results quickly. Please, consider a parallel coordinates plot or something alike.*

    Thank you for your suggestion, we have already included a boxplot in Figure 7 to illustrate model performance for KGE and objective function $KGE_{\alpha\beta}$. In Table 4, we have highlighted the best-performing values to aid readers in quickly identifying the best values for each method. Additionally, we have improved the caption of Table 4 for enhanced clarity.

38. *Results, page 20, figure 7: Please, add horizontal axis title.*

    Thank you for the suggestion, we will add a horizontal axis title to Figure 7 in the manuscript.

39. *Results, page 20, lines 351-353: Were eddy-covariance measurements available at the other sites? If so, the same analysis should be done and presented for those sites, for complete insights from this research.*

    No, for this study eddy-covariance measurements were only available at the Hohes Holz site. Therefore, we were able to evaluate the evapotranspiration simulations at this specific site only.

40. *Results, page 21, lines 354-355: 'Panel C displays the scatter plot that reveals no systematic over or underestimation of the observed actual evapotranspiration": The dashed line in the figure does not show the 1:1-line. How then does the scatter plot reveal no over or underestimation?*

    Thank you for pointing this out. The reviewer is right – the dashed lines in the figure do not represent the 1:1 line (identity line). Instead, they correspond to the best-fit regression lines corresponding to the data for the growing and non-growing seasons. These two regression lines provide insights into how well our models capture ET variations during these distinct seasons. We can also specifically estimate some summary statistics reflecting the over/underestimation of simulated values of ET. We will change the line in the revised manuscript.

41. *Results, page 21, lines 357-360. Given pieces of discussion that occur in the current Results chapter, please discuss the differences in correlations between observed and simulated evapotranspiration between different seasons.*

    In the discussion section, we will add how the correlations between observed and simulated evapotranspiration vary in different seasons. This plot provides insights into the seasonal variations in the relationship between observed and simulated ET. It suggests that the model performs best during winter, while its performance during summer is comparatively weaker. The correlation coefficients (r values) for each season are as follows: autumn [SON] (r = 0.79), spring [MAM] (r = 0.77), summer [JJA] (r = 0.42), and winter [DJF] (r = 0.87). It is worth noting that winter shows the highest correlation between observed and simulated ET, while summer exhibits the lowest correlation. The most significant deviation in terms of RMSE is evident during the summer, when evapotranspiration is highest, while the smallest difference is

in winter when evapotranspiration has less impact. The model slightly overestimates evapotranspiration in summer and spring, previously addressed in the response to question 41.

[Figure]

42. *Results, page 21, lines 370-376; This paragraph seems to relate to the paragraph and results I mentioned in my previous comment. If this is correct, please restructure the text so this becomes clearer.*

Thank you for your feedback. We will update the figure in the revised manuscript accordingly.

43. *Results, page 22, figure 8: If the two RMSE boxplots are combined into a single one with a single vertical axis domain, could this help the comparison?*

We believe that merging the two RMSE boxplots into a single plot with a single vertical axis domain is not suitable in this case. The reason is that the Y-axis values for ET during the growing and non-growing seasons significantly differ (due to differences in ET values between these two seasons). Combining them into one plot would result in the non-growing season boxplot being too small to visualize and making it difficult to distinguish the mean values within the boxplots. Therefore, we prefer to keep them separated for clarity.

44. *Discussion, page 23, lines 389-390: "Therefore, we extended this uniform-averaging scheme by a vertical weighting scheme to mimic the sensitivity of the neutrons to the upper layer" Was this a contribution done through the work in this research or should previous work be referenced here?*

In our study, we incorporated the vertical weighting scheme for soil moisture in Desilets method, into the mHM model, and we applied and tested it across various landcover sites. In past studies, the techniques of both weighted and non-weighted soil moisture approaches in the context of CRNS have been discussed (rivera et al. 2014, baroni et al. 2015, schreiner et al. 2016, zreda et al. 2016, Schrön et al. 2017, vather et al. 2019, barbosa et al. 2021)

45. *Discussion, page 23, lines 407-408: "…, indicating that the dynamic vegetation effect is just a minor observational issue (…)". The abundant vegetation does affect the CRNS measurement precision. How does that affect the calibration process and further analysis of this study?*

Thank you for your comment. We will remove these lines because they are related to CRNS data calibration at the field site, which is not directly relevant to our study on hydrological modeling.

46. *Discussion, page 24, lines 420-421: "... the results confirm the findings from Zink et al.(2017)." Please expand a small bit on this reference. Which type of soil moisture data did they use?*

Zink et al. (2017) utilized soil moisture observations, obtained from eddy covariance stations, to evaluate modeled soil moisture. These soil moisture measurements were collected using Time-Domain Reflectometer (TDR) or Frequency-Domain Reflectometer (FDR) sensors, which have a control volume of ten to hundreds of cubic centimeters only. Because of variations in spatial representativeness and sampling depth, they did not directly compare observed and simulated soil moisture. Instead, their objective was to analyze the temporal dynamics of soil moisture by normalizing the respective soil moisture time series (as described in Koster et al., 2009). We will expand the texts in the revise mansucrit to include these aspects.

47. *Discussion, page 24, line 435-436: "... while the weighted approach N(Des,U) shows a slightly better performance that the other two methods ..." How significant was the difference, i.e. What is meant with 'slightly'?*

The Grosses Bruch site with the uniformly weighted approach $N_{(Des,U)}$ shows a "slightly better" performance than the $N_{(Des,W)}$, means that in terms of correlation and another performance indices (i.e., KGE, NSE, PBIAS), as shown in Table 4. $N_{(Des,U)}$ (0.85, 0.69,0.7%) and $N_{(Des,W)}$ (0.81, 0.60, -1.3%).

[Figure]

48. *Discussion, page 25, line 451: "We also included offset hydrogen pools in the form of lattice water to the $N_0$ calibration function, ..." Was soil organic matter included? If not, why not? Another factor, vegetation (including intercepted water), was this corrected for in this study? If not why not? If so, what did the results indicate? How substantial was the effect of vegetation at the different sites?*

Unfortunately, soil organic matter was not explicitly parameterized in the version of mHM used for this study. The intercepted water on leaves and in the litter layer can be particularly challenging to quantify, especially in forested stations such as Hohes Holz (Bogena et al., 2013; Schrön et al., 2017). The assessment of mHM with evapotranspiration

data from eddy covariance stations at Hohes Holz site showed deficiencies in mHM. Especially in summer and spring, deviations of the modeled and observed ET indicate room for improving the representation of vegetation dynamics within mHM. However, for other sites (Grosses Bruch, Hordorf, and Cunnerdorf), we did not have eddy covariance stations to check the evapotranspiration of the measured vs. simulated ones. In lines 405-452, we discussed in detail the simulation of neutron counts and the factors influencing these simulations in comparison to observations at our study sites. We explored this using the three approaches: Desiles (Uniform, Non-Uniform), and COSMIC.

49. *Conclusion and future outlook, page 27, lines 500-503: Different sources of uncertainty regarding the neutron modelling are mentioned here. I wondered, given modelling tools are available to give an estimation of the size of the contributions from the different factors on neutron intensity, were such estimations made within this study? If so, what did they tell?*

Thank you for your question, our analysis primarily addresses the uncertainty of the model parameters, and we will clarify this in the revised manuscript. To assess parameter uncertainty in mHM with respect to neutron counts, we employed Latin hypercube sampling involving 100 000 parameter sets. we took the top 10 best parameter sets as a behavioral solution. In Supplement Figures S1-S5, we present the Probability Density Function (PDF) plots of all parameter sets for our study sites, both prior and posterior to the simulation. Our analysis result shows that $N_{0,Des}$, $N_{0,COSMIC}$, rootFractionCoefficient_pervious, and rootFractionCoefficient_forest e, are the most sensitive model parameters.

50. *Conclusion and future outlook, page 27, lines 509-511: "... provides a more realistic representation of soil moisture dynamics as well as evapotranspiration, particularly at the forest site". If I have understood the manuscript correctly, evapotranspiration was evaluated at one site only. The sentence here in the conclusions chapter seems to suggest a broad result for evapotranspiration. Please, rewrite to make this explicit.*

Yes, evapotranspiration was evaluated at one site because we have the eddy covariance flux observation only at the Hohes Holz. Thank you for your feedback, we will update the texts in the revised manuscript accordingly.

51. *Please check for textual imperfections throughout the manuscript. Three examples from the abstract, introduction, and results:*

   - *P1, L3: "... due to their hectare scale footprint and ..." -> "... due to its hectare scale footprint and ... "*
   - *P2, L21: "the mass" -> water mass? Carbon mass? Both or more?*
   - *P13, L285: The words 'uniform prior distribution range for' should be repeated before "$N_0$,cosmic", or rephrase in another way*

Thank you for your thorough review and your valuable feedback. We appreciate your attention to detail, and we will address these issues accordingly.

**References**

Boeing, F., Rakovec, O., Kumar, R., Samaniego, L., Schrön, M., Hildebrandt, A., Rebmann, C., Thober, S., Müller, S., Zacharias, S., Bogena, H., Schneider, K., Kiese, R., Attinger, S., and Marx, A.: High-resolution drought simulations and comparison to soil moisture observations in Germany, Hydrology and Earth System Sciences, 26, 5137–5161, https://doi.org/10.5194/hess-26-5137-2022, 2022.

Bogena, H. R., Schrön, M., Jakobi, J., Ney, P., Zacharias, S., Andreasen, M., Baatz, R., Boorman, D., Duygu, M. B., Eguibar-Galán, M. A., et al.: COSMOS-Europe: a European network of cosmic-ray neutron soil moisture sensors, Earth System Science Data, 14, 1125–1151, 2022.

Cinkus, G., Mazzilli, N., Jourde, H., Wunsch, A., Liesch, T., Ravbar, N., Chen, Z., and Goldscheider, N.: When best is the enemy of good–critical evaluation of performance criteria in hydrological models, Hydrology and Earth System Sciences Discussions, 2022, 1–25, 2022.

Desilets, D., Zreda, M., and Ferré, T. P.: Nature's neutron probe: Land surface hydrology at an elusive scale with cosmic rays, Water Resources Research, 46, 2010.

Shuttleworth, J., Rosolem, R., Zreda, M., and Franz, T.: The COsmic-ray Soil Moisture Interaction Code (COSMIC) for use in data assimilation, Hydrology and Earth System Sciences, 17, 3205–3217, 2013.

Zink, M., Kumar, R., Cuntz, M., and Samaniego, L.: A high-resolution dataset of water fluxes and states for Germany accounting for parametric uncertainty, Hydrology and Earth System Sciences, 21, 1769–1790, 2017.

Zreda, M., Desilets, D., Ferré, T., and Scott, R. L.: Measuring soil moisture content non-invasively at intermediate spatial scale using cosmic-ray neutrons, Geophysical research letters, 35, 2008.

---

## Author Comment (AC2)

*Reviewer comments are presented in italics*, while the authors' responses are in blue.

**1 Referee Comments and Responses**

**Anonymous Referee 2:**

*The paper describes the simulation neutron count rates using a gridded hydrological model, which estimates soil water content in different depth layers. The simulated counts are compared with CRNS neutron counts to optimise the model (mHM) calibration. This approach addresses the issue of the variable measurement depth of CRNS. In simulating neutron counts, published models are used such as COSMIC, or the CRNS neutron count to soil moisture calibration function is inverted. However, these relationships are further fitted through optimisation of the $N_0$ parameter, and different published schemes for vertical soil moisture content weighting are tested. Other mHM parameters are also optimised, but how this is done and in what order, is not explicitly described. Which parameters are optimised is only shown in the supplementary information, and the results and discussion do not cover these parameters.*
Firstly, we want to sincerely thank the reviewer for dedicating their time and expertise to review our work. Your insightful comments have significantly improved our manuscript. We will provide a comprehensive response to address your comments and suggestions.

*[Line 52:] by this point or earlier you should say why there could be an issue with depth averaging.*

Several studies investigated depth-weighting schemes and hydrogen pools' effects on measurement depth. Franz et al. (2012) showed that shallow wetting fronts impact measurement depth in sandy soils with non-uniform moisture profiles. Baroni and Oswald (2015) tested three different depth weighting techniques (vertically varying weights, uniform weights, and taking the effect of above-ground biomass into account), which yielded different measurement depths varying from 23 to 28 cm. Using vertically varying weights and taking into account other hydrogen pools gave the best measurement depth estimates.

*[Line 70:] delete 'eventually'*

This is also noted by reviewer 1 (comment number 11), we will make the suggested change in the manuscript.

*[Line 75:] still no intro as to the motivation to do this - rather than use the derived SM!*

We appreciate the reviewer feedback. The primary methodological focus is on inverting soil hydraulic parameterization in mHM/COSMIC by directly comparing modelled neutron counts with measured ones. To obtain soil moisture accurately from CRNS measurements, one needs prior knowledge of the soil moisture profile. However, CRNS typically lacks this information, which mHM can provide. We will improve the introduction to make sure readers understand the primary motivation of our study more clearly.

*[Line 86:] still not clear what is the objective of this study?*

The objective of this study is to establish a framework to incorporate CRNS data into the mesoscale Hydrological Model (mHM) by comparing empirical and physics-based approaches for neutron count estimation to improve soil water content parameters in mHM across different vegetation types in Germany. To do this we compared modelled with neutron counts to infer soil hydraulic parameters. We will clarify this main goal in the manuscript to help readers better understand.

*[Line 89:] If the objective is a technical comparison of methods, then why do this at grid scale, not a point scale, actually at the CRNS station? This may have complicating factors e.g. mixed land cover, soils, and topography modelled across a grid cell.*

Using soil moisture (SM) directly simplifies the process but may not account for CRNS depth variability under different soil wetness conditions. The COSMIC approach, being more physics-based, allows for depth integration and the non-uniform contribution of signals with depth. CRNS measurement is based on a spatial footprint of around 150 m together with a vertical penetration depth of typically up to $\sim$ 15-80 cm depth. Sub-grid heterogeneity in mHM is captured explicitly within its multiscale parameter regionalization (MPR) technique (Samaniego et al., 2010). We simulated the neutron counts in mHM at the location where the CRNS instrument is installed to measure neutron counts. We used the extracted input data in mHM specific to the CRNS location, including temperature and precipitation from the German Weather Service (DWD), along with other data sources discussed in the data availability section.

*[Line 99:] there seems to be an implicit assumption that working with simulated counts is better than using CRNS derived SM - did you test this?*

No, we have not performed the test. But the reason for this preference is that mHM provides soil moisture data at different layers, whereas CRNS derived SM values represent an integrated value and we do not know uniquely which simulated layered soil moisture corresponds to the CRNS derived SM value, which even depends on the vertical SM distribution. However, also COSMIC is based on some assumptions and internal parameters, and thus may be more straighforward, but not necessarily more accurate.

*[Line 102-3:] does this mean different land cover in grid or between grid cells?*

We address land cover variations within each individual grid cell. In our study, the mHM model utilizes a fixed grid cell size of $0.015625°$(approximately 1.2 km$^2$), with one grid cell assigned to represent each individual site.

*[Line 108:] homogeneous at the CRNS hectare scale, but not at 1 km scale! - see photo of Grosses Burch. Please be more specific – and how might in-grid heterogeneity affect optimisation?*

In the mHM model, the sub/in-grid heterogeneity is explicitly captured with its multiscale parameter regionalization (MPR) technique (Samaniego et al., 2010).

*[Line 115:] Are these seasonal biomass fluctuations included in the CRNS count simulation?*

No, this is not feasible as data of biomass changes were not available. However, Boeing et al. 2022 indicated that the saisonal changes of biomass are rather irrelevant, even in Hohes Holz, as soil moisture changes are the much larger source of variation represented by the CRNS measurements.

*[Line 159-160:] 'the sensitivity to the highest...' re-phrase, this is not clear.*

We will revise the sentence to make it more clear in the revised manuscript.

*[Line 170:] delete 'time constant'.*

We will make the suggested change in the manuscript.

*[Line 183-87:] it may make sense to use grid averages for mHM, but this complicates the evaluation of the methods – how representative is the CRNS station of the wider grid properties? If the station is not representative of the grid, then inappropriate parameter changes are forced to match soil moisture or CRNS counts to soil properties that do not match the CRNS site.*

Thank you for your comment, The spatial resolutions in mHM are defined at different levels:
L1 and L2: $0.01562°$ x $0.01562°$ is eq.$\sim$ 1.2 x 1.2 km$^2$. Level 1 (L1) describes the spatial resolution, as which dominant hydrological processes are modelled and Level 2 (L2) describes the resolution of the meteorological forcing data.
L0: $0.001953125° \times 0.001953125°$. Level 0 (L0) describes the sub-grid variability of relevant basin characteristics, which include information on soil, vegetation, topography and geology, among other basin relevant geophysical characteristics. These sub-grid heterogeneities are reflected to generate effective model parameters (e.g., soil porosity, field capacity, lattice water content) in mHM with the multiscale parameter regionalization (MPR) technique (Samaniego et al.,2010).

CRNS data was preferred over any other ground-based measurements i.e., point-scale (TDR, FDR) and satellite data, due to its ability to provide a horizontal footprint of around 150 m together with a vertical penetration depth of $\sim$ 15-80 cm depth. These feature make them relevant for the modelling scale, along with its higher temporal resolution that allow us to capture the fine details of soil moisture changes. We used site-specific input data and calibrated parameters to establish the mHM model at each site and by this we address the CRNS station's spatial comparability with the grid-level model simulations.

*[Line 187:] How can varying $N_0$ properly account for biomass changes? Biomass is dynamic in time, while $N_0$ is constant in time?*

A correction for biomass changes could by introduced in two ways, either with a separate correction factor on the measured neutrons or with a temporal dependency on the $N_0$ constant (Baatz et al., 2014). Mathematically, the two approaches are similar. However, due to the lack of biomass data, usually no correction is considered in this study. Vather et al. (2020) emphasizes the importance of considering changes in biomass, but especially in situations with sudden changes in biomass, such as harvest in agriculture or clearings in forestry. The study demonstrates how the difference in hydrogen content within the measurement area leads to different calibration values ($N_0$ values). While $N_0$ should ideally be constant across different study areas once all environmental hydrogen sources are considered, it can vary due to the presence of growing biomass, as observed in the study by Hawdon et al. (2014). In their research, they found that approximately 80% of $N_0$ variation could be attributed to the effects of biomass, but only after accounting for other hydrogen sources. So, while $N_0$ itself does not change with time, its variation across study areas can indirectly reflect changes in biomass levels and as such indicate missing biomass corrections. Those corrections are complicated since temporal biomass data is usually not available.

*[Line 212:]change 'neutrons' to neutron flux or count rate. Eq. 9 is bulk density here the same as Eq. 1? (a different undefined symbol is used here and in Eq. 10)*

Thank you for pointing out, we will make the suggested change in the revised manuscript: "neutrons" to neutron count rate. Regarding Eq. 9 and Eq. 10, the COSMIC method employs the bulk density and lattice water for each model soil depth. However, in Equation 1, the same bulk density is used for the Desilets method. To enhance clarity, we will make the change in the manuscript by replacing the $\varrho_{\text{bulk}}$ to $\varrho_{\text{b}}$ in the manuscript.

$$X_{\text{soil}}(z) = \Delta z \varrho_{\text{b}} \tag{9}$$

*[Line 238:] These symbols are not in Eq.14 ? what are $L_{30}$ and $L_{31}$?*

$L_3$ represents the fast neutron soil and water attenuation lengths, according to literature Shuttleworth et al. (2013); Rosolem et al. (2014), these values have been expressed as $L_3$ = -31.65 + 99.29$\varrho_{\text{b}}$, with the parameters correlated to soil bulk density. Where $\varrho_{\text{b}}$ (g cm$^3$) is the soil bulk density. $L_3$ are site-specific parameters, and we integrated them into the mHM model as part of the parameters to be optimized.

*[Line 257:] Table 2 shows model performance measures, not model parameters and their ranges - please add table of model parameters optimised. (it is in supplementary)*

We will make the suggested change in the revised manuscript.

*[Line 269:] presentation of results is confusing, as it does not show hydraulic parameters (only refers to Table S1). Plots and tables only show CRNS parameters. Fig. 4 – add units (. . . are these counts per hour?)*

In this study, we used a total of 29 parameters for the Desilets method and 31 parameters for the COSMIC method which includes hydrologic processes related to: snow, soil moisture, and neutron counts dynamics. For clarity, we have included box plots showing the other parameters in Figure S3 in the supplementary materials. Regarding Figure 4, the unit for $N_0$ is counts per hour (cph), which represents the site-specific, time-invariant calibration parameter. We will include the units in the figure in the revised manuscript.

[Figure]

*[Line 301:] This gives the impression that what has been done here is to optimise neutron count match, by varying some model parameters, especially $N_0$, and some root depth parameters ....there could be an issue that while counts may agree well, systematic bias in SM could be compensated for by varying $N_0$; i.e., better efficiency in simulating neutron counts may not necessarily lead to better efficiency in SM modelling. Section 3.2, Fig. 5 and Fig.6 should show these also as CRNS SM plots, as the non-linear counts to SM hides the magnitude of discrepancies.*

Our primary goal was to optimize the agreement between modeled neutron counts and observed neutron counts from the three method i.e., Desilets method ($N_{\mathrm{Des,U}}$, $N_{\mathrm{Des,W}}$) and COSMIC method ($N_{\mathrm{COSMIC}}$). A direct comparison between the model and observation is feasible only via neutron counts because CRNS soil moisture is an integrated value, while mHM represents soil moisture in different layers. By improving the modeling of neutron count measurements, we are inherently improving the representation of SM in mHM.

[Figure]

**Figure 1.** Observed soil water content from CRNS vs simulated data from mHM in 0–5, 5-25, and 25–60 cm depth .

*[Line 410:] This study does not assess the absolute soil water quantity (most results are presented as neutron counts).*

Thank you for your comment. We will revise these texts making it clear that our study does focus on capturing neutron counts and not absolute soil water content, in the revised manuscript.

*[Line 412-13:] 'Incorporating CRNS data…' this statement is untrue and does not makes sense. This is misleading as it implies data assimilation, whereas neutron data is not incorporated, it is simulated by mHM and then compared with observed counts.*

We are sorry for this confusion. The reviewer is correct; namely, we simulate neutron data within the mHM model and subsequently compare it with observed counts, rather than incorporating observed neutron data directly. We will revise the relevant sections to accurately reflect this statement.

*[Line 450:] 'After optimizing the soil hydraulic properties… 'This is not properly described in the method – is this done after CRNS parameters are optimised or at the same time in some form of iteration? How did you do this? Fig.9 – surely belongs in Results?*

We understand that this part was not very clear in the submitted version of the manuscript; and we will make every attempt to clarify it in the revised manuscript. In brief, we have optimized the soil moisture, snow, and neutron counts related parameters simultaneously, as illustrated in the flow diagram. We will provide a clearer description in the methodology section, outlining how this simultaneous parameter optimization is achieved through iterative methods.
Regarding Fig.9, we present the result of neutron counts corresponding to optimized parameter sets.

**References**

Baatz, R., Bogena, H., Franssen, H.-J. H., Huisman, J., Qu, W., Montzka, C., and Vereecken, H.: Calibration of a catchment scale cosmic-ray probe network: A comparison of three parameterization methods, Journal of Hydrology, 516, 231–244, 2014.

Baroni, G. and Oswald, S.: A scaling approach for the assessment of biomass changes and rainfall interception using cosmic-ray neutron sensing, Journal of Hydrology, 525, 264–276, 2015.

Franz, T. E., Zreda, M., Ferre, T., Rosolem, R., Zweck, C., Stillman, S., Zeng, X., and Shuttleworth, W.: Measurement depth of the cosmic ray soil moisture probe affected by hydrogen from various sources, Water Resources Research, 48, 2012.

Hawdon, A., McJannet, D., and Wallace, J.: Calibration and correction procedures for cosmic-ray neutron soil moisture probes located across Australia, Water Resources Research, 50, 5029–5043, 2014.

Rosolem, R., Hoar, T., Arellano, A., Anderson, J. L., Shuttleworth, W. J., Zeng, X., and Franz, T. E.: Translating aboveground cosmic-ray neutron intensity to high-frequency soil moisture profiles at sub-kilometer scale, Hydrology and Earth System Sciences, 18, 4363–4379, 2014.

Shuttleworth, J., Rosolem, R., Zreda, M., and Franz, T.: The COsmic-ray Soil Moisture Interaction Code (COSMIC) for use in data assimilation, Hydrology and Earth System Sciences, 17, 3205–3217, 2013.

Vather, T., Everson, C. S., and Franz, T. E.: The applicability of the cosmic ray neutron sensor to simultaneously monitor soil water content and biomass in an Acacia mearnsii forest, Hydrology, 7, 48, 2020.

---

## Author Comment (AC3)

* * *
*Community comments are presented in italics*, while the authors' responses are in blue.

**1 Community Comments and Responses**

**CC – community by Markus Köhli**

*In their manuscript "Improved representation of soil moisture simulations through incorporation of cosmic-ray neutron count measurements in a large-scale hydrologic model"*

*the authors Fatima et al. describe the combination of several years of data from CRNS instruments at selected experimental sites and hydrological modeling. The reviewer appreciates the approach of the authors to move towards a more comprehensive picture of how CRNS can help to understand hydrological dynamics however, there are significant concerns about the methodology, data representation, completeness in the representation of CRNS, and lack of transparency in the choice of models and parameters in the paper. The reviewer suggest addressing these issues to improve the quality and validity of the research.*

**General:**

*[- ] Without any apparent reason or justification this paper selectively uses the soil depths 0-5 cm, 5-25 cm and 25-60 cm. Except for the top soil layer, none of these classes are representative for either hydrological processes or measurement depths of CRNS. What is the reason for the authors to use that scheme? This sampling is too coarse and not deep enough to be acceptable and needs to be refined. This sampling can and will lead to non-obvious systematics and therefore would create misleading results. Specifically, as the authors focus on statistical analysis methods this whole approach seems questionable.*

We agree that our choice of vertical resolution can appear arbitrary for readers that are not experienced with intricacy of large-scale hydrological modelling and their application in mesoscale settings. While we agree that the vertical resolution could also be critical with respect to capturing small-scale hydrological processes, it has been, however, shown in the past that the chosen depth resolution is reasonably accurate to describe the relevant hydrological processes in the root-zone relevant soil processes for mesoscale hydrological models Al-Shrafany et al. (2014); Crow et al. (2018); Vereecken et al. (2015); Mohajerani et al. (2021); Downer and Ogden (2004). Furthermore, the chosen depth resolution and stratification ensure that the most important supporting points for the description of depth-dependent neutron sensitivity are captured as justified by recent literature. Andreasen et al. (2020); Franz et al. (2020); Baatz et al. (2017); Iwema et al. (2017); Andreasen et al. (2016); Han et al. (2015).

Besides, we have also conducted a set of experiments with varying soil depths, ranging from 3 to 6 layers, specifically (0-5 cm, 5-15 cm, 15-25 cm, 25-30 cm, 30-60 cm, and 60-100 cm). In the model settings, our analysis pointed out that the simulated neutron count results were not significantly influenced by the number of soil layers (see Fig. 1 below for more details). Based on our prior experiences of modelling with mHM, we adopt the same soil layer (0–5 cm, 5–25 cm, and 25–60 cm) depths as utilized in the study by Boeing et al. (2022) - who have extensively evaluated mHM simulations against observed soil moisture across different locations in Germany.

The theoretical measurement depths of CRNS are in the range of 10 to 80 cm according to Zreda et al. (2008), Franz et al. (2012), and Köhli et al. (2015). For intermediate soil moisture conditions (0.1–0.4 $m^3/m^3$), the average CRNS measurement depth is between 15 and 30 cm, but the highest sensitivity is in the upper layers, 0–5 cm (Schrön et al., 2017). It is correct that the accuracy of numerical calculations (such as our COSMIC model set-up) would benefit from higher resolved soil profiles, however, our experiments demonstrated that varying soil depths from 3 to 6 layers did not have a substantial impact on the simulated neutron count results in our model setting.

Finally, we would like to emphasis past modelling studies that use CRNS measurements in their respective model based investigation where certain modelling choices are inevitable. For example, Iwema et al. (2017) used a land surface model to investigate the impact of reducing the scale mismatch between surface energy flux and soil moisture observations of CRNS point measurement data. The model used with grid cells of 1 km$^2$, parameters calibrated with eddy-covariance flux data and point-scale soil moisture data. In the study they used point-scale soil moisture data from the soil layers up to 30 cm depth and 2012- 2015 data were simulated. Other factors limit the limited effect of calibrating soil parameters on soil moisture dynamics and surface energy fluxes spatial-temporal stability. Barbosa et al. (2021) study used HYDRUS-1D model for the simulation of soil moisture time series at the 6 different depths of 2, 20, 30, 40, 60, and 100 cm and compared it to independent soil moisture measurements of PR2. HYDRUS 1D is coupled with the COSMIC Operator and Richards nonlinear partial differential equation was used and compared the neutron count rate with soil moisture simulated data and inversely calibrated the soil hydraulic parameters based on CRNS data. Zhao et al. (2021) used the land surface model (the Community Land Model; CLM version 3.5), a coupled land surface-subsurface model (CLM-ParFlow) is applied, and used the soil layer depth of 5cm and 20 cm.

[Figure]

**Figure 1.** Simulated neutron counts from mHM using different soil horizon depths: three soil layers (5 cm, 25 cm, 60 cm), five soil layers (5 cm, 15 cm, 25 cm, 40 cm, 60 cm), and six soil layers (5 cm, 15 cm, 25 cm, 40 cm, 60 cm, 100 cm).

*[- ] Using just the $N_0$ method and the COSMIC operator in order to draw the general conclusions the authors would like to present is with respect to the state of the art not justified. Either narrow down the scope to a more exemplary analysis or include other models like the UCF method from Franz et al. or the UTS method from Köhli et al., which are both not even mentioned in the overview, yet mentioned in the discussion.*

Thank you for hinting at the other N-SM conversion functions, we will make it clearer that this is an exemplary study and other methods could be tested in future research. Our choice for COSMIC and Desilets was based on their widespread use in past/recent studies: e.g., Desilets method Desilets et al. (2010) by Baroni and Oswald (2015); Schreiner-McGraw et al. (2016); Zreda (2016); Avery et al. (2016); Vather et al. (2019, 2020); González-Sanchis et al. (2020); Power et al. (2021); Barbosa et al. (2021); Chen et al. (2022) and COSMIC Method Shuttleworth et al. (2013); Rosolem et al. (2014) used by Baatz et al. (2014); Iwema et al. (2015); Han et al. (2015, 2016); Brunetti et al. (2019); Mwangi et al. (2020); Patil et al. (2021). The UCF method from Franz et al. (2013) has its own limitation as has been shown low experimental performance in the past (McJannet et al. 2014, Baatz et al. 2014). The recent UTS method from Köhli et al. (2021) was published after the start of our study, it still requires sound experimental validation, and it adds further complexity (such as air humidity dependency) which is not feasible for mHM at this state. In fact, the cited paper shows that the UTS function behaves very similar to the COSMIC approach. Hence, we believe that we have covered the main processes of the N-SM conversion approaches. Nevertheless, we will mention the recent developments of N-SM conversion functions in the revised manuscript as an opportunity for future elaborated studies.

*[- ] The study heavily focuses on modeling and statistical analysis. Throughout the manuscript, the reader encounters a large amount of seemingly arbitrary choices, which in the end provides the impression that results are selected and tinkered in order provide a realistic picture. The authors want to show that they uses methods which are accepted in the community, from the statistical and the modeling perspective, then, however, they introduce a significant amount of modifications and ad hoc assumptions which may put the whole approach into question. Examples are: In which way should a sophisticated hydrological model help, if the authors choose to simply take only three layers, for which the choice of hydrological parameters is also not really transparent? What is the reason to take only the 1% of the model runs with the best KGE (that number clearly depends on the initial parameter ranges the authors arbitrarily chose)? As the choice of the neutron model and their parametrization is not at all according to any standard, what does such tight constraint say other than the whole analysis is tuned to fit one specific ad hoc assumption. In which way is the KGE modified by the authors introducing a bias on this analysis? Why are 99 % of the model runs excluded if there are significant deviations from the models and the data, even visible in the plots presented? To be clear on that point: If significant deviations can be observed between model and data, any tight statistical constraint (in matching them incongruently) will lead inevitably to wrong results.*

As stated above, we have analysed different soil depths ranging from 3 to 6 layers (0-5 cm, 5-15 cm, 15-25 cm, 25-30 cm, 30-60 cm, and 60-100 cm) but found that the number of layers did not significantly affect neutron count results. As a result, we adopted the same layer depths used in Boeing et al. (2022).

We selected 1% of the model run with the best $KGE_{\alpha,\beta}$ because this simulation yielded neutron counts that captured well towards observed values. Increasing the number of simulations is certainly possible but this could introduce another level of complexity.

Regarding the revision of KGE, Gupta et al.2009 proposed the KGE as a weighted combination of the three components (bias, variability, and correlation terms). We opted not to consider it in our assessment (objective function), as it accounted for 33% of the total weighting in the overall KGE score. High correlation values stem from the fact that the seasonality of SM is an inherent characteristic in the northern hemisphere, where precipitation minus evaporation is mostly driven by evapotranspiration. Evapotranspiration is higher in summer and lower in winter, and soil moisture indirectly will result from low ET values in winter and high values in summer. This aspect is inherent, and we do not want to emphasis in our optimization these characteristics. That does not mean we are neglecting seasonality, but do not want our parameters to capture the seasonality aspects. Even if a random parameter is selected, the correlation will always be higher because the meteorological forcing is the precipitation - evaporation is seasonal, and seasonal is coming from forcing. Previous studies have also introduced revisions to KGE Mizukami et al. (2019).

The COSMIC model is the established standard for Cosmic-Ray Neutron Sensor (CRNS) forward modeling. There are

several studies in the literature that have utilized COSMIC as the recognized model for CRNS simulations. These studies include (Baatz et al. 2014), (Iwema et al.2015), (Han et al. 2015), (Han et al. 2016), (Brunetti et al. 2019), (Mwangi et al. 2020), and (Patil et al. 2021).

*[- ] Instead of overloading the manuscript with a multitude of different statistical measures, the authors should focus on providing a reasonable basis for comparing model and data. The authors rather present in the manuscript their own struggles and the reader does not learn anything from that way of analyzing a problem the authors fabricated in an intransparent way.*

We are using the five most established measures in hydrology to evaluate the model performance on observations (KGE, modified KGE, NSE, $R^2$, and PBIAS). As it is well known from the literature Nash and Sutcliffe (1970); Gupta et al. (2009), each of these measures has its own justification to assess the quality of the performance in terms of bias, dynamics, temporal and static errors, etc. The fact that our results are not only based on a single measure but rather in agreement across the many different measures shows that our conclusions are particularly robust (see Table 4). We believe that this procedure strengthens our study and demonstrates that neutron counts could improve hydrological model results not only in terms of correlation but also in reducing biases and improving overall performance measures.

**General figure layout:**

*[- ] the neutron data is plotted as quite large dots, which scatter significantly. Either smaller dots should be chosen or some type of smoothing.*

Thank you for your suggestion. We will consider using smaller dots in the neutron data plots to improve their clarity and readability.

[Figure]

*[l88:] "the physics-based model COSMIC" - COSMIC is not physics-based, it is not based on a comprehensive physical interaction picture. It selectively takes specific processes and invents arbitrary mathematical representations for them.*

As was demonstrated in a large number of accepted literature, COSMIC is an analytical-based model incorporating key physics-based processes important for CRNS applications in conjunction with models. COSMIC simplifies the physical process tailored to CRNS applications by mimicking a similar neutron transport behavior at the vertical axis as compared to more complex Monte Carlo models that actually implemented detailed physical interaction processes in three dimensions Shuttleworth et al. (2013); Rosolem et al. (2014). The model includes descriptions of various physical processes, such as neutron flux degradation with soil depth, creation of fast neutrons within the soil, and scattering of fast neutrons, all depending on soil composition and water content. It has been validated with detailed physics-based simulations and has been empirically confirmed in many subsequent studies i.e., Baatz et al. (2014), Iwema et al. (2015), Han et al. (2015), Han et al. (2016), Brunetti et al. (2019), Mwangi et al. (2020), Patil et al. (2021)). Naturally, not all detailed physical processes can be represented by a mathematical model, particularly the 3D neutron transport at scales of 1 to 300 meters. However, at the scale of interest in hydrological and land surface modeling (1 km), COSMIC can be considered adequate. In this study, we rely on the published model structure, while any model improvement with regard to additional physical processes will have to be discussed in a dedicated separate paper.

*[l99+:] "What is the best approach to simulate CRNS neutron counts in a hydrological model considering the heterogeneity of vertical soil moisture profiles?" - This paper provides an exemplary data analysis which is insufficient for generalizations of the mentioned type.*

Thank you for your comment. In our study, we checked the capability of a mesoscale hydrological model that can simulate the neutron counts by taking into account the vertical heterogeneity of different soil layers and comparing them with the measured neutron counts based on hector scale footprint. To account for sub-grid heterogeneity, which is captured with the multiscale parameter regionalization (MPR) technique in mHM (Samaniego et al., 2010), it allows us to study how model parameters vary across different scales of modeling.

*[l159+:] "Simulations from mHM revealed that the sensitivity to the highest soil water content was observed at 5 cm depth (...)" - this sentence is highly confusing, grammatically and in the context of the manuscript as for example there is no information provided anywhere about a layer specifically at 5 cm depth.*

Thank you. We will revise the sentence.

*[l165+:] Theoretically, the $N_0$ parameter, which represents the neutron count rate level of the particular CRNS probe used for rather dry soil at the local conditions, should be site-specific" - please describe theories which underline the theoretical reasoning that $N_0$ is site specific. The mentioned references do not provide that information. In case the $N_0$ equation is an inadequate representation of the neutron count rate a site-specific behavior would be a result.*

Thank you for this remark. The site-specific nature of the $N_0$ parameter is a well-recognized aspect within the Cosmic-Ray Neutron Sensor (CRNS) community, though there may be other future approaches for $N_0$ accounting directly for various site-specific influences. Currently, $N_0$ is typically calibrated for each sensor at each location via in-situ soil sampling campaigns or local soil moisture networks. Many authors have confirmed this observation (Zreda et al. 2012, Hawdon et al. 2014), while others investigated data from several different sensors and found non-identical $N_0$ values even if the detectors are similar (Fersch et al. 2019, Heistermann et al. 2021, Bogena et al. 2022). This indicates that the influencing factors on $N_0$ are not yet fully understood though a recent study has outlined that a minimal info from the site could be sufficient to determine the $N_0$ even without local soil moisture data (Heistermann et al., HESSD, 2023, https://doi.org/10.5194/hess-2023-169). Since the mHM model does not include these effects, we infer the value of $N_0$ through the calibration procedure.

*[l172:] "may be impacted by factors such as soil chemistry" - within the field of CRNS researchers claim that this method would be independent of soil chemistry. The reviewer is curious how the authors come to this assumption.*

Regarding CRNS data, the absolute neutron intensity varies by location due to varying soil chemical compositions, although these variations are generally small for epithermal and fast neutrons (ranging from 1 eV to $10^6$ eV) because

neutron absorption is insignificant. Thermal neutrons have a different sensitivity to varying soil moisture content than do fast and epithermal neutrons because thermal neutrons are highly dependent on on chemical composition of the soil matrix and soil water (Zreda et al. 2008, Andreasen et al. 2019, Rasche et al. 2021). This effect often is negligible, but we still mention it here since the variation of soil chemistry within our four sites is significant, particularly due to a high range of organic input below, on, and above the surface. We will better express this aspect by changing the formulation to "additional hydrogen pools (e.g., from organic material)".

*[l172:] "heterogeneity" - which type of 'heterogeneity' do the authors refer to? In case 'heterogeneity' refers to topographical heterogeneity the whole approach of this analysis is questionable.*

Thank you for pointing out this part. We would like to clarify the type of 'heterogeneity' we refer here is related to not only terrain (topography) but also local soil and vegetation characteristics. We will revise the sentence to better capture this aspect e.g, through "... neutrons are sensitive to all kinds of hydrogen in the footprint, hence the variable $\theta$ denotes not only soil moisture, $\theta_{sm}$, but is rather assumed to also include lattice water, $\theta_{lw}$, as well as water equivalent from soil organic carbon, $\theta_{org}$, and vegetation biomass, $\theta_{bio}$."

*[l182:] "derived from neutron particle physics modeling" - in which way are these parameters derived from particle physics and empirical (as mentioned above) at the same time?*

Thank you for pointing this out. In the revised manuscript, we will formulate the sentence as follows: "parameters $a_i$ were determined empirically by Desilets et al. (2010) who derived $a_0 = 0.0808$, $a_1 = 0.372$, and $a_2 = 0.115$ for values of $\theta > 0.02 \, gg^{-1}$."

*[l200:] The weighting scheme as presented is incomplete. (5)-(7) only take into account the weight of one depth. As to the model the authors used, the weight needs to be calculated by an integral over the depth weighting function for the height of the soil profile, not just the weighting function by itself.*

Thank you for your valuable feedback. We implemented the weighting procedure properly but have not adequately explained the procedure in the manuscript. This will be fixed in the revision. In the weighted-averaging approach, the weights are determined following Schrön et al. (2017), where the vertical contribution of layer $i$ is:

$$N_{\text{Des},W} = N_{\text{Des}}(\theta_{\text{avg}}(w_i)) \tag{1}$$

$$\text{where} \quad w_i = \int_{z_{i,\text{min}}}^{z_{i,\text{max}}} e^{-2z/D} \, \mathrm{d}z \tag{2}$$

$$\propto e^{-2z_{i,\text{min}}/D} - e^{-2z_{i,\text{max}}/D}$$

Here, the integral goes through each horizon from $z_{i,\text{min}}$ to $z_{i,\text{max}}$ in 1 mm steps and summes up the weight over the whole layer. The fact that the integral over an exponential function is again an exponential function is the reason for our simplified description using only $z_i$ in the original manuscript.

*[l203:] What is the depth $z_i$? As the authors use soil horizons of considerable height, how do the authors calculate $z_i$?*

Thank you for your remarks. To calculate the influence of soil moisture in different depths, the soil horizons were subdevided into smaller fractions and integrated over smaller steps of $z_i$. We will better clarify this in the manuscript in the revised manuscript.

*[l210+:] The COSMIC model only mimicks the mentioned subset of physical processes, in no way it represents them. Analytically COSMIC only represents an exponential N(theta) function, with an arbitrary parameter adaptation (8). The underlying mentioned physical processes are not responsible for the signal generation within CRNS as it lacks the spatial neutron transport, which CRNS claims to use as a unique feature compared to other methods.*

Thank you for your comment. According toa decent number of published and widely accepted literature, COSMIC is able to adequately mimic the neutron generation from soil moisture profiles Shuttleworth et al. (2013); Rosolem et al.

(2014) Baatz et al. (2014), Iwema et al. (2015), Han et al. (2015), Han et al. (2016), Brunetti et al. (2019), Mwangi et al. (2020), Patil et al. (2021) ). Of course, a mathematical model can never represent the full range of physical processes involved, however, as the authors from Shuttleworth et al. argue, it aims at representing the main processes in the vertical dimension (such as signal attenuation and neutron production/evaporation based on soil composition). This dimension is most relevant for large-scale models (500–4000 m). Spatial neutron transport at typical scales of 1–300 m might be necessary for detailed and complex terrain, as was shown by, e.g. Schattan et al., (2019), Schrön et al. (2023), and Köhli et al. (2023). But on average the typical mHM pixel size is at least one order of magnitude larger than the scale of lateral neutron transport, while the chosen sites exhibit homogeneous land use. Hence, we consider the 1D model assumptions adequate for the target application.

*[(11):] (11) is missing*

Thank you for pointing this out. This is a layout issue and will be resolved in the revision.

*[l241:] explain the term "geometric integral"*

We are here referring to equation 8 as mentioned in the text. As was explained, it resembles an integral of the vertical neutron transport, geometrically projected to the vertical axis. See also Schuttleworth et al. (2013).

*[l275:] What are "COSMOS models"?*

We will correct it to "COSMIC model".

*[l293:] "COSMIC is physically based, a loss of the physical meaning of the parameters in question would be very critical." - as described above COSMIC takes an incomplete subset of physical processes in order to justify its model. In this sentence the authors echo the critics which have been mentioned with respect to this model. Without the representation of neutron transport, for example, any possible source-only model can hardly justify itself to be correct.*

As stated above and as has been demonstrated in plenty of respective literature, the COSMIC model represents the vertical neutron transport in a 1D soil column in a physically consistent manner. Based on several confirming studies in the literature (see above), we consider it as a reasonable approach to be implemented within mesoscale hydrological models, in which horizontal resolution is orders of magnitude larger than the scale of lateral neutron transport. For detailed small-scale processes, snowpack quantification, and heterogeneous terrain, we would agree that COSMIC could be improved, but this is out of the scope of this study and probably even unnecessary given the large spatial scales on which typical hydrological and land-surface models operate. During the revision of the manuscript, we will reflect on these parts to make them clearer.

*[Fig. 4:] Why is the depiction of the histograms so coarse? Please enlarge the scale to the relevant range.*

The plots already show the full range of the 100 000 data for the prior range and 10 data points taken from the objective function which is the posterior range in the x-axis. The main message of the plots is to show how the posterior distribution was constrained compared to the respective prior distribution. That is why we plot both distributions along with their full ranges.

*[Fig. 4:] As many columns have the same height, the reviewer questions the representativeness of the results. The exact same height could mean that either the model provides on the basis of the 100 000 data sets the same values or most of the results were discarded and the authors want to draw conclusions from just a few values.*

The reviewer seems to misinterpret the Figure and we would be happy to sharpen the corresponding explanations in the revised manuscript. The sample size for all sites is fixed at N = 100 000 as the prior range, and the iterations are conducted within the $N_0$ range of 600-1500 along with another 28 parameters for Desilets method and 30 parameters for the COSMIC method for each site. Consequently, when utilizing the prior sample range, it appears identical for all sites due to this uniform setup.

*[l323:] Given the fact that the authors chose to represent the soil in very coarse layers, the "crowding cows" in the otherwise very wet catchment, seem to be a distracting and out of scope reasoning by the authors and at that point do not strengthen the scientific quality of the material presented.*

Evidence for the influence of crowding cows at this site has been mentioned in Schrön et al. (2017) and may introduce additional uncertainty to the data in specific periods (Aug to Sept). We agree that this is not a major issue particularly since there are no cows for the major part of the year. We further demonstrated that the choice of vertical layers is adequate, as explained above, and will add this to the revised manuscript.

*[Fig. 5 and Fig. 6.:] Both are plotted in such a tiny way, that it is hard to identify the different lines from each other and blur the relevant deviations.*

Thank you for the feedback, we will replot the data to improve visibility and make it easier to distinguish the lines and relevant deviations.

*[ l379+:] citep missing. "which are typically 5–60 cm and sensitive to shallow soil moisture" the reference tells that CRNS actually measures deeper than that and for that reason the choice of only simulating up to 60 cm seems wrong or incomplete.*

Thank you for your feedback. We will add the citation with the citep command. Our calculation was based on the formulas given in the citation and specific to the investigated sites. In the revised manuscript, we will mention that the theoretical measurement depth for the cosmic-ray probe varies, ranging from 12 cm in wet soils to 76 cm in dry soils (Zreda et al., 2008, 2012; Rosolem et al., 2013). For the sites in our study, we typically do not see measurement depths beyond 60 cm (Bogena et a. 2022).

*[l445:] "Overall, the three methods (NDes,U, NDes,W, and NCOSMIC) in mHM were able to consistently simulate the neutron count" - the results were inconsistent and the variability was only partially covered.*

Thank you for your comment. In the paragraph, the statement is that "Overall, the three methods ($N_{\mathrm{Des,U}}$, $N_{\mathrm{Des,W}}$, and $N_{\mathrm{COSMIC}}$) in mHM were able to consistently simulate the neutron count variability throughout the available data period." Here the consistency in simulating neutron count variability means that mHM has the capability of capturing the general trend and pattern of the simulated data, as illustrated in (Figs. 5 and 6). The light grey color shows the top 1% of the model run with the best $KGE_{\alpha\beta}$ values. Furthermore, we have provided a comprehensive discussion of the performance and variability of neutron count simulations with the three methods at each site, from lines 405-455.

**References**

Al-Shrafany, D., Rico-Ramirez, M. A., Han, D., and Bray, M.: Comparative assessment of soil moisture estimation from land surface model and satellite remote sensing based on catchment water balance, Meteorological Applications, 21, 521–534, 2014.

Andreasen, M., Jensen, K. H., Zreda, M., Desilets, D., Bogena, H., and Looms, M. C.: Modeling cosmic ray neutron field measurements, Water Resources Research, 52, 6451–6471, 2016.

Andreasen, M., Jensen, K. H., Bogena, H., Desilets, D., Zreda, M., and Looms, M. C.: Cosmic ray neutron soil moisture estimation using physically based site-specific conversion functions, Water Resources Research, 56, e2019WR026 588, 2020.

Avery, W. A., Finkenbiner, C., Franz, T. E., Wang, T., Nguy-Robertson, A. L., Suyker, A., Arkebauer, T., and Muñoz-Arriola, F.: Incorporation of globally available datasets into the roving cosmic-ray neutron probe method for estimating field-scale soil water content, Hydrology and Earth System Sciences, 20, 3859–3872, 2016.

Baatz, R., Bogena, H., Franssen, H.-J. H., Huisman, J., Qu, W., Montzka, C., and Vereecken, H.: Calibration of a catchment scale cosmic-ray probe network: A comparison of three parameterization methods, Journal of Hydrology, 516, 231–244, 2014.

Baatz, R., Hendricks Franssen, H.-J., Han, X., Hoar, T., Bogena, H. R., and Vereecken, H.: Evaluation of a cosmic-ray neutron sensor network for improved land surface model prediction, Hydrology and Earth System Sciences, 21, 2509–2530, 2017.

Barbosa, L. R., Coelho, V. H. R., Scheiffele, L. M., Baroni, G., Ramos Filho, G. M., Montenegro, S. M., Almeida, C. d. N., and Oswald, S. E.: Dynamic groundwater recharge simulations based on cosmic-ray neutron sensing in a tropical wet experimental basin, Vadose Zone Journal, 20, e20 145, 2021.

Baroni, G. and Oswald, S.: A scaling approach for the assessment of biomass changes and rainfall interception using cosmic-ray neutron sensing, Journal of Hydrology, 525, 264–276, 2015.

Boeing, F., Rakovec, O., Kumar, R., Samaniego, L., Schrön, M., Hildebrandt, A., Rebmann, C., Thober, S., Müller, S., Zacharias, S., Bogena, H., Schneider, K., Kiese, R., Attinger, S., and Marx, A.: High-resolution drought simulations and comparison to soil moisture observations in Germany, Hydrology and Earth System Sciences, 26, 5137–5161, https://doi.org/10.5194/hess-26-5137-2022, 2022.

Brunetti, G., Bogena, H., Baatz, R., Huisman, J. A., Dahlke, H., and Vereecken, H.: On the information content of cosmic-ray neutron data in the inverse estimation of soil hydraulic properties, Vadose Zone Journal, 18, 1–24, 2019.

Chen, X., Song, W., Shi, Y., Liu, W., Lu, Y., Pang, Z., and Chen, X.: Application of Cosmic-Ray Neutron Sensor Method to Calculate Field Water Use Efficiency, Water, 14, 1518, 2022.

Crow, W. T., Milak, S., Moghaddam, M., Tabatabaeenejad, A., Jaruwatanadilok, S., Yu, X., Shi, Y., Reichle, R. H., Hagimoto, Y., and Cuenca, R. H.: Spatial and temporal variability of root-zone soil moisture acquired from hydrologic modeling and AirMOSS P-band radar, IEEE journal of selected topics in applied earth observations and remote sensing, 11, 4578–4590, 2018.

Desilets, D., Zreda, M., and Ferré, T. P.: Nature's neutron probe: Land surface hydrology at an elusive scale with cosmic rays, Water Resources Research, 46, 2010.

Downer, C. W. and Ogden, F. L.: Appropriate vertical discretization of Richards' equation for two-dimensional watershed-scale modelling, Hydrological Processes, 18, 1–22, 2004.

Franz, T. E., Wahbi, A., Zhang, J., Vreugdenhil, M., Heng, L., Dercon, G., Strauss, P., Brocca, L., and Wagner, W.: Practical data products from cosmic-ray neutron sensing for hydrological applications, Frontiers in Water, 2, 9, 2020.

González-Sanchis, M., García-Soro, J. M., Molina, A. J., Lidón, A. L., Bautista, I., Rouzic, E., Bogena, H. R., Hendricks Franssen, H.-J., and del Campo, A. D.: Comparison of Soil Water Estimates From Cosmic-Ray Neutron and Capacity Sensors in a Semi-arid Pine Forest: Which Is Able to Better Assess the Role of Environmental Conditions and Thinning?, Frontiers in water, 2, 552 508, 2020.

Gupta, H. V., Kling, H., Yilmaz, K. K., and Martinez, G. F.: Decomposition of the mean squared error and NSE performance criteria: Implications for improving hydrological modelling, Journal of Hydrology, 377, 80–91, 2009.

Han, X., Franssen, H.-J., Rosolem, R., Jin, R., Li, X., and Vereecken, H.: Correction of systematic model forcing bias of CLM using assimilation of cosmic-ray Neutrons and land surface temperature: a study in the Heihe Catchment, China, Hydrology and earth system sciences, 19, 615–629, 2015.

Han, X., Franssen, H.-J. H., Bello, M. Á. J., Rosolem, R., Bogena, H., Alzamora, F. M., Chanzy, A., and Vereecken, H.: Simultaneous soil moisture and properties estimation for a drip irrigated field by assimilating cosmic-ray neutron intensity, Journal of Hydrology, 539, 611–624, 2016.

Iwema, J., Rosolem, R., Baatz, R., Wagener, T., and Bogena, H.: Investigating temporal field sampling strategies for site-specific calibration of three soil moisture–neutron intensity parameterisation methods, Hydrology and Earth System Sciences, 19, 3203–3216, 2015.

Iwema, J., Rosolem, R., Rahman, M., Blyth, E., and Wagener, T.: Land surface model performance using cosmic-ray and point-scale soil moisture measurements for calibration, Hydrology and Earth System Sciences, 21, 2843–2861, 2017.

Mizukami, N., Rakovec, O., Newman, A. J., Clark, M. P., Wood, A. W., Gupta, H. V., and Kumar, R.: On the choice of calibration metrics for "high-flow" estimation using hydrologic models, Hydrology and Earth System Sciences, 23, 2601–2614, 2019.

Mohajerani, H., Zema, D. A., Lucas-Borja, M. E., and Casper, M.: Understanding the water balance and its estimation methods, in: Precipitation, pp. 193–221, Elsevier, 2021.

Mwangi, S., Zeng, Y., Montzka, C., Yu, L., and Su, Z.: Assimilation of cosmic-ray neutron counts for the estimation of soil ice content on the eastern Tibetan Plateau, Journal of geophysical research: Atmospheres, 125, e2019JD031 529, 2020.

Nash, J. E. and Sutcliffe, J. V.: River flow forecasting through conceptual models part I—A discussion of principles, Journal of Hydrology, 10, 282–290, 1970.

Patil, A., Fersch, B., Hendricks Franssen, H.-J., and Kunstmann, H.: Assimilation of cosmogenic neutron counts for improved soil moisture prediction in a distributed land surface model, Frontiers in Water, p. 115, 2021.

Power, D., Rico-Ramirez, M. A., Desilets, S., Desilets, D., and Rosolem, R.: Cosmic-Ray neutron Sensor PYthon tool (crspy 1.2. 1): an open-source tool for the processing of cosmic-ray neutron and soil moisture data, Geoscientific Model Development, 14, 7287–7307, 2021.

Rosolem, R., Hoar, T., Arellano, A., Anderson, J. L., Shuttleworth, W. J., Zeng, X., and Franz, T. E.: Translating aboveground cosmic-ray neutron intensity to high-frequency soil moisture profiles at sub-kilometer scale, Hydrology and Earth System Sciences, 18, 4363–4379, 2014.

Schreiner-McGraw, A. P., Vivoni, E. R., Mascaro, G., and Franz, T. E.: Closing the water balance with cosmic-ray soil moisture measurements and assessing their relation to evapotranspiration in two semiarid watersheds, Hydrology and Earth System Sciences, 20, 329–345, 2016.

Schrön, M., Köhli, M., Scheiffele, L., Iwema, J., Bogena, H. R., Lv, L., Martini, E., Baroni, G., Rosolem, R., Weimar, J., et al.: Improving calibration and validation of cosmic-ray neutron sensors in the light of spatial sensitivity, Hydrology and Earth System Sciences, 21, 5009–5030, 2017.

Shuttleworth, J., Rosolem, R., Zreda, M., and Franz, T.: The COsmic-ray Soil Moisture Interaction Code (COSMIC) for use in data assimilation, Hydrology and Earth System Sciences, 17, 3205–3217, 2013.

Vather, T., Everson, C., and Franz, T. E.: Calibration and validation of the cosmic ray neutron rover for soil water mapping within two South African land classes, Hydrology, 6, 65, 2019.

Vather, T., Everson, C. S., and Franz, T. E.: The applicability of the cosmic ray neutron sensor to simultaneously monitor soil water content and biomass in an Acacia mearnsii forest, Hydrology, 7, 48, 2020.

Vereecken, H., Huisman, J.-A., Hendricks Franssen, H.-J., Brüggemann, N., Bogena, H. R., Kollet, S., Javaux, M., van der Kruk, J., and Vanderborght, J.: Soil hydrology: Recent methodological advances, challenges, and perspectives, Water resources research, 51, 2616–2633, 2015.

Zhao, H., Montzka, C., Baatz, R., Vereecken, H., and Franssen, H.-J. H.: The Importance of Subsurface Processes in Land Surface Modeling over a Temperate Region: An Analysis with SMAP, Cosmic Ray Neutron Sensing and Triple Collocation Analysis, Remote Sensing, 13, 3068, 2021.

Zreda, M.: Land-surface hydrology with cosmic-ray neutrons: Principles and applications, Journal of the Japanese Society of Soil Physics, 132, 25–30, 2016.

---

## Author Response (AR1)

**Author Response to Referee #1**

**Improved representation of soil moisture processes through incorporation of cosmic-ray neutron count measurements in a large-scale hydrologic model**

Fatima et al.
*Hydrol. Earth Syst. Sc.,* `doi:egusphere-2023-1548`
* * *
**RC:** *Referee Comment*,     AR: *Author Response*,     ☐ Manuscript text

**RC:** *The authors have investigated how cosmic-ray neutron soil moisture data can help improve the simulation of both soil moisture and evapotranspiration with a hydrological model. They used three relevant methods to incorporate neutron counts into large-scale hydrological modelling and compared their performance. Using cosmic-ray neutron soil moisture data to calibrate the hydrological model improved the simulation of both soil moisture and evapotranspiration.*

*I have found the manuscript interesting and mostly well written. The significance of the work is clear to me. I do have a few suggestions to make the contribution of the presented research stronger. These include both suggestions to improve readability and some suggestions to solidify the outcomes with some more elaborate explanations of the methodologies and a few small additional analyses.*

**AR:** *We appreciate the reviewer comments and detailed suggestions. We greatly value your efforts to improve the paper structure. We have made the recommended revisions to the introduction and methodology section. We are especially grateful for the referee positive feedback. In this document, we present our comprehensive responses and outline our strategy for addressing the reviewer's comments in a future revision of this manuscript.*

**Major comments:**

**RC:** *Please, consider shortening the paragraphs of lines 32-48, 49-60, and 61-72 of the Introduction on pages 2 and 3, to help the reader understand the story line better. It is now a broader literature review that might help the reader to get the key message. Some references that are highly relevant can (and in many cases do already) enter the story in the Results and Discussion.*

**AR:** *Thank you for your suggestion to improve the Introduction. We have shortened paragraphs as recommended and considered moving relevant references to the Results and Discussion section for improved clarity.*

**RC:** *Introduction, page 3, L83: "... neutron counts at scales of 1.2 km x 1.2 km" and Conclusion and future outlook, page 26, lines 483-484 "... for simulating neutron counts at the $0.01562^o$ x $0.01562^o$ grid ...". Please, clarify in which way the neutron count simulations evaluated at this scale. Scale mismatch between model grid cell size, different model inputs, and different model calibration/validation data assimilation data should be an important aspect of this study. Please, include a discussion on the impact of scale*

*mismatches in the manuscript. Clarification can be done in the Introduction and/or Materials and Methods and Discussion.*

AR: *We are grateful for the reviewer's feedback. We have explained the mHM model setups, including spatial resolution, in more detail. Level 1 (L1) describes the spatial resolution, as which dominant hydrological processes are modelled and Level 2 (L2) describes the resolution of the meteorological forcing data. Level 0 (L0) describes the subgrid variability of relevant basin characteristics, which includes information on the soil as well as land use, topography, and geology. In the method part added the explanation of the spatial resolution:*

> The model is executed over six years (2014-2020) with a daily time step, and the spatial resolution of the mHM grid cells is fixed at L1 and L2: 0.01562° x 0.01562° is eq.~ 1.2  km × 1.2 km using the WGS84 Coordinate Systems. Level 1 (L1) describes the spatial resolution, as dominant hydrological processes are model and Level 2 (L2) describes the resolution of the meteorological forcing data. L0: 0.001953125° × 0.001953125°. Level 0 (L0) describes the subgrid variability of relevant basin characteristics, which includes information on the soil as well as land use, topography, and geology.

RC: **If the model produces other output than soil moisture and evapotranspiration, meaning other water fluxes, can the authors discuss how the estimation of these fluxes changes under calibration with CRNS-data? If observations are available, please include these in comparison, or at least mention such analyses as recommendations for next research steps. It is important to verify that other model outputs do not deteriorate, or better, actually improve simultaneously with evapotranspiration simulation.**

AR: *For this study, we had access to eddy-covariance measurement data only at the Hohes Holz site, which allowed us to perform the cross-validation of evapotranspiration simulations at this site. In the discussion section, we have added how the correlations between observed and simulated evapotranspiration vary in different seasons. The correlation coefficients (r) for each season are as follows: autumn [SON] (r = 0.79), spring [MAM] (r = 0.77), summer [JJA] (r = 0.42), and winter [DJF] (r = 0.87). It is worth noting that winter shows the highest correlation between observed and simulated ET, while summer exhibits the lowest correlation. The most significant deviation in terms of RMSE is evident during the summer, when evapotranspiration is highest, while the smallest difference is in winter when evapotranspiration has less impact. The model slightly overestimates evapotranspiration in summer and spring, possibly because of the absence of a dynamic vegetation growth module in the mHM, also discussed for evapotranspiration in Zink et al. (2017).*

> To cross-evaluate our results,  we generated and filtered the 100 000 regionalized parameter sets based on observed neutron counts for behavioral solutions. After selecting the most effective solutions, we conduct cross-validation by comparing the mHM simulations of  evapotranspiration against observational data from eddy covariance measurements ICOS (Warm Winter, 2022; Pohl et al., 2023) at the *Hohes Holz*.

RC: **A comparison with in-situ soil moisture observations is now briefly discussed in the Discussion, page 23, line 411. I suggest that the authors move this forward and make it more prominent by showing a comparison in a figure and expand the discussion. If in-situ soil moisture data were available at the other sites, these should be discussed too. If such data are not available, please mention this explicitly. Given the grid cell size of >1 km, satellite remote sensed soil moisture data is relevant too. Please discuss the**

*relevance of CRNS data compared to satellite data at this modelling scale. To my opinion, this issue should be discussed. Implementing actual calibration and/or validation/ data assimilation with point scale soil moisture data and satellite remote sensing soil moisture data, I think should be a recommendation in the final chapter of this manuscript and should be considered by the researchers as interesting future work.*

AR: *Thank you for your feedback. In the COSMOS-Europe data paper Bogena et al. (2022); Boeing et al. (2022), which is a key reference in soil moisture studies across Europe, soil moisture data from 66 CRNS stations deployed across Europe (referred to as COSMOS-Europe) is presented. This paper also includes the study sites that we focused on. In our paper, our primary focus was to establish a framework to invert soil hydraulic parameterization in mHM/COSMIC by directly comparing modelled neutron counts with measured ones. The on-site intensity of epithermal neutrons is directly linked to the soil moisture within the vertical and horizontal CRNS footprint.*

*Neutron count measurements capture soil moisture variability, as they are closely inter-linked (Zreda et al., 2008; Desilets et al., 2010; Shuttleworth et al., 2013). Comparing modeled soil moisture (SM) with observations presents challenges due to scale mismatches, both in spatial extent and vertical depth. Compared to point measurements of soil moisture, CRNS measurements have a clear advantage here due to their significantly larger footprint, making them a more suitable choice for comparison. The great advantage of CRNS over satellite data is that CRNS not only covers the few top centimeters of the soil as satellite measurements ('surface soil moisture'), but provides information on a vertical integral of soil moisture for about 15-50 cm. Furthermore, CRNS time series have a much higher temporal resolution than current satellite data. We have added this suggestion to the concluding section of the manuscript, emphasizing the significance of such future work for researchers in this field. Due to the reasons mentioned above including scale mismatch between model simulations and observations of soil moisture, we have put less emphasis on soil moisture simulation comparisons in the revised manuscript. This will also help concentrating the main focus of the study towards the neutron count simulations.*

*In the conclusion section we added:*

> To optimize accuracy and understanding, we recommend integrating both CRNS and satellite remote sensing data into mHM. Improving the model predictions will contribute to reducing the uncertainties associated with drought and flood management strategies and informed agricultural decisions.

RC: *Results section, page 12, lines 269-275, to the reader it is now not crystal clear which parameters were calibrated? Just the neutron related parameters or also other mHM parameters? Please clarify this textually and include a manuscript main text table (or other mechanism) to make this instantly clear.*

AR: *Thank you for your question. We have provided a comprehensive explanation of the parameter sets used in the methods section to ensure clarity and transparency. In our study, we used a total of 29 parameters for the Desilets method and 31 parameters for the COSMIC method, which include snow, soil moisture, and neutrons modules. For clarity, we have included box plots showing the calibrated range of all parameters in Figures S6–S9 and Table S1 in the supplementary materials.*

The sensitivity and uncertainty analysis performed in this study use a Latin Hypercube Sampling (LHS) approach, resulting in parameter distributions that  large sample size was chosen to comprehensively explore the parameter sets and capture a wide range of possible parameter combinations in the prior range. The LHS approach creates a random value between the min and max values of the parameter set. Initial parameter ranges and exploratory model runs are set based on literature values (Boeing et al., 2022; Kumar et al., 2013). Supplementary Table S1 shows the values for the parameters in all 100 000 simulations and the selected 29 parameters for the Desilets method and 31 parameters for the COSMIC method, which include snow, soil moisture, and neutrons modules as behavioral simulations, with the posterior mean of the top 10 best parameters set. For further information and additional details about the calibrated parameters for each site, refer to Supplementary Table S2. Among the calibrated parameters, the $N_0$ parameters are different in each method since this parameter  does not exactly have the same physical meaning in the Desilets and the COSMIC methods.

RC: ***Please, consider creating either one section Results and Discussion, or move bits of preliminary discussion (p 15, lines 312-315, 318-324, 325-330, 370-376) from the Results section to the Discussion section.***

AR: *Thank you for your feedback. We have reorganized the manuscript by moving the mentioned preliminary discussion segments (p 15, lines 312-315, 318-324, 325-330, 370-376) from the Results section to the Discussion section as requested.*

The better performance of $N_{\text{COSMIC}}$ and $N_{\text{Des,w}}$ over $N_{\text{Des,U}}$ demonstrates the benefits of explicitly resolving individual soil moisture profiles, bulk densities, and lattice water, as opposed to a uniform average across the layers. This perception, however, might depend on site-specific soil profile characteristics and be less prominent if profiles are largely uniform or incorrectly resembled by the model structure. We also included offset hydrogen pools in the form of lattice water to the $N_0$ calibration function, which was important for more accurate soil moisture estimates, confirming initial suggestions by Bogena et al. (2013). Moreover, a strong correlation between biomass and the $N_0$ parameter was reported in several studies (Franz et al., 2013; Hawdon et al., 2014; Baatz et al., 2014, 2015). In our study, we pass the $N_0$ parameter as a calibration parameter set in mHM. In using the CRNS soil moisture measurement the drier locations show larger deviations than the wetter locations (Iwema et al., 2015). The possibility of using simulated high-resolution soil moisture profiles instead of a few measurements at different soil depths could further increase the accuracy of the model predictions (Brunetti et al., 2019). One of the primary sources of uncertainty at the *Grosses Bruch* site is surface ponding and shallow groundwater, as well as the loamy texture of the soil. Those factors contribute to the formation of permanent water ponds in the area and may introduce uniform or even inverse soil moisture profiles which directly influence the neutron emissions, but cannot be captured by the mHM model. Another factor is the time-variable effect of crowding cows near the station, which may influence the CRNS signal, but is challenging to correct in the CRNS measurement (Schrön et al., 2017). We incorporate the CRNS parameter set in mHM, and some parameters related to soil moisture and neutron counts are effectively constrained based on the objective function using $\text{KGE}_{\alpha\beta}$. However, there is still room for improvement, particularly with regard to the coefficient in root fractions distributed across soil layers.

**Other, specific comments:**

RC:	*Improved representation of soil moisture simulations..." I doubt if the word "representation" in relation to "soil moisture simulations" is well chosen. 'representation of soil moisture processes' or 'representation of soil moisture measurements' sounds logic, but here it seems as if the representation of soil moisture simulations is improved. Please think if this is really what you mean and if so, please consider if will be understood by the wider audience.*

AR:	*Thank you for your feedback, regarding the title of our manuscript we would like to move to a revised title as: "Improved representation of soil moisture processes ...".*

> Improved representation of soil moisture  processes through incorporation of cosmic-ray neutron count measurements in a large-scale hydrologic model

RC:	*Abstract: P1,L12-14: "A Monte Carlo simulation with Latin hypercube sampling approach . . . " Please, consider removing this sentence or writing it in more understandable wording for the audience. It is now hard to see the exact relevance of the technical details given, like 'N = 100 000'. What does such a number tell the audience?*

AR:	*Thank you. We have revised the texts to make it more clear. Furthermore, to avoid confusion with 'N' representing neutron counts, we have switched to the notation 'S' for the sample size to improve the clarity of the text.*

>  We use a Monte Carlo simulation method, specifically the Latin hypercube sampling approach  with a large sample size ($S$ = 100 000)  to explore and constrain the (behavioral) mHM parameterizations against observed CRNS neutron counts.

RC:	*Abstract, P1, L15-17: "We find that the non-uniform weighting scheme in the Desilets method provides the most reliable performance, whereas the more commonly used approach with uniformly weighted average soil moisture overestimates the observed CRNS neutron counts". How did COSMIC perform compared to the two Desilets methods?*

AR:	*In Table 4, we present a comprehensive evaluation of model performance based on three different approaches. Although we did not elaborate on COSMIC in the abstract, we have addressed the performance of COSMIC compared to the two Desilets methods in the abstract as suggested.*

> We find that the non-uniform weighting scheme in the Desilets method  and COSMIC method provides the most reliable performance, whereas the more commonly used approach with uniformly weighted average soil moisture overestimates the observed CRNS neutron counts.

RC:	*Introduction, page 2, L49-41: Please, improve textually by building a logical bridge between the paragraph of lines 32-48 and of lines 49-69. As is, HYDRUS-1D is introduced suddenly and in a way that makes is seem as a very key model, without being clear why so.*

AR:	*Thank you for the suggestion. We have revised the introduction to improve the flow of text between lines 32-48 and lines 49-69, as well as explain the relevance and significance of HYDRUS-1D in the context of our study.*

> Furthermore, depth-weighting schemes and hydrogen pools' effects on measurement depth revealed valuable insights. Shallow wetting fronts in sandy soils significantly impact measurement depth Franz et al. (2012). Baroni and Oswald (2015) assessed three weighting techniques, resulting in depths varying from 23 to 28 cm, optimal estimates were achieved using vertically varying weights and considering additional hydrogen pools.

**RC:** *Introduction, page 3, L73-74: The word "Eventually" and the wording with which the mHM model is introduced, at the start of this sentence and paragraph, make it seems as if the mHM model is a key hydrological model, that is the logical end-point of a discovery process and that is the standard that every reader should instantly know. It might be a well-known model, but it is one of many. Please, to help the reader understand the position of the mHM model, rewrite this to a more neutral wording.*

**AR:** *Thank you. We have made the suggested change in the revised manuscript. We deleted the word "Eventually".*

>  The mesoscale Hydrological Model (Samaniego et al., 2010; Kumar et al., 2013, mHM;) is known for its spatially distributed hydrologic predictions at a large scale incorporating scale-aware regionalized parameterization technique.

**RC:** *Introduction, page 3, L83: "The COSMIC method is complex …". What is meant by complex here? Please, clarify for the reader.*

**AR:** *We have revised the sentence in the introduction to avoid using the term 'complex' and provide a more descriptive explanation instead, i.e., "The COSMIC method enables a comprehensive representation of the neutron generation process, which is computationally more demanding than using the analytical Desilets equation ". We deleted the word "complex", rephrase the sentence.*

> The COSMIC method  enables a comprehensive representation of the neutron  generation process, which is computationally more demanding than using the analytical Desilets equation (Desilets et al., 2010).

**RC:** *Materials and Methods, page 4, L105: "four sites". Why does this number differ from that on line 90 of the introduction (page 4), which says "three"?*

**AR:** *Here the three different sites mentioned are according to the respective landcover states: i.e, agriculture, deciduous forests, and grasslands. Four sites are mentioned in terms of CRNS locations from where we utilized the measured neutron counts data.*

> In this study, we  established a framework to incorporate CRNS data into the mesoscale Hydrological Model (mHM) to compare empirical and physics-based approaches for neutron count estimation to improve soil water content parameters in mHM across different vegetation types in Germany. To do this, we compared modelled with measured neutron counts to infer soil hydraulic parameters.

**RC:** *Materials and Methods, page 4, line 111: "… producing methane fluxes". How is this relevant to the research presented? If it is relevant, it should be mentioned here and maybe discussed later on.*

AR: *The idea to mention the methane flux was related to provide few site-specific environmental conditions of the Grosses Bruch site. But we agree with the reviewer, that mentioning them at this point probably rather confuses and does not illuminate much to the study. We have taken these parts out in the revision of the manuscript.*

*We deleted "and is thus prone to producing methane fluxes" it is not relevant to discuss in this paper.*

RC: **Materials and Methods, page 5, Table 1: Please say in the caption that precipitation and temperature are yearly averages and in the table itself, say '[mm/year]' for Precipitation.**

AR: *We have made the suggested change to the revised manuscript.*

> Table 1. Geographical characteristics of study sites: Site Names, Geographic Coordinates, Climatic Data (Annual Precipitation in mm/year, Annual Mean Temperature in ° C), and the Periods Covered in Observed and Simulated Datasets.

RC: **Materials and Methods, page 6-7: The first reference to figure 2 is now on line 151. I think that by referring the reader earlier (from line 138 onwards), it will be easier for them to understand the methodology, with this key figure in hand.**

AR: *Thank you for your suggestion. We have made the recommended change by referring to Figure 2 (Flowchart) earlier in section 2.3 on Model setup.*

*We have moved the reference of Figure 2 to the mentioned position referred by the reviewer.*

RC: **Materials and Methods, page 7, figure 2: In this figure, a 'Neutrons' module now appears in the upper part (modules of mesoscale hydrologic model mHM) and below, where the different neutron models are mentioned. Is this how the modelling actually works? Is there one neutron module in the mHM and then, the outcomes (neutron counts) of these are fed to the neutron models? Please adjust the figure and/or make very clear in the manuscript text how the different bits are actually connected. In addition, please clarify if the short arrows between the left and right bits connect 'Spatial Data' to 'Model Setup' and 'Model Setup' to 'Performance Matrix' or if the connections are actually 'Spatial Data' to 'mesoscale hydrologic model' and 'CRNS-methods output' to 'Performance Matrix'**

AR: *In our study, we considered all mHM calibration parameters related to snow, soil moisture, and neutrons modules, leading to a total of 29 parameters employed for the Desilets method and 31 parameters for the COSMIC method. The simulation of soil water content considered these three mHM modules to estimate neutron counts. To comprehensively cover the parameter ranges, we sampled 100 000 (prior) parameter sets. Finally, we focused on the top 10 best performing (posterior) parameter sets based on the objective function, $KGE_{\alpha\beta}$, for further analysis and evaluation. We intend to enhance the clarity of the model setup section in the revised manuscript by providing additional descriptions, ensuring a clear understanding for readers.*

*We enhance the arrow line to ensure a clear representation of the connection in the flow diagram.*

[Figure]

Figure 1: (Figure updated in the revised manuscript) Flowchart depicting the methodology employed for calculating CRNS neutron counts through the utilization of the LHS technique for parameterization in mHM. The computation of CRNS neutron count is carried out through three distinct approaches: $N_{\text{Des,U}}$, $N_{\text{Des,W}}$, and $N_{\text{COSMIC}}$.

**RC:** *Materials and Methods, page 8, lines 158-159: "We compared these simulated values with the measured soil water content obtained through CRNS" This suggests soil moisture values were compared. Is this true or were actually neutron counts computed from mHM soil moisture simulation compared to neutron counts?*

**AR:** *In the revised manuscript, we have clearly stated that we are comparing neutron counts, not soil moisture. Our main objective is to optimize the parameterization of soil hydraulic properties in mHM/Cosmic based on the comparison between measurement and modelled neutron counts. We have adjusted the text in the revised manuscript to ensure that this objective is clearly reflected, thereby eliminating any potential confusion.*

>  Our main objective is to optimize the parameterization of soil hydraulic properties in mHM based on the comparison between measurement and modelled neutron counts. Simulations from mHM revealed that the sensitivity to soil water content near the surface, particularly at a depth of 5 cm, was higher, and this sensitivity decreased with increased depth, also indicating that SWC is highly responsive to precipitation.

**RC:** *Materials and Methods, page 8, 2.4.1: Please, restructure this paragraph, such that parameter names are mentioned after this equation, to improve the readability.*

**AR:** *Thank you for the feedback, we have changed the Desilets method section as suggested in the revised manuscript. We added the parameter name after the equation.*

> Among the four parameters, three of which are coefficients parameters $a_i$ were determined empirically by (Desilets et al., 2010) who derived $a_0 = 0.0808$, $a_1 = 0.372$, and $a_2 = 0.115$, and are considered as constants for values of $\theta > 0.02\ gg^{-1}$.

**RC:** *Materials and Methods, page 9, lines 185-186: "Organic water equivalent ...". Please rephrase this sentence.*

**AR:** *We rephrased the sentence in the Materials and Methods section.*

>  Regarding the variables of Soil Organic Carbon (SOC) and biomass, it's important to note that these variables are often not readily available, especially when it comes to biomass data.

**RC:** *Materials and Methods, page 10, line 204: "... does not get too small and SWC is not too high". Please, quantify.*

**AR:** *We have revised the text in the manuscript to specify that the lower limit for bulk density (BD) was defined as 1.0 g/cm³, addressing the reviewer's concern.*

> It should be noted that the equation for  $D$ is valid for $\varrho_b > 1.0\ \mathrm{g\,cm^{-3}}$ and soil moisture contents above $\theta > 2\ \%$ Kasner et al. (2022).

**RC:** *Materials and Methods, page 11, line 234: COSMIC parameter alpha is mentioned here, but was also mentioned on line 221. This seems confusing. Please, check and improve/clarify.*

**AR:** *Thank you for pointing this out. It was a typo, and we have removed the mention of $\alpha_{COSMIC}$ from line 234.*

**RC:** *Materials and Methods, page 11, lines 237-238: The parameters within the formula on line 237 seem not to match the parameters on line 238.*

**AR:** *We agree, there was a missing equation after the one presented in line 237. We have made the necessary revisions in the manuscript by including the missing equation(s) after line 237. Thanks for spotting this. We deleted the line 239 because we have already mentioned these parameters in Equ 2.*

>  The parameter $L_3$ is correlated with the soil bulk density and according to the model code, $L_{30}$ and $L_{31}$ nomenclature are given as per the model code in mHM (`https://github.com/mhm-ufz`).
>
> $$L_3 = L_{30}\varrho_{\mathrm{b}} - L_{31}. \tag{1}$$
>
>  The regional formulation of the COSMIC method has been revised to include the $\theta_{\mathrm{lw}}$ lattice water content as well.

**RC:** *Materials and Methods, page 11, lines 248-249: "However, we modify KGE (Eq. 15) by removing the correlation coefficient rho, as it is just a measure of temporal signature and is largely dominated by seasonality alone". Why should seasonality not be included? Why is the correlation coefficient not relevant? Please, clarify this better for the reader.*

**AR:** *Thank you for your comment. Gupta et al. (2009) proposed the KGE as a weighted combination of the three components (bias, variability, and correlation terms), given that our simulation already exhibited satisfactory correlation due to strong seasonality, we opted not to consider it in our assessment (objective function), as it accounted for 33% of the total weighting in the overall KGE score.*
*Seasonality is an inherent characteristic in the northern hemisphere where precipitation minus evaporation is mostly driven by evapotranspiration. Even if a random parameter is selected correlation will always be higher because the meteorological forcing is the precipitation - evaporation is seasonal. The study by Cinkus et al. (2022) examined the limitations of commonly used hydrological performance criteria, particularly the Kling-Gupta Efficiency (KGE) and its variants, in model calibration and evaluation. In the revised version, we have explained why it is necessary to exclude the correlation from the KGE.*

>  We use the general concept of the  KGE  as a weighted combination of the three components (bias, variability, and correlation terms) to evaluate our simulation (Gupta et al., 2009). We excluded the correlation component from (Eq. 13)  as our simulation already exhibited satisfactory correlation due to strong seasonality, we opted not to consider it in our assessment (objective function), as it accounted for 33% of the total weighting in the overall KGE score. Seasonality is an inherent characteristic in the northern hemisphere where precipitation minus evaporation is mostly driven by evapotranspiration. Even if a random parameter is selected correlation will always be higher because the meteorological forcing is the precipitation - evaporation is seasonal.

**RC:** *Results, p12, line 273: "… in all 10 000 simulations": why was this number chosen? How do we know it*

*is sufficient, insufficient, or too large? If only the N-parameters from the neutron models were calibrated, this seems like a large number.*

AR: *Thank you for your comment. The choice of 100 000 simulations was determined to ensure reasonably good coverage of the parameter sets within their prescribed range, given the relatively high number of parameters involved in our study. We sampled 29 parameters for the Desilets method and 31 parameters for the COSMIC method that are not only related to neutron count module ($N_0$) but also to other snow, soil, and vegetation processes that affects the soil water dynamics in mHM. We included information the number of parameter sets taken from the methods used (Desilets and COSMIC) in order to clearly the why we took the 100 000 number of simulations here is the supporting paragraph*

> Supplementary Table S1 shows the values for the parameters in all 100 000 simulations and the selected 29 parameters for the Desilets method and 31 parameters for the COSMIC method, which include snow, soil moisture, and neutrons modules as behavioral simulations, with the posterior mean of the top 10 best parameters set.

RC: *Results, p12, line 270: "... parameter distributions that almost cover the entire prior ...". Why the word almost here, what is meant with it? Why is it significant to mention 'almost'?*

AR: *In response to this question, we have extended the relevant section in the manuscript with additional explanations. We used the wording 'almost' to recognize that we couldn't be completely sure we sampled every possible parameter set, we meant that this large sample size was chosen to comprehensively explore the parameter sets and capture a wide range of possible parameter combinations. We removed the word 'almost':*

> The sensitivity and uncertainty analysis performed in this study use a Latin Hypercube Sampling (LHS) approach, resulting in parameter distributions that  large sample size was chosen to comprehensively explore the parameter sets and capture a wide range of possible parameter combinations in the prior range.

RC: *Results, p12, lines 277-278: "... Most of the high-sensitive parameters show more peaked densities in a narrower range of parameter values, reflecting the significance of variations in model parameter values ...". Please, explain exactly why the statement is true.*

AR: *Thank you for mentioning this aspect. We have included the explanation in the manuscript as mentioned. Our study aimed to determine optimal $N_0$ values by refining the parameter range for $N_0$ using the three approach ($N_{Des,U}$, $N_{Des,W}$, and $N_{COSMIC}$), the parameter set range set for $N_0$ ranges between (600–1500) for Desilets method and (100–400) for COSMIC method. Through a calibration process, we adjusted these parameters to align more closely with observed data. From the iteration of 100 000 parameter sets, we selected the top 10 sets that yielded a narrower range of $N_0$ values, providing the best fit to the observed data. By 'more peaked densities,' we mean that following calibration from the posterior distribution, the figure (Fig. 4) displays the x-axis in gray, representing the original parameter range (600–1 500) prior distribution for Desilets method and (100–400) for COSMIC method. Meanwhile, the colored sections in brown, green, and purple indicate the parameter values of the top-performing sets for each study site.*

Fig. 4  displays the x-axis in gray, representing the original parameter range (600–1500) prior distribution for the Desilets method and (100–400) for COSMIC method. Meanwhile, the colored sections in brown, green, and purple indicate the parameter values of the calibration from the posterior distribution taken from the top-performing parameter sets for each study site.

**RC:** *Results, p13, lines 291-292: "..., indicating that the model has the potential to generate accurate cosmic-ray soil moisture estimates even under dry conditions." Please, explain why 'even under dry conditions'? Is high performance under these conditions a surprise? If so, why?*

AR: *We mention that "even under dry conditions" emphasizes the mHM performs well under dry conditions, we highlight the model ability to simulate a wide range of moisture conditions. In contrast, some hydrological models, such as HBV and PREVAH (PREecipitation Runof EVApotranspiration Hydrological response unit model; Viviroli et al., 2009), have shown weaker performance in simulating soil moisture, particularly during dry conditions, as demonstrated by Orth et al. (2015), with slightly better agreement with observations during wet conditions.*

The estimated values of $N_{0,\text{Des}}$ and $N_{0,\text{COSMIC}}$ obtained in our study are close to the optimal values, indicating that the model has the potential to generate accurate cosmic-ray soil moisture estimates even under dry conditions.  In contrast, some hydrological models, such as HBV and PREVAH (PREecipitation Runof EVApotranspiration Hydrological response unit model; Viviroli et al. 2009), have demonstrated weaker performance in simulating soil moisture, particularly during dry conditions (Orth et al., 2015), with slightly better agreement with observations observed during wet conditions.

**RC:** *Results, p13, line 293: "..., a loss of the physical meaning of the parameters in question would be very critical". Why would this be critical?*

AR: *Thank you for pointing that out. We agree that the sentence does not contribute to the clarity of the text, and we have removed it.*

**RC:** *Results, p13, line 295: "One of the important additions of this work ...". Was incorporating lattice water count added by this study for the first time?*

AR: *Yes, the inclusion of lattice water in the neutron counts module of mHM is an additional aspect of our study.*

**RC:** *Results, page 14, figure 4: Please, try to make this figure easily readable in greyscale, this would help readers who print to read the paper carefully.*

AR: *Thank you for this suggestion. We have accordingly updated the figure in the revised manuscript. We have changed the color to be easily readable in grayscale in the manuscript.*

**RC:** *Results, page 14, line 308: "Furthermore, the behavioral simulation ensembles captured more variations in the COSMIC method compared to the Desilets method after the application of the objective function (i.e. KGEalpha,beta)". Do you know why? Please discuss here if Results and Discussion are combined.*

AR: *Thank you for your question. The broader confidence interval, indicating a greater range of variations, implies a higher degree of uncertainty in the COSMIC method ($N_{COSMIC}$). The COSMIC approach explicitly accounts for water content snow, vegetation interception, and root-zone soil processes that may likely lead to a better representation of observed neutron count variation compared to Desilets that empirically represent such processes. Added the paragraph in the Discussion section.*

> Overall, the three methods ($N_{Des,U}$, $N_{Des,W}$, and $N_{COSMIC}$) in mHM were able to consistently simulate the neutron count variability throughout the available data period. However, a broader confidence interval is observed, indicating a greater range of variations, which implies a higher degree of uncertainty in the $N_{COSMIC}$. The COSMIC approach explicitly accounts for water content snow, vegetation interception, and root-zone soil processes that may likely lead to a better representation of observed neutron count variation compared to Desilets that empirically represent such processes.

RC: *Results, page 16, lines 318-322. How are you sure that surface ponding and shallow groundwater and other mentioned factors are a major cause of uncertainty? Was an uncertainty analysis performed? Please, if so, discuss these briefly. If not, on which observations is this discussion based?*

AR: *No, a formal uncertainty analysis was not performed. Our discussion regarding these factors, particularly for the Grosses Bruch site, is based on prior observations of field data explained in Schrön et al. (2017). Ponding in the wet season is a common phenomenon on this site and these effects are explicitly not considered in the mHM model; and therefore we identify and mention them for future model development.*

*Added the citation for 'Schrön et al. (2017)' and relocated this paragraph from the Results section to the Discussion section, as recommended in Question 6.*

RC: *Results, page 16, lines 325-330: Please, provide references to support the discussion in this paragraph.*

AR: *Thank you for pointing this out, we have added the following references to support the statement (Massoud et al., 2019 and Zink et al., 2017).*

> The incorporation of dynamic vegetation in models is important as it can impact the model parameter LAI, which in turn can affect root water uptake and soil water content. Currently, these factors are not considered in the models, leading to a permanent and systematic shift in these variables each year (Zink et al., 2017; Massoud et al., 2019).

RC: *Results, page 17, figure 5: The figure could be interpreted more easily and quicker if the choice of colours stated in the caption (red and black) for the different Desilets daily neutron counts, are put in the figure legend.*

AR: *Thank you for the hint and suggestion to improve the clarity of figure 5. We have made the necessary adjustments to the figure legend, with the name $N_{(Des,W)}$, and $N_{sim(Des,Uni)}$. We added the figure legend for Fig.5 according to the caption.*

RC: *Results, page 18, line 340: An LHS sample 100 000 seems a lot for just the N-parameters from the three neutron models. Why was this sample size chosen?*

AR: *The choice of 100 000 simulations was determined to ensure thorough coverage of the parameter sets, given the relatively large number of parameters involved in our study. 29 parameters for the Desilets method and 31*

*parameters for the COSMIC method, we aimed to comprehensively explore the possible combinations of these parameter sets values. These 100 000 simulations enable us to fully capture the distribution of parameter values. We added the discussion about the parameters of the methods selected for calibration in the Model set-up section.*

**RC:** *Results, page 19, table 4: A figure could help the reader to get a clear overview of these results quickly. Please, consider a parallel coordinates plot or something alike.*

AR: *Thank you for your suggestion, we have already included a boxplot in Figure 7 to illustrate model performance for KGE and objective function KGE$_{\alpha\beta}$. In Table 4, we have highlighted the best-performing values to aid readers in quickly identifying the best values for each method. Additionally, we have improved the caption of Table 4 for enhanced clarity. We have used the boxplot instead, but you can find the parallel coordinates plot we presented in the supplementary materials, in Figure S3.*

**RC:** *Results, page 20, figure 7: Please, add horizontal axis title.*

AR: *Thank you for the suggestion, we have added a horizontal axis title to Figure 7 in the manuscript. We have added the horizontal axis title as "Num of parameter sets.*

**RC:** *Results, page 20, lines 351-353: Were eddy-covariance measurements available at the other sites? If so, the same analysis should be done and presented for those sites, for complete insights from this research.*

AR: *No, for this study eddy-covariance measurements were only available at the Hohes Holz site. Therefore, we were able to evaluate the evapotranspiration simulations at this specific site only. We are changing the title of Section 3.4 to be more specific to make the reader aware that we are discussing evapotranspiration at the Hohes-Holz site only. Section 3.4 Title now is:*

>  Comparing evapotranspiration at Hohes-Holz: eddy covariance  observed data  vs mHM simulation

**RC:** *Results, page 21, lines 354-355: 'Panel C displays the scatter plot that reveals no systematic over or underestimation of the observed actual evapotranspiration": The dashed line in the figure does not show the 1:1-line. How then does the scatter plot reveal no over or underestimation?*

AR: *Thank you for pointing this out. The reviewer is right – the dashed lines in the figure do not represent the 1:1 line (identity line). Instead, they correspond to the best-fit regression lines corresponding to the data for the growing and non-growing seasons. These two regression lines provide insights into how well our models capture ET variations during these distinct seasons. We can also specifically estimate some summary statistics reflecting the over/underestimation of simulated values of ET. We have changed the line in the revised manuscript. We have changed the sentence for panel (c)*

> Panel (c) displays the scatter plot  incorporating linear regression models to quantify the relationships between observed and mHM-simulated ETa during both the growing and non-growing seasons.

**RC:** *Results, page 21, lines 357-360. Given pieces of discussion that occur in the current Results chapter, please discuss the differences in correlations between observed and simulated evapotranspiration between different seasons.*

AR: *In the discussion section, we have added how the correlations between observed and simulated evapotranspiration vary in different seasons. This plot provides insights into the seasonal variations in the relationship between observed and simulated ET. It suggests that the model performs best during winter, while its performance during summer is comparatively weaker. The correlation coefficients (r values) for each season are as follows: autumn [SON] (r = 0.79), spring [MAM] (r = 0.77), summer [JJA] (r = 0.42), and winter [DJF] (r = 0.87). It is worth noting that winter shows the highest correlation between observed and simulated ET, while summer exhibits the lowest correlation. The most significant deviation in terms of RMSE is evident during the summer, when evapotranspiration is highest, while the smallest difference is in winter when evapotranspiration has less impact. The model slightly overestimates evapotranspiration in summer and spring, previously addressed in the response to question 3.*

*The discussion we added in the Result part:*

> This plot provides insights into the seasonal variations in the relationship between observed and simulated ET. It suggests that the model performs best during winter, while its performance during summer is comparatively weaker. The correlation coefficients (r values) for each season are as follows: autumn [SON] (r = 0.79), spring [MAM] (r = 0.77), summer [JJA] (r = 0.42), and winter [DJF] (r = 0.87). It is worth noting that winter shows the highest correlation between observed and simulated ET, while summer exhibits the lowest correlation. The most significant deviation in terms of RMSE is evident during the summer, when evapotranspiration is highest, while the smallest difference is in winter when evapotranspiration has less impact. The model slightly overestimates evapotranspiration in summer and spring, possibly because of the absence of a dynamic vegetation growth module in the mHM, also discussed for evapotranspiration in Zink et al. (2017)

*We have added the different seasonal correlations figure in Supplementary Figure S4.*

[Figure]

Figure 2: (New supplementary Figure S4) Correlations between observed and simulated evapotranspiration between different seasons. The correlation coefficients (r values) for each season are as follows: autumn [SON] (r = 0.79), spring [MAM] (r = 0.77), summer [JJA] (r = 0.42), and winter [DJF] (r = 0.87).

**RC:** *Results, page 21, lines 370-376; This paragraph seems to relate to the paragraph and results I mentioned in my previous comment. If this is correct, please restructure the text so this becomes clearer.*

AR: *Thank you for your feedback. To improve clarity, we have restructured the text explicitly in the revised manuscript accordingly. We have removed the paragraph to avoid repetition.*

**RC:** *Results, page 22, figure 8: If the two RMSE boxplots are combined into a single one with a single vertical axis domain, could this help the comparison?*

AR: *We believe that merging the two RMSE boxplots into a single plot with a single vertical axis domain is not suitable in this case. The reason is that the Y-axis values for ET during the growing and non-growing seasons significantly differ (due to differences in ET values between these two seasons). Combining them into one plot would result in the non-growing season boxplot being too small to visualize and making it difficult to distinguish the mean values within the boxplots. Therefore, we prefer to keep them separated for clarity. We are not combining the two RMSE boxes because of the clear visualization.*

**RC:** *Discussion, page 23, lines 389-390: "Therefore, we extended this uniform-averaging scheme by a vertical weighting scheme to mimic the sensitivity of the neutrons to the upper layer" Was this a contribution done through the work in this research or should previous work be referenced here?*

AR: *In our study, we incorporated the vertical weighting scheme for soil moisture in Desilets method, into the mHM model, and we applied and tested it across various landcover sites. In past studies, the techniques of both weighted and non-weighted soil moisture approaches in the context of CRNS have been discussed. We added the reference of the vertical weighting scheme:*

> Therefore, we extended this uniform-averaging scheme  with a vertical weighting scheme to mimic the sensitivity of the neutrons to the upper layers both weighted and non-weighted soil moisture approaches in the context of CRNS have been discussed (...).

**RC:** *Discussion, page 23, lines 407-408: "…, indicating that the dynamic vegetation effect is just a minor observational issue (…)". The abundant vegetation does affect the CRNS measurement precision. How does that affect the calibration process and further analysis of this study?*

AR: *Thank you for your comment, we have removed these lines because they are related to CRNS data calibration at the field site, which is not directly relevant to our study on hydrological modeling. We removed the paragraph:*

> One specific limitation is that the model does not fully account for the fact that trees at the site have access to deeper water sources, which can result in water stress being experienced at later times. Still, we get very good results in terms of KGE, for instance, indicating that these issues are of minor importance and that all three methods in mHM representation of the forest are already performing quite well. ~~Also, the CRNS method may be influenced by temporal biomass variation in the forest (Baatz et al., 2015), but many recent studies have confirmed the good performance of CRNS in forests compared to below-ground soil moisture profiles, indicating that the dynamic vegetation effect is just a minor observational issue (Bogena et al., 2013; Andreasen et al., 2017; Schrön et al., 2017; Boeing et al., 2022; Bogena et al., 2022). It is worth noting that most of the studies on drought analysis look at the anomaly of soil moisture, while our study tries to assess the absolute soil water quantity and the properties that can determine the soil water content.~~

**RC:** *Discussion, page 24, lines 420-421: "… the results confirm the findings from Zink et al.(2017)." Please expand a small bit on this reference. Which type of soil moisture data did they use?*

AR: *Zink et al. (2017) utilized soil moisture observations, obtained from eddy covariance stations, to evaluate modeled soil moisture. These soil moisture measurements were collected using Time-Domain Reflectometer (TDR) or Frequency-Domain Reflectometer (FDR) sensors, which have a control volume of ten to hundreds of cubic centimeters only. Because of variations in spatial representativeness and sampling depth, they did not directly compare observed and simulated soil moisture. Instead, their objective was to analyze the temporal dynamics of soil moisture by normalizing the respective soil moisture time series (as described in Koster et al., 2009). We have expanded the texts in the revised manuscript to include these aspects.*

*We added the explanation about the soil moisture data used by Zink et al.(2017):*

> However, the model performs well in winter, with a high correlation between observed and simulated values of evapotranspiration, the results confirm the findings from Zink et al. (2017), who used mHM to estimate evapotranspiration, groundwater recharge, soil moisture, and runoff with 4 km spatial and daily temporal resolutions (1951–2010). They  utilized soil moisture observations from eddy covariance stations employing Time-Domain Reflectometer (TDR) or Frequency-Domain Reflectometer (FDR) sensors. Due to disparities in spatial representativeness and sampling depth, a direct comparison between observed and simulated soil moisture was not feasible, their findings revealed deviations in evapotranspiration during spring and in cropland areas,  while soil moisture estimations  exhibited good agreement with observed dynamics.

**RC:** *Discussion, page 24, line 435-436: "… while the weighted approach N(Des,U) shows a slightly better performance that the other two methods …" How significant was the difference, i.e. What is meant with 'slightly'?*

**AR:** *The Grosses Bruch site with the uniformly weighted approach $N_{(Des,U)}$ shows a "slightly better" performance than the $N_{(Des,W)}$, means that in terms of correlation and another performance indices (i.e., KGE, NSE, PBIAS), as shown in Table 4. $N_{(Des,U)}$ (0.85, 0.69, 0.7%) and $N_{(Des,W)}$ (0.81, 0.60, -1.3%).*

*Supporting information Figure:*

[Figure]

**RC:** *Discussion, page 25, line 451: "We also included offset hydrogen pools in the form of lattice water to the $N_0$ calibration function, …" Was soil organic matter included? If not, why not? Another factor, vegetation (including intercepted water), was this corrected for in this study? If not why not? If so, what did the results indicate? How substantial was the effect of vegetation at the different sites?*

AR: *Unfortunately, soil organic matter was not explicitly parameterized in the version of mHM used for this study. The intercepted water on leaves and in the litter layer can be particularly challenging to quantify, especially in forested stations such as Hohes Holz (Bogena et al., 2013; Schrön et al., 2017). The assessment of mHM with evapotranspiration data from eddy covariance stations at Hohes Holz site showed deficiencies in mHM. Especially in summer and spring, deviations of the modeled and observed ET indicate room for improving the representation of vegetation dynamics within mHM. However, for other sites (Grosses Bruch, Hordorf, and Cunnerdorf), we did not have eddy covariance stations to check the evapotranspiration of the measured vs. simulated ones. In lines 405-452, we discussed in detail the simulation of neutron counts and the factors influencing these simulations in comparison to observations at our study sites. We explored this using the three approaches: Desiles (Uniform, Non-Uniform), and COSMIC.*

RC: ***Conclusion and future outlook, page 27, lines 500-503: Different sources of uncertainty regarding the neutron modelling are mentioned here. I wondered, given modelling tools are available to give an estimation of the size of the contributions from the different factors on neutron intensity, were such estimations made within this study? If so, what did they tell?***

AR: *Thank you for your question, our analysis primarily addresses the uncertainty of the model parameters, and we have clarified this in the revised manuscript. To assess parameter uncertainty in mHM with respect to neutron counts, we employed Latin hypercube sampling involving 100 000 parameter sets. we took the top 10 best parameter sets as a behavioral solution. In Supplement Figures S6–S9, we present the Probability Density Function (PDF) plots of all parameter sets for our study sites, both prior and posterior to the simulation. Our analysis result shows that $N_{0,Des}$, $N_{0,COSMIC}$, rootFractionCoefficient_pervious, and rootFractionCoefficient_forest e, are the most sensitive model parameters.*

*In Supplement Figures S6–S9, we present the Probability Density Function (PDF) plots of all parameter sets for our study sites, both prior and posterior to the simulation.*

RC: ***Conclusion and future outlook, page 27, lines 509-511: "... provides a more realistic representation of soil moisture dynamics as well as evapotranspiration, particularly at the forest site". If I have understood the manuscript correctly, evapotranspiration was evaluated at one site only. The sentence here in the conclusions chapter seems to suggest a broad result for evapotranspiration. Please, rewrite to make this explicit.***

AR: *Yes, evapotranspiration was evaluated at one site because we have the eddy covariance flux observation only at the Hohes Holz. Thank you for your feedback, we have updated the texts in the revised manuscript accordingly. In conclusion, we clarified the evapotranspiration at only one site.*

> In conclusion, the incorporation of neutron counts estimation into mHM by accounting for vertical soil moisture profiles improves the model's accuracy and provides a more realistic representation of soil moisture dynamics  at all four study sites and evapotranspiration at *Hohes Holz* site. This research presents a direction for future studies to explore.

RC: ***Please check for textual imperfections throughout the manuscript. Three examples from the abstract, introduction, and results:***

- P1, L3: "... due to their hectare scale footprint and ..." -> "... due to its hectare scale footprint and ... "

- P2, L21: "the mass" -> water mass? Carbon mass? Both or more?

- P13, L285: The words 'uniform prior distribution range for' should be repeated before "$N_0$,cosmic", or rephrase in another way

AR: *Thank you for your thorough review and your valuable feedback. We appreciate your attention to detail, and we have addressed these issues accordingly.*

* * *
**RC:** *Referee Comment*,     AR: *Author Response*,     ☐ Manuscript text

**RC:** *The paper describes the simulation neutron count rates using a gridded hydrological model, which estimates soil water content in different depth layers. The simulated counts are compared with CRNS neutron counts to optimise the model (mHM) calibration. This approach addresses the issue of the variable measurement depth of CRNS. In simulating neutron counts, published models are used such as COSMIC, or the CRNS neutron count to soil moisture calibration function is inverted. However, these relationships are further fitted through optimisation of the $N_0$ parameter, and different published schemes for vertical soil moisture content weighting are tested. Other mHM parameters are also optimised, but how this is done and in what order, is not explicitly described. Which parameters are optimised is only shown in the supplementary information, and the results and discussion do not cover these parameters.*

 AR: *Firstly, we want to sincerely thank the reviewer for dedicating their time and expertise to review our work. Your insightful comments have significantly improved our manuscript. We have provided a comprehensive response to address your comments and suggestions.*

**RC:** *Line 52: by this point or earlier you should say why there could be an issue with depth averaging.*

 AR: *Several studies investigated depth-weighting schemes and hydrogen pools' effects on measurement depth. Franz et al. (2012) showed that shallow wetting fronts impact measurement depth in sandy soils with non-uniform moisture profiles. Baroni and Oswald (2015) tested three different depth weighting techniques (vertically varying weights, uniform weights, and taking the effect of above-ground biomass into account), which yielded different measurement depths varying from 23 to 28 cm. Using vertically varying weights and taking into account other hydrogen pools gave the best measurement depth estimates. We added the explanation regarding the depth averaging in the revised manuscript.*

> Furthermore, depth-weighting schemes and hydrogen pools' effects on measurement depth revealed valuable insights. Shallow wetting fronts in sandy soils significantly impact measurement depth (Franz et al., 2012). Baroni and Oswald (2015) assessed three weighting techniques, resulting in depths varying from 23 to 28 cm, optimal estimates were achieved using vertically varying weights and considering additional hydrogen pools.

**RC:** *Line 70: delete 'eventually'*

 AR: *This is also noted by reviewer 1 (comment number 11), we have made the suggested change in the manuscript.*

**RC:** *Line 75: still no intro as to the motivation to do this - rather than use the derived SM!*

**AR:** *We appreciate the reviewer feedback. The primary methodological focus is on inverting soil hydraulic parameterization in mHM/COSMIC by directly comparing modelled neutron counts with measured ones. To obtain soil moisture accurately from CRNS measurements, one needs prior knowledge of the soil moisture profile. However, CRNS typically lacks this information, which mHM can provide. We have improved the introduction to make sure readers understand the primary motivation of our study more clearly.*

**RC:** *Line 86: still not clear what is the objective of this study?*

**AR:** *The objective of this study is to establish a framework to incorporate CRNS data into the mesoscale Hydrological Model (mHM) by comparing empirical and physics-based approaches for neutron count estimation to improve soil water content parameters in mHM across different vegetation types in Germany. To do this we compared modelled with neutron counts to infer soil hydraulic parameters. We have clarified this main goal in the manuscript to help readers better understand.*

> In this study, we  established a framework to incorporate CRNS data into the mesoscale Hydrological Model (mHM) to compare empirical and physics-based approaches for neutron count estimation to improve soil water content parameters in mHM across different vegetation types in Germany. To do this, we compared modelled with measured neutron counts to infer soil hydraulic parameters.

**RC:** *Line 89: If the objective is a technical comparison of methods, then why do this at grid scale, not a point scale, actually at the CRNS station? This may have complicating factors e.g. mixed land cover, soils, and topography modelled across a grid cell.*

**AR:** *Using soil moisture (SM) directly simplifies the process but may not account for CRNS depth variability under different soil wetness conditions. The COSMIC approach, being more physics-based, allows for depth integration and the non-uniform contribution of signals with depth. CRNS measurement is based on a spatial footprint of around 150 m together with a vertical penetration depth of typically up to $\sim$ 15-80 cm depth. Sub-grid heterogeneity in mHM is captured explicitly within its multiscale parameter regionalization (MPR) technique (Samaniego et al., 2010). We simulated the neutron counts in mHM at the location where the CRNS instrument is installed to measure neutron counts. We used the extracted input data in mHM specific to the CRNS location, including temperature and precipitation from the German Weather Service (DWD), along with other data sources discussed in the data availability section. Our objective is to improve the parameterization of soil water content within the hydrological model mHM. The Grid scale is more comparable to the hydrological model than the point scale.*

**RC:** *Line 99: there seems to be an implicit assumption that working with simulated counts is better than using CRNS-derived SM - did you test this?*

**AR:** *No, we have not performed the test. But the reason for this preference is that mHM provides soil moisture data at different layers, whereas CRNS-derived SM values represent an integrated value. This value also changes with soil moisture and depends on the vertical soil moisture profile. With these two different dimensions of data, it is impossible to uniquely tell which simulated layered soil moisture corresponds to the CRNS-derived SM value. It is thus impossible to conduct a fair comparison on soil moisture only.*

**RC:** *Line 102-3: does this mean different land cover in grid or between grid cells?*

**AR:** *We address land cover variations within each individual grid cell. In our study, the mHM model utilizes*

*a fixed grid cell size of 0.015625° (approximately 1.2 km $^2$), with one grid cell assigned to represent each individual site.*

*We updated the research question as:*

> Is the mHM at approx. 1 km resolution capable of capturing the dynamics of hectare-scale CRNS measurements at different landcover sites in a grid including 2 agriculture sites, 1 forest site, and 1 meadow site?

**RC:** *Line 108: homogeneous at the CRNS hectare scale, but not at 1 km scale! - see photo of Grosses Burch. Please be more specific – and how might in-grid heterogeneity affect optimisation?*

**AR:** *In the mHM model, the sub/in-grid heterogeneity is explicitly captured with its multiscale parameter regionalization (MPR) technique (Samaniego et al., 2010). We removed the term "homogeneous" concerning different land cover, i.e., agriculture, forest, and meadow.*

**RC:** *Line 115: Are these seasonal biomass fluctuations included in the CRNS count simulation?*

**AR:** *No, this is not feasible as data on biomass changes were not available. However, Boeing et al. 2022 indicated that the seasonal changes of biomass are much less important, even in Hohes Holz, as soil moisture changes are the much larger source of variation represented by the CRNS measurements.*

> Also, Bogena et al. (2022) indicated that the influence of seasonal changes of biomass on the CRNS signal is much less important than the influence of changing soil moisture, even in Hohes Holz, as changes in soil moisture are the much larger source of variation represented by the CRNS measurements.

**RC:** *Line 159-160: 'the sensitivity to the highest. . .' re-phrase, this is not clear.*

**AR:** *We have revised the sentence to make it more clear in the revised manuscript. We deleted the sentence.*

**RC:** *Line 170: delete 'time constant'.*

**AR:** *We have made the suggested change in the manuscript. Deleted 'time constant'.*

**RC:** *Line 183-87: it may make sense to use grid averages for mHM, but this complicates the evaluation of the methods – how representative is the CRNS station of the wider grid properties? If the station is not representative of the grid, then inappropriate parameter changes are forced to match soil moisture or CRNS counts to soil properties that do not match the CRNS site.*

**AR:** *Thank you for your comment, The spatial resolutions in mHM are defined at different levels:*
*L1 and L2: 0.01562° x 0.01562° is eq.$\sim$ 1.2 x 1.2 km$^2$. Level 1 (L1) describes the spatial resolution, as which dominant hydrological processes are modelled and Level 2 (L2) describes the resolution of the meteorological forcing data.*
*L0: 0.001953125° × 0.001953125°. Level 0 (L0) describes the sub-grid variability of relevant basin characteristics, which include information on soil, vegetation, topography and geology, among other basin relevant geophysical characteristics. These sub-grid heterogeneities are reflected to generate effective model parameters (e.g., soil porosity, field capacity, lattice water content) in mHM with the multiscale parameter regionalization (MPR) technique (Samaniego et al.,2010).*

*CRNS data was preferred over any other ground-based measurements i.e., point-scale (TDR, FDR) and satellite data, due to its ability to provide a horizontal footprint of around 150 m together with a vertical penetration depth of $\sim$ 15-80 cm depth. These feature make them relevant for the modelling scale, along with its higher temporal resolution that allow us to capture the fine details of soil moisture changes.*

*We used site-specific input data and calibrated parameters to establish the mHM model at each site and by this we address the CRNS station's spatial comparability with the grid-level model simulations. In the Model-setup section, we added information regarding the spatial resolutions of mHM.*

> The model is executed over six years (2014–2020) with a daily time step, and the spatial resolution of the mHM grid cells is fixed at L1 and L2: 0.01562° x 0.01562° is eq.$\sim$ 1.2  km $\times$ 1.2 km using the WGS84 Coordinate Systems. Level 1 (L1) describes the spatial resolution, as which dominant hydrological processes are modelled and Level 2 (L2) describes the resolution of the meteorological forcing data. L0: 0.001953125° $\times$ 0.001953125°. Level 0 (L0) describes the subgrid variability of relevant basin characteristics, which includes information on the soil as well as land use, topography, and geology.

**RC:** *Line 187: How can varying $N_0$ properly account for biomass changes? Biomass is dynamic in time, while $N_0$ is constant in time?*

**AR:** *A correction for biomass changes could by introduced in two ways, either with a separate correction factor on the measured neutrons or with a temporal dependency on the $N_0$ Baatz et al. (2014). Mathematically, the two approaches are similar. However, due to the lack of biomass data, usually no correction is considered in this study. Vather et al. (2020) emphasizes the importance of considering changes in biomass, but especially in situations with sudden changes in biomass, such as harvest in agriculture or clearings in forestry. The study demonstrates how the difference in hydrogen content within the measurement area leads to different calibration values ($N_0$ values).*

*While $N_0$ should ideally be constant across different study areas once all environmental hydrogen sources are considered, it can vary due to the presence of growing biomass, as observed in the study by Hawdon et al. (2014). In their research, they found that approximately 80% of $N_0$ variation could be attributed to the effects of biomass, but only after accounting for other hydrogen sources. So, while $N_0$ itself does not change with time, its variation across study areas can indirectly reflect changes in biomass levels and as such indicate missing biomass corrections. Those corrections are complicated since temporal biomass data is usually not available.*

>  Regarding the variables of Soil Organic Carbon (SOC) and biomass, it's important to note that these variables are often not readily available, especially when it comes to biomass data. To address this, the free parameter $N_0$ is utilized to account for these unknowns.

**RC:** *Line 212: change 'neutrons' to neutron flux or count rate. Eq. 9 is bulk density here the same as Eq. 1? (a different undefined symbol is used here and in Eq. 10)*

**AR:** *Thank you for pointing out, that we have made the suggested change in the revised manuscript: "neutrons" to neutron count rate. Regarding Eq. 9 and Eq. 10, the COSMIC method employs the bulk density and lattice water for each model soil depth. However, in Equation 1, the same bulk density is used for the Desilets*

*method. To enhance clarity, we have made the change in the manuscript by replacing the $\varrho_{bulk}$ to $\varrho_b$ in the manuscript. Added the above Eq. 9.*

$$X_{\text{soil}}(z) = \Delta z \varrho_{\text{bulk}b} \tag{9}$$

**RC:** *Line 238: These symbols are not in Eq.14 ? what are $L_{30}$ and $L_{31}$?*

**AR:** *$L_3$ represents the fast neutron soil and water attenuation lengths, according to literature Shuttleworth et al. (2013); Rosolem et al. (2014), these values have been expressed as $L_3 = -31.65 + 99.29\varrho_b$, with the parameters correlated to soil bulk density. Where $\varrho_b$ (g cm$^3$) is the soil bulk density. $L_3$ are site-specific parameters, and we integrated them into the mHM model as part of the parameters to be optimized. We restructure the paragraph:*

> However, the $L_3$  (g cm$^{-2}$)  parameter vary with soil bulk density $\varrho_b$ which change with depth.  The parameter $L_3$ is correlated with the soil bulk density and according to the model code, $L_{30}$ and $L_{31}$ nomenclature are given as per the model code in mHM (`https://github.com/mhm-ufz`).

**RC:** *Line 257: Table 2 shows model performance measures, not model parameters and their ranges - please add table of model parameters optimised. (it is in supplementary)*

**AR:** *We have made the suggested change in the revised manuscript.*

> A summary of the individual parameters and their ranges can be found in Supplementary Table S1 and model performance measures are shown in Table 2.

**RC:** *Line 269: presentation of results is confusing, as it does not show hydraulic parameters (only refers to Table S1). Plots and tables only show CRNS parameters. Fig. 4 – add units (... are these counts per hour?)*

**AR:** *In this study, we used a total of 29 parameters for the Desilets method and 31 parameters for the COSMIC method which includes hydrologic processes related to: snow, soil moisture, and neutron counts dynamics. For clarity, we have included box plots showing the other parameters in Figures S6–S9 in the supplementary materials. Regarding Figure 4, the unit for $N_0$ is counts per hour (cph), which represents the site-specific, time-invariant calibration parameter. We have included the units in the figure in the revised manuscript.*

> Supplementary Table S1 shows the values for the parameters in all 100 000 simulations and the selected 29 parameters for the Desilets method and 31 parameters for the COSMIC method, which include snow, soil moisture, and neutrons modules as behavioral simulations, with the posterior mean of the top 10 best parameters set. For further information and additional details about the calibrated parameters for each site, refer to Supplementary Table S2. Among the calibrated parameters, the $N_0$ parameters are different in each method since this parameter  does not exactly have the same physical meaning in the Desilets and the COSMIC methods.

*We have updated the figure in the revised manuscript.*

[Figure]

Figure 1: Probability Density Function (PDF) of the mHM parameter $N_0$ [cph] for two different approaches: (a) the Desilets method, and (b) the COSMIC method. The prior PDF of the original sample, consisting of 100 000 data points, is represented by the grey color. The behavioral PDF, obtained after applying the objective function, is shown for weighted (brown), uniform (green), and COSMIC (purple). The black dashed line represents the one $N_0$ [cph] value that best fits the data.

RC: *Line 301: This gives the impression that what has been done here is to optimise neutron count match, by varying some model parameters, especially $N_0$, and some root depth parameters ....there could be an issue that while counts may agree well, systematic bias in SM could be compensated for by varying $N_0$; i.e., better efficiency in simulating neutron counts may not necessarily lead to better efficiency in SM modelling. Section 3.2, Fig. 5, and Fig. 6 should show these also as CRNS SM plots, as the non-linear counts to SM hides the magnitude of discrepancies.*

AR: *Our primary goal was to optimize the agreement between modeled neutron counts and observed neutron counts from the three methods, i.e., Desilets method ($N_{Des,U}$, $N_{Des,W}$) and COSMIC method ($N_{COSMIC}$). A direct comparison between the model and observation is feasible only via neutron counts because CRNS soil moisture is an integrated value, while mHM represents soil moisture in different layers. By improving the modeling of neutron count measurements, we are inherently improving the representation of SM in mHM.*

*In Supplement Figure S1, we presented the soil moisture distribution across different layers by mHM.*

[Figure]

Figure 2: Observed soil water content from CRNS vs simulated data from mHM in 0–5, 5-25, and 25–60 cm depth .

**RC:** *Line 410: This study does not assess the absolute soil water quantity (most results are presented as neutron counts).*

 AR: *Thank you for your comment. We have revised these texts making it clear that our study does focus on capturing neutron counts and not absolute soil water content, in the revised manuscript.*

 *We have deleted the line from 407– 412.*

**RC:** *Line 412-13: 'Incorporating CRNS data. . . ' this statement is untrue and does not makes sense. This is misleading as it implies data assimilation, whereas neutron data is not incorporated, it is simulated by mHM and then compared with observed counts.*

 AR: *We are sorry for this confusion. The reviewer is correct; namely, we simulate neutron data within the mHM model and subsequently compare it with observed counts, rather than incorporating observed neutron data directly. We have revised the relevant sections to accurately reflect this statement.*

>  Simulation of neutron data within the mHM model and subsequently comparing it with observed counts can enhance the accuracy and precision of  soil moisture measurements.

**RC:** *Line 450: 'After optimizing the soil hydraulic properties... 'This is not properly described in the method – is this done after CRNS parameters are optimised or at the same time in some form of iteration? How did you do this? Fig.9 – surely belongs in Results?*

 AR: *We understand that this part was not very clear in the submitted version of the manuscript; and we have made every attempt to clarify it in the revised manuscript. In brief, we have optimized the soil moisture, snow, and neutron counts related parameters simultaneously, as illustrated in the flow diagram. We have provided a clearer description in the methodology section, outlining how this simultaneous parameter optimization is achieved through iterative methods.*
*Regarding Fig.9, we present the result of neutron counts corresponding to optimized parameter sets.*

*We added in the method part more explanation about the CRNS parameters optimization in mHM the number of parameters we took during the calibration process. Fig.9 We only kept the evapotranspiration plot.*

* * *
**RC:** *Referee Comment*,      AR: *Author Response*,      ☐ Manuscript text

**RC:** *In their manuscript "Improved representation of soil moisture simulations through incorporation of cosmic-ray neutron count measurements in a large-scale hydrologic model" the authors Fatima et al. describe the combination of several years of data from CRNS instruments at selected experimental sites and hydrological modeling. The reviewer appreciates the approach of the authors to move towards a more comprehensive picture of how CRNS can help to understand hydrological dynamics however, there are significant concerns about the methodology, data representation, completeness in the representation of CRNS, and lack of transparency in the choice of models and parameters in the paper. The reviewer suggest addressing these issues to improve the quality and validity of the research.*

**AR:** *Thank you for taking the time to voluntarily review our manuscript.*

**General:**

**RC:** *Without any apparent reason or justification this paper selectively uses the soil depths 0-5 cm, 5-25 cm and 25-60 cm. Except for the top soil layer, none of these classes are representative for either hydrological processes or measurement depths of CRNS. What is the reason for the authors to use that scheme? This sampling is too coarse and not deep enough to be acceptable and needs to be refined. This sampling can and will lead to non-obvious systematics and therefore would create misleading results. Specifically, as the authors focus on statistical analysis methods this whole approach seems questionable.*

**AR:** *We understand that our choice of vertical resolution can appear arbitrary for readers that are not experienced with the characteristics of large-scale hydrological modelling and their application in mesoscale settings. While we agree that the vertical resolution could also be critical with respect to capturing small-scale hydrological processes, it has been, however, shown in the past that the chosen depth resolution is reasonably accurate to describe the relevant hydrological processes in the root-zone relevant soil processes for mesoscale hydrological models (Al-Shrafany et al., 2014; Crow et al., 2018; Vereecken et al., 2015; Mohajerani et al., 2021; Downer and Ogden, 2004). Furthermore, the chosen depth resolution and stratification ensure that the most important supporting points for the description of depth-dependent neutron sensitivity are captured as justified by recent literature (Andreasen et al., 2020; Franz et al., 2020; Baatz et al., 2017; Iwema et al., 2017; Andreasen et al., 2016; Han et al., 2015).*

*To reinforce this fact, we have also conducted a set of experiments with varying soil depths, ranging from 3 to 6 layers, specifically (0-5 cm, 5-15 cm, 15-25 cm, 25-30 cm, 30-60 cm, and 60-100 cm). In the model*

*settings, our analysis pointed out that the simulated neutron count results were not significantly influenced by the number of soil layers (see Fig. 1 below for more details). Based on our prior experiences of modelling with mHM, we adopt the same soil layer (0-5 cm, 5-25 cm, and 25-60 cm) depths as utilized in the study by Boeing et al. (2022) - who have extensively evaluated mHM simulations against observed soil moisture across different locations in Germany. The theoretical measurement depths of CRNS are in the range of 10 to 80 cm according to Zreda et al. (2008), Franz et al. (2012), and Köhli et al. (2015). For intermediate soil moisture conditions (0.1 to 0.4 m$^3$/m$^3$), the average CRNS measurement depth is between 15 and 30 cm, but the highest sensitivity is in the upper layers, 0 to 5 cm (Schrön et al., 2017). It is correct that the accuracy of numerical calculations (such as our COSMIC model set-up) would benefit from higher resolved soil profiles, however, our experiments demonstrated that varying soil depths from 3 to 6 layers did not have a substantial impact on the simulated neutron count results in our model setting.*

*Finally, we would like to highlight past modelling studies that successfully used CRNS measurements in their respective model-based investigation where certain modelling choices are inevitable. For example, Iwema et al. (2017) used a land surface model to investigate the impact of reducing the scale mismatch between surface energy flux and soil moisture observations of CRNS point measurement data. The model used with grid cells of 1 km$^2$, parameters calibrated with eddy-covariance flux data and point-scale soil moisture data. In the study they used point-scale soil moisture data from the soil layers up to 30 cm depth and 2012–2015 data were simulated. Other factors limit the limited effect of calibrating soil parameters on soil moisture dynamics and surface energy fluxes spatial-temporal stability. Barbosa et al. (2021) study used HYDRUS-1D model for the simulation of soil moisture time series at the 6 different depths of 2, 20, 30, 40, 60, and 100 cm and compared it to independent soil moisture measurements of PR2. HYDRUS 1D is coupled with the COSMIC Operator and Richards nonlinear partial differential equation was used and compared the neutron count rate with soil moisture simulated data and inversely calibrated the soil hydraulic parameters based on CRNS data. Zhao et al. (2021) used the land surface model (the Community Land Model; CLM version 3.5), a coupled land surface-subsurface model (CLM-ParFlow) is applied, and used the soil layer depth of 5 cm and 20 cm.*

> We analyzed the soil water content data at different soil layers (0–5 cm, 5-25 cm, and 25-60 cm)  in mHM, as utilized in the study by Boeing et al. (2022). The accuracy of numerical calculations (such as Shuttleworth et al. (2013) set up) would benefit from higher resolved soil profiles, however, our experiments demonstrated that varying soil depths from 3 to 6 layers did not have a substantial impact on the simulated neutron count results in mHM.

*We added Figure 1 from this very response as a new Figure S2 in the supplement.*

[Figure]

Figure 1: (Added to supplements as Figure S2) Simulated neutron counts from mHM with different soil horizon depths: three layers (5, 25, 60 cm), five layers (5, 15, 25, 40, 60 cm), and six layers (5, 15, 25, 40, 60, 100 cm).

**RC:** *Using just the $N_0$ method and the COSMIC operator in order to draw the general conclusions the authors would like to present is with respect to the state of the art not justified. Either narrow down the scope to a more exemplary analysis or include other models like the UCF method from Franz et al. or the UTS method from Köhli et al., which are both not even mentioned in the overview, yet mentioned in the discussion.*

**AR:** *Thank you for bringing up t at the other N-SM conversion functions, we have made it clearer that this is an exemplary study and other methods could be tested in future research. Our choice for COSMIC and Desilets was based on their widespread use in past/recent studies: e.g., the Desilets method (Desilets et al., 2010) used by Baroni and Oswald (2015); Schreiner-McGraw et al. (2016); Zreda (2016); Avery et al. (2016);*

*Vather et al. (2019, 2020); González-Sanchis et al. (2020); Power et al. (2021); Barbosa et al. (2021); Chen et al. (2022), and the COSMIC Method (Shuttleworth et al., 2013; Rosolem et al., 2014) used by Baatz et al. (2014); Iwema et al. (2015); Han et al. (2015, 2016); Brunetti et al. (2019); Mwangi et al. (2020); Patil et al. (2021). The UCF method from Franz et al. (2013) has its own limitation as has been shown low experimental performance in the past (McJannet et al. 2014, Baatz et al. 2014). The recent UTS method from Köhli et al. (2021) was published after the start of our study, it still requires sound experimental validation, and it adds further complexity (such as air humidity dependency) which is not feasible for mHM at this state. In fact, the cited paper shows that the UTS function behaves very similar to the COSMIC approach. Hence, we believe that we have covered the main processes of the N-SM conversion approaches. Nevertheless, we have mentioned the recent developments of N-SM conversion functions in the revised manuscript as an opportunity for future elaborated studies.*

> The COSMIC method  enables a comprehensive representation of the neutron  generation process, which is computationally more demanding than using  an analytical formulation (e.g., Desilets et al., 2010; Köhli et al., 2021).

> The simulated time series tended to slightly underestimate the CRNS neutron count rate, particularly during the dry season. This effect could be explained by the known limitations of the equations under very dry conditions, while recent approaches exist (Köhli et al., 2021) that could lead to further improvement in future studies.

**RC:** *The study heavily focuses on modeling and statistical analysis. Throughout the manuscript, the reader encounters a large amount of seemingly arbitrary choices, which in the end provides the impression that results are selected and tinkered in order provide a realistic picture. The authors want to show that they uses methods which are accepted in the community, from the statistical and the modeling perspective, then, however, they introduce a significant amount of modifications and ad hoc assumptions which may put the whole approach into question. Examples are: In which way should a sophisticated hydrological model help, if the authors choose to simply take only three layers, for which the choice of hydrological parameters is also not really transparent? What is the reason to take only the 1% of the model runs with the best KGE (that number clearly depends on the initial parameter ranges the authors arbitrarily chose)? As the choice of the neutron model and their parametrization is not at all according to any standard, what does such tight constraint say other than the whole analysis is tuned to fit one specific ad hoc assumption. In which way is the KGE modified by the authors introducing a bias on this analysis? Why are 99 % of the model runs excluded if there are significant deviations from the models and the data, even visible in the plots presented? To be clear on that point: If significant deviations can be observed between model and data, any tight statistical constraint (in matching them incongruently) will lead inevitably to wrong results.*

**AR:** *As stated above, we have analysed different soil depths ranging from 3 to 6 layers (0-5 cm, 5-15 cm, 15-25 cm, 25-30 cm, 30-60 cm, and 60-100 cm) but found that the number of layers did not significantly affect neutron count results. As a result, we adopted the same layer depths used in Boeing et al. (2022).*
*We selected 1% of the model run with the best $KGE_{\alpha,\beta}$ because this simulation yielded neutron counts that captured well towards observed values. Increasing the number of simulations is certainly possible but this could introduce another level of complexity.*
*Regarding the revision of KGE, Gupta et al.2009 proposed the KGE as a weighted combination of the three components (bias, variability, and correlation terms). We opted not to consider it in our assessment*

*(objective function), as it accounted for 33% of the total weighting in the overall KGE score. High correlation values stem from the fact that the seasonality of SM is an inherent characteristic in the northern hemisphere, where precipitation minus evaporation is mostly driven by evapotranspiration. Evapotranspiration is higher in summer and lower in winter, and soil moisture indirectly will result from low ET values in winter and high values in summer. This aspect is inherent, and we do not want to emphasis in our optimization these characteristics. That does not mean we are neglecting seasonality, but do not want our parameters to capture the seasonality aspects. Even if a random parameter is selected, the correlation will always be higher because the meteorological forcing is the precipitation - evaporation is seasonal, and seasonal is coming from forcing. Previous studies have also introduced revisions to KGE (Mizukami et al., 2019).*

*The COSMIC model is the established standard for Cosmic-Ray Neutron Sensor (CRNS) forward modeling. There are several studies in the literature that have utilized COSMIC as the recognized model for CRNS simulations, such as Baatz et al. (2014), Iwema et al. (2015), Han et al. (2015), Han et al. (2016), Brunetti et al. (2019), Mwangi et al. (2020), and Patil et al. (2021).*
* * *
In this study, we use the general concept of the Kling-Gupta efficiency KGE for designing the objective function, which is widely employed in hydrological modeling to assess the efficiency of a model (Gupta et al., 2009). We use the general concept of the  KGE  as a weighted combination of the three components (bias, variability, and correlation terms) to evaluate our simulation (Gupta et al., 2009). We excluded the correlation component from (Eq. 13)  as our simulation already exhibited satisfactory correlation due to strong seasonality, we opted not to consider it in our assessment (objective function), as it accounted for 33% of the total weighting in the overall KGE score. Seasonality is an inherent characteristic in the northern hemisphere where precipitation minus evaporation is mostly driven by evapotranspiration. Even if a random parameter is selected correlation will always be higher because the meteorological forcing is the precipitation - evaporation is seasonal.
* * *
**RC:** *Instead of overloading the manuscript with a multitude of different statistical measures, the authors should focus on providing a reasonable basis for comparing model and data. The authors rather present in the manuscript their own struggles and the reader does not learn anything from that way of analyzing a problem the authors fabricated in an intransparent way.*

AR: *We are using the five most established measures in hydrology to evaluate the model performance on observations (KGE, modified KGE, NSE, $R$, and PBIAS). As it is well known from the literature Nash and Sutcliffe (1970); Gupta et al. (2009), each of these measures has its own justification to assess the quality of the performance in terms of bias, dynamics, temporal and static errors, etc. The fact that our results are not only based on a single measure but rather in agreement across the many different measures shows that our conclusions are particularly robust (see Table 4). We believe that this procedure strengthens our study and demonstrates that neutron counts could improve hydrological model results not only in terms of correlation but also in reducing biases and improving overall performance measures.*

*We have already included the best parameters and the performance of the parameter sets of the study site in the supplementary material Table S2. Additionally, we have revised both the flow diagram and the model setup to enhance clarity in the revised manuscript.*

**Specific commments:**

**RC:** *The neutron data is plotted as quite large dots, which scatter significantly. Either smaller dots should be chosen or some type of smoothing.*

AR: *Thank you for your suggestion. We have considered using smaller dots in the neutron data plots to improve their clarity and readability. We have made changes to the neutron data plots.*

*We have updated the figure in the revised manuscript.*

[Figure]

Figure 2: (Updated in the revised manuscript) Simulated daily time series of $N_{\text{COSMIC}}$ for the four sites. The black lines represent the median of the behavioural simulation ensembles that satisfy the objective function which is LHS10 ensemble members. The light grey shaded areas represent the 95% CI of the simulation ensembles corresponding to different levels of constraining which is LHS1000 ensemble members, and the observation is shown in grey points.

**RC:** *Line l88: "the physics-based model COSMIC" - COSMIC is not physics-based, it is not based on a comprehensive physical interaction picture. It selectively takes specific processes and invents arbitrary mathematical representations for them.*

AR: *As was demonstrated in a large number of accepted literature, COSMIC is an analytical-based model incorporating key physics-based processes important for CRNS applications in conjunction with models.*

*COSMIC simplifies the physical process tailored to CRNS applications by mimicking a similar neutron transport behavior at the vertical axis as compared to more complex Monte Carlo models that actually implemented detailed physical interaction processes in three dimensions Shuttleworth et al. (2013); Rosolem et al. (2014). The model includes descriptions of various physical processes, such as neutron flux degradation with soil depth, creation of fast neutrons within the soil, and scattering of fast neutrons, all depending on soil composition and water content. It has been validated with detailed physics-based simulations and has been empirically confirmed in many subsequent studies i.e., Baatz et al. (2014), Iwema et al. (2015), Han et al. (2015), Han et al. (2016), Brunetti et al. (2019), Mwangi et al. (2020), Patil et al. (2021)). Naturally, not all detailed physical processes can be represented by a mathematical model, particularly the 3D neutron transport at scales of 1 to 300 meters. However, at the scale of interest in hydrological and land surface modeling (1 km), COSMIC can be considered adequate. In this study, we rely on the published model structure, while any model improvement with regard to additional physical processes will have to be discussed in a dedicated separate paper.*

**RC:** *Lines 99+: "What is the best approach to simulate CRNS neutron counts in a hydrological model considering the heterogeneity of vertical soil moisture profiles?" - This paper provides an exemplary data analysis which is insufficient for generalizations of the mentioned type.*

AR: *Thank you for your comment. In our study, we checked the capability of a mesoscale hydrological model that can simulate the neutron counts by taking into account the vertical heterogeneity of different soil layers and comparing them with the measured neutron counts based on hector scale footprint. To account for sub-grid heterogeneity, which is captured with the multiscale parameter regionalization (MPR) technique in mHM (Samaniego et al., 2010), it allows us to study how model parameters vary across different scales of modeling.*

*We evaluated three methods, namely Desilets' uniform and non-uniform weighting schemes, and the COSMIC method in mHM. Based on objective function, we present the results of the most effective method for each study site in Table 4.*

**RC:** *Lines 159+: "Simulations from mHM revealed that the sensitivity to the highest soil water content was observed at 5 cm depth (...)" - this sentence is highly confusing, grammatically and in the context of the manuscript as for example there is no information provided anywhere about a layer specifically at 5 cm depth.*

AR: *Thank you. We have revised the sentence.*

> We analyzed the soil water content data at different soil layers (0–5 cm, 5-25 cm, and 25-60 cm)  in mHM, as utilized in the study by Boeing et al. (2022).

**RC:** *Lines 165+: Theoretically, the $N_0$ parameter, which represents the neutron count rate level of the particular CRNS probe used for rather dry soil at the local conditions, should be site-specific" - please describe theories which underline the theoretical reasoning that $N_0$ is site specific. The mentioned references do not provide that information. In case the $N_0$ equation is an inadequate representation of the neutron count rate a site-specific behavior would be a result.*

AR: *Thank you for this remark. The site-specific nature of the $N_0$ parameter is a well-recognized aspect within the Cosmic-Ray Neutron Sensor (CRNS) community, though there may be other future approaches for $N_0$*

*accounting directly for various site-specific influences. Currently, $N_0$ is typically calibrated for each sensor at each location via in-situ soil sampling campaigns or local soil moisture networks. Many authors have confirmed this observation (Zreda et al. 2012, Hawdon et al. 2014), while others investigated data from several different sensors and found non-identical $N_0$ values even if the detectors are similar (Fersch et al. 2019, Heistermann et al. 2021, Bogena et al. 2022). This indicates that the influencing factors on $N_0$ are not yet fully understood though a recent study has outlined that a minimal info from the site could be sufficient to determine the $N_0$ even without local soil moisture data (Heistermann et al., HESSD, 2023, https://doi.org/10.5194/hess-2023-169). Since the mHM model does not include these effects, we infer the value of $N_0$ through the calibration procedure.*

**RC:** ***Lines 172: "may be impacted by factors such as soil chemistry" - within the field of CRNS researchers claim that this method would be independent of soil chemistry. The reviewer is curious how the authors come to this assumption.***

AR: *Regarding CRNS data, the absolute neutron intensity varies by location due to varying soil chemical compositions, although these variations are generally small for epithermal and fast neutrons (ranging from 1 eV to $10^6$ eV) because neutron absorption is insignificant. Thermal neutrons have a different sensitivity to varying soil moisture content than do fast and epithermal neutrons because thermal neutrons are highly dependent on on chemical composition of the soil matrix and soil water (Zreda et al. 2008, Andreasen et al. 2019, Rasche et al. 2021). This effect often is negligible, but we still mention it here since the variation of soil chemistry within our four sites is significant, particularly due to a high range of organic input below, on, and above the surface. We have expressed this aspect by changing the formulation to "additional hydrogen pools (e.g., from organic material)".*

> This coefficient is specific to the particular CRNS detector and may be impacted by factors such as  terrain (topography) but also local soil, vegetation characteristics, and additional hydrogen pools (e.g., from organic material) at each observation site Schrön et al. (2021).

**RC:** ***Lines 172: "heterogeneity" - which type of 'heterogeneity' do the authors refer to? In case 'heterogeneity' refers to topographical heterogeneity the whole approach of this analysis is questionable.***

AR: *Thank you for pointing out this part. We would like to clarify the type of 'heterogeneity' we refer here is related to not only terrain (topography) but also local soil and vegetation characteristics. We have revised the sentence to better capture this aspect e.g, through "... neutrons are sensitive to all kinds of hydrogen in the footprint, hence the variable $\theta$ denotes not only soil moisture, $\theta_{sm}$, but is rather assumed to also include lattice water, $\theta_{lw}$, as well as water equivalent from soil organic carbon, $\theta_{org}$, and vegetation biomass, $\theta_{bio}$."*

*We changed the sentence structure:*

> This coefficient is specific to the particular CRNS detector and may be impacted by factors such as  terrain (topography) but also local soil, vegetation characteristics, and additional hydrogen pools (e.g., from organic material) at each observation site Schrön et al. (2021).

**RC:** ***Line 182: "derived from neutron particle physics modeling" - in which way are these parameters derived from particle physics and empirical (as mentioned above) at the same time?***

AR: *Thank you for pointing this out. In the revised manuscript, we have formulated the sentence as follows: "parameters $a_i$ were determined empirically by Desilets et al. (2010) who derived $a_0 = 0.0808$, $a_1 = 0.372$, and $a_2 = 0.115$ for values of $\theta > 0.02\ gg^{-1}$."*

*We formulated the sentence:*

> Among the four parameters, three of which are coefficients (parameters $a_i$ were determined empirically by (Desilets et al., 2010) who derived $a_0 = 0.0808$, $a_1 = 0.372$, and $a_2 = 0.115$) derived from neutron particle physics modeling (Zreda et al., 2008; Desilets et al., 2010), and are considered as constants for values of $\theta > 0.02\ gg^{-1}$.

**RC:** ***Line 200: The weighting scheme as presented is incomplete. (5)-(7) only take into account the weight of one depth. As to the model the authors used, the weight needs to be calculated by an integral over the depth weighting function for the height of the soil profile, not just the weighting function by itself.***

AR: *Thank you for your valuable feedback. We implemented the weighting procedure properly but have not adequately explained the procedure in the manuscript. This will be fixed in the revision. In the weighted-averaging approach, the weights are determined following Schrön et al. (2017), where the vertical contribution of layer $i$ is:*

> In the weighted-averaging approach, the weights are determined based on Schrön et al. (2017):
>
> $$N_{\text{Des,W}} = N_{\text{Des}}\left(w_i = w\theta_{\text{avg}}(z_i w, \theta)\right),\tag{5}$$
>
> $$\text{where}\quad w(z)_i = \int_{z_{i,\min}}^{z_{i,\max}} e^{-2z/D - 2\,z/D}\,dz \propto e^{-2\,z_{i,\min}/D} - e^{-2\,z_{i,\max}/D}\tag{6}$$
>
> $$\text{and}\quad D = \varrho_b^{-1}\left(p_0 + p_1\left(p_2 + e^{-p_3\,r}\right)\frac{p_4 + \theta}{p_5 + \theta}\right).\tag{7}$$
>
> Here, the integral goes through each horizon from $z_{i,\min}$ to $z_{i,\max}$ in 1 mm steps and sums up the weight over the whole layer. $z_i$ is the depth of the given soil moisture layer $i$, $D$ is the average vertical footprint depth of the neutrons, $p_i$ are numerical parameters presented in Schrön et al. (2017), and $r$ (m) represents the distance from the sensor.

*Here, the integral goes through each horizon from $z_{i,min}$ to $z_{i,max}$ in 1 mm steps and summes up the weight over the whole layer. The fact that the integral over an exponential function is again an exponential function is the reason for our simplified description using only $z_i$ in the original manuscript. Updated the equation as mentioned above in the revised manuscript.*

**RC:** ***Line 203: What is the depth $z_i$? As the authors use soil horizons of considerable height, how do the authors calculate $z_i$?***

AR: *Thank you for your remarks. To calculate the influence of soil moisture in different depths, the soil horizons were subdevided into smaller fractions and integrated over smaller steps of $z_i$. We have clarified this in the manuscript in the revised manuscript. We added the formula:*

*(See changes in the previous answer)*

**RC:** *Lines 210+: The COSMIC model only mimicks the mentioned subset of physical processes, in no way it represents them. Analytically COSMIC only represents an exponential N(theta) function, with an arbitrary parameter adaptation (8). The underlying mentioned physical processes are not responsible for the signal generation within CRNS as it lacks the spatial neutron transport, which CRNS claims to use as a unique feature compared to other methods.*

**AR:** *Thank you for your comment. According to a decent number of published and widely accepted literature, COSMIC is able to adequately mimic the neutron generation from soil moisture profiles Shuttleworth et al. (2013); Rosolem et al. (2014) Baatz et al. (2014), Iwema et al. (2015), Han et al. (2015), Han et al. (2016), Brunetti et al. (2019), Mwangi et al. (2020), Patil et al. (2021) ). Of course, a mathematical model can never represent the full range of physical processes involved, however, as the authors from Shuttleworth et al. argue, it aims at representing the main processes in the vertical dimension (such as signal attenuation and neutron production/evaporation based on soil composition). This dimension is most relevant for large-scale models (500–4000 m). Spatial neutron transport at typical scales of 1–300 m might be necessary for detailed and complex terrain, as was shown by, e.g. Schattan et al., (2019), Schrön et al. (2023), and Köhli et al. (2023). But on average the typical mHM pixel size is at least one order of magnitude larger than the scale of lateral neutron transport, while the chosen sites exhibit homogeneous land use. Hence, we consider the 1D model assumptions adequate for the target application.*

**RC:** *(11) is missing*

**AR:** *Thank you for pointing this out. This is a layout issue and will be resolved in the revision. We addressed the layout issue by ensuring the correct sequencing of equation numbers.*

**RC:** *Line 241: explain the term "geometric integral"*

**AR:** *We are here referring to equation 8 as mentioned in the text. As was explained, it resembles an integral of the vertical neutron transport, geometrically projected to the vertical axis. See also Schuttleworth et al. (2013).*

**RC:** *Line 275: What are "COSMOS models"?*

**AR:** *We have corrected it to "COSMIC model". We have corrected the spelling to 'COSMIC' in the final manuscript.*

**RC:** *Line 293: "COSMIC is physically based, a loss of the physical meaning of the parameters in question would be very critical." - as described above COSMIC takes an incomplete subset of physical processes in order to justify its model. In this sentence the authors echo the critics which have been mentioned with respect to this model. Without the representation of neutron transport, for example, any possible source-only model can hardly justify itself to be correct.*

**AR:** *As stated above and as has been demonstrated in plenty of respective literature, the COSMIC model represents the vertical neutron transport in a 1D soil column in a physically consistent manner. Based on several confirming studies in the literature (see above), we consider it as a reasonable approach to be implemented within mesoscale hydrological models, in which horizontal resolution is orders of magnitude larger than the scale of lateral neutron transport. For detailed small-scale processes, snowpack quantification, and heterogeneous terrain, we would agree that COSMIC could be improved, but this is out of the scope of this study and probably even unnecessary given the large spatial scales on which typical hydrological and land-surface models operate. During the revision of the manuscript, we have reflected on these parts to make them clearer.*

**RC:** *Fig. 4: Why is the depiction of the histograms so coarse? Please enlarge the scale to the relevant range.*

**AR:** *The plots already show the full range of the 100 000 data for the prior range and 10 data points taken from*

*the objective function which is the posterior range in the x-axis. The main message of the plots is to show how the posterior distribution was constrained compared to the respective prior distribution. That is why we plot both distributions along with their full ranges.*

**RC:** *Fig. 4: As many columns have the same height, the reviewer questions the representativeness of the results. The exact same height could mean that either the model provides on the basis of the 100 000 data sets the same values or most of the results were discarded and the authors want to draw conclusions from just a few values.*

AR: *The reviewer seems to misinterpret the Figure and we would be happy to sharpen the corresponding explanations in the revised manuscript. The sample size for all sites is fixed at N = 100 000 as the prior range, and the iterations are conducted within the $N_0$ range of 600-1500 along with another 28 parameters for Desilets method and 30 parameters for the COSMIC method for each site. Consequently, when utilizing the prior sample range, it appears identical for all sites due to this uniform setup.*

> Supplementary Table S1 shows the values for the parameters in all 100 000 simulations and the selected 29 parameters for the Desilets method and 31 parameters for the COSMIC method, which include snow, soil moisture, and neutrons modules as behavioral simulations, with the posterior mean of the top 10 best parameters set. For further information and additional details about the calibrated parameters for each site, refer to Supplementary Table S2. Among the calibrated parameters, the $N_0$ parameters are different in each method since this parameter  does not exactly have the same physical meaning in the Desilets and the COSMIC methods.  Fig. 4  displays the x-axis in gray, representing the original parameter range (600–1500) prior distribution for the Desilets method and (100–400) for COSMIC method. Meanwhile, the colored sections in brown, green, and purple indicate the parameter values of the calibration from the posterior distribution taken from the top-performing parameter sets for each study site.

**RC:** *Line 323: Given the fact that the authors chose to represent the soil in very coarse layers, the "crowding cows" in the otherwise very wet catchment, seem to be a distracting and out of scope reasoning by the authors and at that point do not strengthen the scientific quality of the material presented.*

AR: *Evidence for the influence of crowding cows at this site has been mentioned in Schrön et al. (2017) and may introduce additional uncertainty to the data in specific periods (Aug to Sept). We agree that this is not a major issue particularly since there are no cows for the major part of the year. We further demonstrated that the choice of vertical layers is adequate, as explained above, and have added the explanation in the discussion part of the revised manuscript.*

One of the primary sources of uncertainty at the *Grosses Bruch* site is surface ponding and shallow groundwater, as well as the loamy texture of the soil. Those factors contribute to the formation of permanent water ponds in the area and may introduce uniform or even inverse soil moisture profiles which directly influence the neutron emissions, but cannot be captured by the mHM model. Another factor is the time-variable effect of crowding cows near the station, which may influence the CRNS signal, but is challenging to correct in the CRNS measurement (Schrön et al., 2017). We incorporate the CRNS parameter set in mHM, and some parameters related to soil moisture and neutron counts are effectively constrained based on the objective function using $KGE_{\alpha\beta}$. However, there is still room for improvement, particularly with regard to the coefficient in root fractions distributed across soil layers.

**RC:** *Fig. 5 and Fig. 6.: Both are plotted in such a tiny way, that it is hard to identify the different lines from each other and blur the relevant deviations.*

**AR:** *Thank you for the feedback, we have replotted the data to improve visibility and make it easier to distinguish the lines and relevant deviations. To distinguish between observed data and model simulations, smaller dots are used for the observed data points.*

**RC:** *Lines 379+: citep missing. "which are typically 5–60 cm and sensitive to shallow soil moisture" the reference tells that CRNS actually measures deeper than that and for that reason the choice of only simulating up to 60 cm seems wrong or incomplete.*

**AR:** *Thank you for your feedback. We have added the citation with the citep command. Our calculation was based on the formulas given in the citation and specific to the investigated sites. In the revised manuscript, we have mentioned that the theoretical measurement depth for the cosmic-ray probe varies, ranging from 12 cm in wet soils to 76 cm in dry soils (Zreda et al., 2008, 2012; Rosolem et al., 2013). For the sites in our study, we typically do not see measurement depths beyond 60 cm (Bogena et a. 2022).*

*We added the citation and restricted the sentence:*

This study assessed the suitability of CRNS observations at four sites to enhance soil moisture representation in mHM.  The theoretical measurement depth for the cosmic-ray probe varies, ranging from $\sim$ 12 cm in wet soils to $\sim$ 76 cm in dry soils (Zreda et al., 2008, 2012; Rosolem et al., 2014).

**RC:** *Line 445: "Overall, the three methods (NDes,U, NDes,W, and NCOSMIC) in mHM were able to consistently simulate the neutron count" - the results were inconsistent and the variability was only partially covered.*

**AR:** *Thank you for your comment. In the paragraph, the statement is that "Overall, the three methods ($N_{Des,U}$, $N_{Des,W}$, and $N_{COSMIC}$) in mHM were able to consistently simulate the neutron count variability throughout the available data period." Here the consistency in simulating neutron count variability means that mHM has the capability of capturing the general trend and pattern of the simulated data, as illustrated in (Figs. 5 and 6). The light grey color shows the top 1% of the model run with the best $KGE_{\alpha\beta}$ values. Furthermore, we*

*have provided a comprehensive discussion of the performance and variability of neutron count simulations with the three methods at each site, from lines 405-455.*

---

## Referee Report (RR1)

I am still troubled by a potential flaw in this manuscript. As I understand the work, the link between true field soil moisture and modelled comparisons has been lost, although Fig. S5 may show that this is not the case (do Fig. 5 measurements use the fitted $N_{0\_Des}$ or a field calibrated $N_0$?).

Note although this paper is about improving soil moisture (process) representation in models (the title), there is only one plot showing calibrated modelled soil moisture, and this is in the Supplementary. In the main paper, only neutron count comparisons are shown. Since both the relationship between neutron counts and soil moisture and the predicted soil moisture are both calibrated, it would seem to me that there is no longer necessarily a representation of true field soil moisture. It is noted that the authors (in previous reply) have chosen to focus on neutron count agreement, rather than soil moisture agreement; however, I would urge the authors to consider how this method can still be traced back to the absolute soil moisture measurement.

To elaborate on this point: as the authors describe, the CRNS method for field soil moisture (SM) measurement has a free calibration parameter $N_0$, or more specifically here $N_{0\_Des}$. Through careful field calibration, normally by collection and moisture analysis of field soil samples, the value of $N_{0\_Des}$ is determined. This provides the crucial link or traceability of the measured neutron counts to soil moisture content – the quantity which hydrologists are actually interested in knowing.

However, in this paper, the field calibration value of $N_{0\_Des}$ is not used. Instead as part of the model calibration period (line 292) the $N_{0\_Des}$ is optimised – presumably by minimising the neutron count rate or soil moisture error of the model (it is not stated what objective function was used). This model calibrated $N_{0\_Des}$ will be different to the field calibration, giving different soil moisture content for a given neutron count – thus the true site-specific calibration of neutron counts to soil moisture has been lost. Whilst the calibrated model may have better agreement with the observed neutron counts, the model output calibrated soil moisture does not necessarily have a similar improvement i.e. the soil moisture could be biased high or low, and that bias accounted for in terms of neutron counts by the model calibrated $N_{0\_Des}$.

Seeing Fig. S5, I actually do not think this flaw really exists – but the detail of the $N_0$ model calibration versus soil moisture calibration needs to be clearer to explain how this potential issue has been dealt with. The authors should justify their approach of a model calibrated $N_0$ versus using the value already known from site specific field calibration of the CRNS. And it may be of value to compare these.

Specifically:

Line 303 "Estimated values of $N_{0\_Des}$ and $N_{0\_COSMIC}$ obtained in our study are close to optimal values" – how do you know that? What are the optimal values? And the inference drawn is not sound – model simulation of dry conditions is not a prerequisite to obtaining accurate $N_0$ values.

Results – Fig. 5 Also show plots of SWC (as per field calibration – observations) and calibrated modelled SWC.

Discussion – I would question the soundness of discussing model performance, when it appears that neutron count comparisons rather than SWC have been calibrated. As the authors have chosen to present neutron count data, then they need to be careful as to what is claimed with regard to soil moisture modelling, or to provide evidence to support those claims.

Line 440 " ...improved not only soil moisture estimation" – NO improvement in soil moisture estimation is shown in the main paper!

Conclusion – Line 507 ...evaluation with soil moisture observations has not been shown.

Several statements in the conclusion are not supported by the paper (at least not without digging into Supplementary material) e.g. Line 525 "improved the soil moisture performance of the model"

---

## Author Response (AR2)

**Author Response to reviewer comment #1**

**Improved representation of soil moisture processes through incorporation of cosmic-ray neutron count measurements in a large-scale hydrologic model**

Fatima et al.
*Hydrol. Earth Syst. Sc.,* `doi:egusphere-2023-1548`
* * *
**RC:** *Referee Comment*,     AR: *Author Response*,     ☐ Manuscript text

**RC:**1. *The reviewer thanks the authors for addressing many of the points raised during the review. There are, however, still some open points which require to be addressed. The authors' reliance on citation of literature appears to be selective, raising concerns that they may be attempting to back up their arguments through simple reference rather than substantive engagement with the cited works. This practice, characterized by citation on a keyword basis without thorough consideration of contextual nuances, risks oversimplifying complex scientific concepts and extrapolating findings beyond their appropriate scope, which may impact the validity of their arguments. Consequently, this approach undermines the integrity of the scientific discourse by potentially perpetuating misconceptions and failing to address or acknowledge inherent limitations or uncertainties within the literature. An illustrative example of this tendency is evident in their persistence regarding the potential influence of speculative factors such as cow activity on model offsets, without adequately substantiating these claims or conducting investigations to validate such hypotheses. Additionally, their insistence on employing a variety of statistical performance measures to evaluate a systematically offset model appears to be a misguided attempt to obfuscate inherent shortcomings rather than directly addressing or mitigating them. This approach not only distracts from the clarity and focus of their analysis but also raises doubts about the overall approach of the model and the transparency of the authors in reporting their findings.*

AR: *Thank you for taking the time to review our manuscript.*

**Detailed comments to the reply of the authors:**

**RC:**2. *"however, our experiments demonstrated that varying soil depths from 3 to 6 layers did not have a substantial impact on the simulated neutron count results in our model setting" (...) The reviewer had raised suspicions about the coarse layer structure as hydrologically the topsoil dynamics happens on a much smaller scale and that could have implications on the CRNS signal. The reviewer suspected that there is a scale mismatch between a layering that might be appropriate for hydrological purposes and CRNS which is sensitive to dynamics on a smaller scale. To be precise: In the way the authors have structured their simulations, they reduce most of the signal dynamics measured by the CRP to two layers. The question of the reviewer may have been misleading as the referring of the authors to literature which used a similar layering for other reasons it would is not directly in favor of their argument. The additional material which is presented by the authors does not support the challenged conclusion. They show, that there are significant deviations for the simulations using 3, 5 or 6 layers. As the residuals are better in the three-layer case, one can assume that the authors have chosen the more coarse representation in order to yield better results. Typically, a more fine layered representation would lead to converging results.*

*In case significant deviations or alternating fit qualities, it hints that there are further systematics with respect to that model parameter. As in the original manuscript of the authors the vertical weighting function was presented incorrectly, this was reason enough for the reviewer to suspect a systematic error based on the choice of layers. However, the only conclusion which can be drawn from the material the authors present here is, that the results are (still) biased by the choice of layers. In case the authors chose to model their system less granularly to yield a better fit quality, the unknown system bias might yield systematically wrong representations. Given that using fewer soil layers reduces specifically the residuals in situations where the deviations are surprisingly large, the assumption is that the authors might simply introduce new errors to compensate other model errors. It could be the case that the authors have - involuntarily as due to their arguments - chosen a for CRNS representative reduction of the layering scheme. That, however, would need to be analyzed separately.*

AR: *The mHM is a large mesoscale hydrological model and we are showing the compatibility of how can we account for neutron count measurements in such a model while respecting the relevant processes and scales represented by it. We are aware that real world is continuous, but for a hydrological model's conceptualization we need to divide the soil moisture profile (i.e. models' subsurface) into layers to simplify the complex equations governing water flow. This facilitates computationally efficient operation of the model on relevant scales. Moreover, the available soil datasets are very coarse and do not allow for more detailed vertical resolution.*

*Hence, we chose the layering based on widely accepted experience, given the available datasets representing soil properties. mHM setups prescribe the soil layering according to the soil dataset available. In global applications we use Soilgrids v2 which prescribe layers as follows: 0–5 cm, 5–15 cm, 15–30 cm, 30–60 cm, 60–100 cm, and 100–200 cm* `https://soil.copernicus.org/articles/7/217/2021/`. *In Germany, we use a much accurate soil map provided by the Federal Institute for Geosciences and Natural Resources (BGR, 2020). The dataset (BUEK 200; BGR, 2020 at available at a resolution of 1:250 000) was discretized at $100 \times 100$ $m^2$ grid cells with varying soil horizons according to corresponding soil ID. The soil horizons in BUEK 200 vary from 1 to 7. For these reasons, for the German Drought Monitor at $1 \times 1$ $km^2$ resolution, we upscale the soil parameters with MPR (Samaniego et al., 2010) and homogenize the soil layering based on end-user's feedback, namely 0–25 cm, and 25–60 cm, and 0–60 cm depths Boeing et al. (2022). The soil layering of mHM is similar to other land surface / hydrological models.*

*As we showed in our previous response in different layering we don't have a high sensitivity of our results.* **Probably we were not clear enough as we showed in the last response that there was no significant deviation in neutron counts simulation for different selections of layering in this study.** *There will be certain degree of dependency of the results depending on this selection but this is inhabitable the focus of this study not about layering. The choice of layering in this study was also made by the availability of the soil datasets, in this case we used (BGR, 2020) which is a global dataset that is not detailed enough to allow for finer vertical resolution.*

*Following the new GDM setup presented by Boeing et al. (2022), we adopted the upscaled soil layering because the present study contributes to the improvement of this system with novel soil monitoring sensors more appropriate to the resolution of the model.*

*We acknowledge that the formula for the vertical weighting function presented in the original manuscript was incorrect. However, we have verified that the corresponding Fortran code in the mHM model is accurate.*

RC:3. ***"Thank you for bringing up t at the other N-SM conversion functions (...)" Thank you for providing additional context regarding the selection of N-SM conversion functions in your study. However, the reviewer would like to address a couple of points: The statement regarding the UCF method from Franz et al. (2013) having low experimental performance in the past, as demonstrated by McJannet et al. (2014) and Baatz et al. (2014), may not accurately represent the broader literature where similar issues have been raised for the $N_0$ method as well. Therefore,***

*it would be more appropriate to acknowledge the mixed findings in the literature rather than categorically dismissing the UCF method based on selected studies. Regarding the recent UTS method from Köhli et al. (2021), it is important to recognize that while it may require further validation, it still represents a noteworthy advancement in the field of N-SM conversion functions. Dismissing it solely on the basis of its publication date and perceived complexity may overlook potential benefits it could offer, especially if it proves to outperform existing methods in certain scenarios. Overall, while the reviewer appreciates the thorough explanation provided for the choice of COSMIC and Desilets methods, it is essential to maintain a balanced and nuanced perspective on the various N-SM conversion approaches available in the literature.*

AR: *We appreciate your comments on the N-SM conversion functions from the literature. However, we have selected the Desilets method and the COSMIC method for specific reasons. Both methods require information from the soil profiles, which is readily available in the mHM model. In contrast, the Universal Transport Solution (UTS) function couples soil moisture with air humidity in a non-separable way, while no atmospheric information about air humidity is available in the distributed hydrological model mHM. Same holds for the UCF function, which additionally requires a number of parameters that relate to hydrogen pools that are not represented by mHM. We agree that these methods together with additional model parameters would have the potential to improve the results of this study, but the implementation of these additional components into mHM is far beyond the scope of our study.*

*We have added this discussion to the manuscript.*

> Previous studies, such as McJannet et al. (2014) or Baatz et al. (2014) , have noted low experimental performance for the Universal Calibration Function (UCF) method described by Franz et al. (2013). However, we have selected the Desilets method, known as the $N_0$ method, and the COSMIC method for specific reasons. Both methods require information from soil profiles, which is readily available in the mHM model. In contrast, the Universal Transport Solution (UTS) function couples soil moisture with air humidity in a non-separable way, while no atmospheric information about air humidity is available in the distributed hydrological model mHM. The same holds for the UCF function, which additionally requires a number of parameters related to hydrogen pools not represented by mHM.

RC:4. *"We are using the five most established measures in hydrology to evaluate the model performance (...)" Thank you for providing insight into your approach to model evaluation and the rationale behind using multiple statistical measures. However, it is important to note that while employing a variety of evaluation metrics can provide a more comprehensive assessment of model performance, it does not inherently address the systematic uncertainties introduced by modeling choices. The use of various performance measures may indeed offer a more nuanced understanding of model behavior, capturing different aspects such as bias, dynamics, and temporal errors. However, it is essential to recognize that these measures are still influenced by the underlying assumptions and parameterizations of the hydrological model itself. As such, simply presenting results that are consistent across multiple measures does not necessarily guarantee robustness in the face of the evident model uncertainties. In addressing the concerns raised by the reviewer, it would be beneficial for the authors to provide a more transparent discussion of the modeling assumptions, limitations, and potential sources of uncertainty. This would help contextualize the interpretation of the evaluation results and provide a clearer understanding of the model's performance and its implications for CRNS measurement in combination with hydrological modeling.*

AR: *We agree that from the model performance matrices the uncertainty cannot be reduced, but we can check the model performance. The source of uncertainty can arise from model parameterization, specifically the representation of soil moisture in the mHM model across different sites. For further*

*details on parameterization, the Supplement provides extensive information on both prior and posterior solutions of each parameter for each site. The mHM currently lacks a vegetation dynamics module and that need to be improved.*

*Uncertainties related to model simulation are discussed in the manuscript in section 3.2. These topics are also discussed in the conclusion section of the manuscript.*

RC:5. **"As was demonstrated in a large number of accepted literature, COSMIC is an analytical-based model incorporating key physics-based processes important for CRNS applications in conjunction with models (...)" The authors might mistake an "analytical model" for a "physics-based model". While it is true that COSMIC is an analytical forward-operator which mimics physical processes it does not mean that there are actually representations of physical processes. The authors here are probably subject to a 'non-sequitur' error. Correlation does not mean causality. COSMIC eventually uses an exponential function with empirically determined parameters. These parameters unfortunately do not correspond to the physical quantities they are supposed to stand for. The COSMIC approach is similar to the UCF approach by Franz et al. (2013). Desilets before used a hyperbola for describing the N-SM relation. Köhli et al. (2021) showed that the most realistic representation of the measured CRP intensity can be described by a combination of hyperbola and exponential function. As far as an exponential function can represent the N-SM relation to some extent, it does not mean COSMIC with its exponential function represents physical processes in any way. The be clear on that point: COSMIC does not use in its analytical description any physically correct attenuation lengths, instead, if one would require to use such instead of empirical parameter adaptations the equation would not work. Furthermore: COSMIC is described as focusing solely on the influence of locally and directly transported neutrons to the detector, which suggests it may not adequately represent the complete physics of the CRNS method. This limitation could undermine its claim to be a physics-based model if it neglects important physical processes. COSMIC makes several assumptions, such as the belief that neutrons in the soil are only produced by other high-energy neutrons, which may not accurately reflect the true physics of neutron interactions in the environment. This suggests a potential oversimplification or misunderstanding of the underlying physical principles. Additionally, researchers have raised criticism regarding the accuracy of mathematical formulations and calculations within COSMIC, including errors in integrating equations in cylindrical coordinates and inaccuracies in mathematical expressions. These issues suggest a lack of rigor and precision in the model's implementation, which is crucial for any model claiming to be physics-based. As the authors in a later statement pave the ground for an entirely opportunistic choice "Hence, we consider the 1D model assumptions adequate for the target application.", the reviewer asks the authors to acknowledge this in their manuscript.**

AR: *We agree that calling the COSMIC model "physics-based" might be a misleading term, although it has been used in previous literature to describe COSMIC. For an analytical approximation of natural phenomena, it is just natural that every single physical process cannot be represented in great detail. Hence, it is also not logical to assume that the parameters of the model are actual physical attenuation lengths. Instead, COSMIC only mimics the overall picture of the neutron-soil interaction phenomenon, and their parameters are effective representations of the processes. In particular, neutron attenuation lengths do not only depend on the material, but also on the neutron energy itself. Since the neutron constantly changes its energy during the path through the soil, one would have to use an infinite number equations and attenuation lengths to mimic this process. Instead, the equations in COSMIC average out all the different processes and resemble the average attenuation in the material with effective parameters.*

*That said, COSMIC is rather an analytical model than a physical model. We have changed this terminology in the manuscript to avoid further confusion.*

*In this study, we refer to COSMIC as an "analytical model" or "emulator" because it simulates neutron counts based on simplified, empirical relationships rather than explicitly representing the*

*complex physical processes. As acknowledged by Köhli et al. (2021), these simplifications result in a model that is not entirely physics-based but serves as an approximation.*

> Here, we test three approaches, (i) the direct calculation of neutrons from the equal-averaged SWC profiles based on (Schrön et al., 2017), (ii) the same with  weighted-average  soil moisture profiles based on Schrön et al. (2017), and (iii) the  neutron forward operator COSMIC by Shuttleworth et al. (2013) .

> Two empirical and one  the forward operator a COSMIC approaches are evaluated for deriving  neutron counts from the soil moisture profile.

> The COSMIC operator also accounts for the full soil moisture profile,  following the track and attenuation of the neutrons in and out of the soil column.

> On the one hand, it is a  method that aims at mimicking the physical processes of neutron transport in the soil in detailed way, but on the other hand, it relies on the detailed representation of the site characteristics in the hydrological model.

**RC:**6. *"Thank you for this remark. The site-specific nature of the $N_0$ parameter is a well-recognized aspect within the Cosmic-Ray Neutron Sensor (CRNS) community (...)" That response unfortunately contains several logical shortcomings:*

- *The authors cite various studies to support the assertion that $N_0$ is site-specific. However, the mere mention of previous research without providing specific evidence or logical reasoning does not sufficiently substantiate the claim.*
- *Additionally, the observation of non-identical $N_0$ values across different sensors, as noted in the referenced studies, does not inherently establish the site-specificity of $N_0$, especially as the authors also cite studies which have shown the opposite of consistent $N_0$ values.*
- *The author suggests that because the mHM model does not explicitly incorporate site-specific influences on $N_0$, the value of $N_0$ is inferred solely through the calibration procedure.*
- *This oversimplified inference overlooks potential complexities involved in determining the site-specific nature of $N_0$ and assumes that the model's omission of certain factors implies their negligible impact on the $N_0$ determination.*
- *As the authors state in their response: Other factors beyond site characteristics may influence the calibration process. This neither means that the $N_0$ parameter should be site-specific nor theoretically is site-specific.*

AR: *Thank you for your detailed feedback on the site-specific nature of the $N_0$ parameter. I appreciate your observations and agree with the points you raised about the $N_0$ value. Therefore in the revised manuscript, we have simulated neutron counts across the sites using the field measurement $N_0$ values. The $N_0$ was taken from Bogena et al. (2022), the COSMOS Europe paper.*

**RC:**7. *"We are here referring to equation 8 as mentioned in the text. As was explained, it resembles an integral of the vertical neutron transport, geometrically projected to the vertical axis." The authors probably mean "projected integral". The term "geometric integral" is already used in mathematics for a different type of calculation procedure.*

AR: *Thank you for the correcting the terminology used. We have updated and revise our manuscript to use the word "projected integral" instead of "geometric integral."*

> The  original formulation of the COSMIC method has been  further extended by the inclusion of layer-wise lattice water content  and bulk density. Furthermore, COSMIC inside mHM has been numerically optimized to substantially increase the computational performance. This includes the calculation of the  projected integral (Eq. 7) based on lookup tables.

**RC:**8. *"Evidence for the influence of crowding cows at this site" As the authors themselves state in their reply that there is no evidence that this is a relevant influence factor, the reviewer asks the authors to remove such distracting assumptions.*

AR: *Crowding cows have been mentioned by Schrön et al. (2017) as a likely influencing factor on the neutron variability at this site. This statement was based on protocols from the land owner. However, we do not have access to exact time tables and number of cows per hour. Therefore, we agree that this statement is vague and we have deleted the sentence to avoid any speculation.*

>

>

**RC:**9. *"Here the consistency in simulating neutron count variability means that mHM has the capability of capturing the general trend and pattern of the simulated data (...)" Thank you for the clarification provided regarding the interpretation of the statement regarding the consistency in simulating neutron count variability. However, the explanation provided does not fully address the concern raised by the reviewer regarding the potentially misleading nature of highlighting only the top 1% of model runs in combination with potential systematic biases. While the reviewer acknowledges that the top 1% of model runs may demonstrate in some cases a stable performance in terms of overall trend and pattern capture or extentension to a large subset of model runs, the reviewer lacks the understanding of the statistical deviations and variations the authors present. With each subset of model runs showing an inconsistent variability it is not easy to follow the arguments brought up by the authors based for example on seasonality. As brought up earlier, by selectively highlighting only the best-performing model runs, there is a risk of overlooking potential weaknesses or limitations in the model performance across a wider range of conditions and scenarios.*

AR: *Thank you for remark concerns regarding our selection of the top 1% of model runs in our analysis. In our study, we focused on the top 1% of model runs to demonstrate the potential of the model under optimal parameter sets. This choice was the need to identify where the model most effectively captures the variability and trends in neutron counts, which are the aims of our research. Depending on the objective of the study, the hydrological community commonly chooses a certain number of parameter sets (e.g., (Smith et al., 2019; Borriero et al., 2022; Demirel et al., 2024)). The parameter prior and posterior solution is shown in the Supplement material showing which parameters most significantly influence mHM performance for each sites.*

**References**

BGR (2020). Digital soil map of Germany 1 : 200,000 (BUEK 200) v0.5. [Accessed: October 7, 2022].

Boeing, F., Rakovec, O., Kumar, R., Samaniego, L., Schrön, M., Hildebrandt, A., Rebmann, C., Thober, S., Müller, S., Zacharias, S., Bogena, H., Schneider, K., Kiese, R., Attinger, S., and Marx, A. (2022). High-resolution drought simulations and comparison to soil moisture observations in germany. *Hydrology and Earth System Sciences*, 26(19):5137–5161.

Bogena, H. R., Schrön, M., Jakobi, J., Ney, P., Zacharias, S., Andreasen, M., Baatz, R., Boorman, D., Duygu, M. B., Eguibar-Galán, M. A., et al. (2022). Cosmos-europe: a european network of cosmic-ray neutron soil moisture sensors. *Earth System Science Data*, 14(3):1125–1151.

Borriero, A., Kumar, R., Nguyen, T. V., Fleckenstein, J. H., and Lutz, S. R. (2022). Uncertainty in water transit time estimation with storage selection functions and tracer data interpolation. *Hydrology and Earth System Sciences Discussions*, 2022:1–24.

Demirel, M. C., Koch, J., Rakovec, O., Kumar, R., Mai, J., Müller, S., Thober, S., Samaniego, L., and Stisen, S. (2024). Tradeoffs between temporal and spatial pattern calibration and their impacts on robustness and transferability of hydrologic model parameters to ungauged basins. *Water Resources Research*, 60(1):e2022WR034193.

Köhli, M., Weimar, J., Schrön, M., Baatz, R., and Schmidt, U. (2021). Soil Moisture and Air Humidity Dependence of the Above-Ground Cosmic-Ray Neutron Intensity. *Frontiers in Water*, 2:544847.

Samaniego, L., Kumar, R., and Attinger, S. (2010). Multiscale parameter regionalization of a grid-based hydrologic model at the mesoscale. *Water Resources Research*, 46(5).

Schrön, M., Köhli, M., Scheiffele, L., Iwema, J., Bogena, H. R., Lv, L., Martini, E., Baroni, G., Rosolem, R., Weimar, J., et al. (2017). Improving calibration and validation of cosmic-ray neutron sensors in the light of spatial sensitivity. *Hydrology and Earth System Sciences*, 21(10):5009–5030.

Shuttleworth, J., Rosolem, R., Zreda, M., and Franz, T. (2013). The cosmic-ray soil moisture interaction code (cosmic) for use in data assimilation. *Hydrology and Earth System Sciences*, 17(8):3205–3217.

Smith, K. A., Barker, L. J., Tanguy, M., Parry, S., Harrigan, S., Legg, T. P., Prudhomme, C., and Hannaford, J. (2019). A multi-objective ensemble approach to hydrological modelling in the uk: an application to historic drought reconstruction. *Hydrology and Earth System Sciences*, 23(8):3247–3268.

**Author Response to reviewer comment #2**

**Improved representation of soil moisture processes through incorporation of cosmic-ray neutron count measurements in a large-scale hydrologic model**

Fatima et al.
*Hydrol. Earth Syst. Sc.,* `doi:egusphere-2023-1548`
* * *
**RC:** *Referee Comment*,     AR: *Author Response*,     ☐ Manuscript text

AR:  *We appreciate the editor and reviewers' time and insightful feedback, which have improved the manuscript and enhanced the clarity of our research.*

**RC:**1. ***I am still troubled by a potential flaw in this manuscript. As I understand the work, the link between true field soil moisture and modelled comparisons has been lost, although Fig. S5 may show that this is not the case (do Fig. 5 measurements use the fitted $N_{0_{Des}}$ or a field calibrated $N_0$?).***

AR:  *Thank you for your inquiry on Fig. S5 in our manuscript. We relized that the $N_0$ is a very sensitive parameter toward the conversion to soil water content from neutron counts using Desilets equaction, we now fixed the $N_0$ value and did the analysis again, Now in the $\boxed{Fig.5}$ , the $N_0$ parameter value is taken from field calibrations, which is documented for each site, including Grosses Bruch, Hohes Holz, Hordorf, and Cunnerdorf, we utilized the measurement data from the COSMOS Europe data paper by (Bogena et al., 2022), where they converted neutron counts to soil moisture, $\theta(N)$, using the methodology from Desilets et al. (2010).*

$$\theta(N) = \frac{0.0808}{N/N_0 - 0.372} - 0.115 \tag{1}$$

*We have updated Fig.9 on soil moisture in the revised manuscript. This includes a left panel with default parameter runs in mHM,and right panel with the calibrated parameter runs using the $N_{Des,U}$ method that include Figure 1 across all the site, The updated a figure for the other methods of soil water content i.e., $N_{Des,W}$ and $N_{COSMIC}$ and has been added to the supplementary materials.*

*To clarify the statement regarding field soil moisture, we added this explanation to the revised manuscript.*

> *In these figures, the grey dots represent the CRNS soil moisture measurements. The $N_0$ parameter values, taken from field measurement, are documented for each site, including Grosses Bruch, Hohes Holz, Hordorf, and Cunnerdorf. We utilized measurement data from COSMOS Europe (Bogena et al., 2022), where neutron counts were converted to soil moisture, $\theta(N)$, using the methodology from Desilets et al. (2010).*

**Detailed comments to the reply of the authors:**

RC:2. *Note although this paper is about improving soil moisture (process) representation in models (the title), there is only one plot showing calibrated modelled soil moisture, and this is in the Supplementary. In the main paper, only neutron count comparisons are shown. Since both the relationship between neutron counts and soil moisture and the predicted soil moisture are both calibrated, it would seem to me that there is no longer necessarily a representation of true field soil moisture. It is noted that the authors (in previous reply) have chosen to focus on neutron count agreement, rather than soil moisture agreement; however, I would urge the authors to consider how this method can still be traced back to the absolute soil moisture measurement.*

AR: *Thank you for your feedback regarding the focus of our manuscript on neutron count comparisons rather than directly on modeled soil moisture, as neutron counts are a reliable measurement for soil moisture. In response to the reviewer concern, we have revised our approach by fixing the $N_0$ parameter based on field measurements. We then reanalyzed the neutron counts and soil moisture data, and the results for the time series of soil water content using the $N_{Des,U}$ method are now included in the manuscript as shown in Figure 1. Additionally, the results for the $N_{Des,W}$ and $N_{COSMIC}$ methods have been added to the Supplementary Material. Now this analysis ensures that our analysis more directly reflects the true field soil moisture.*

RC:3. ***To elaborate on this point: as the authors describe, the CRNS method for field soil moisture (SM) measurement has a free calibration parameter $N_0$, or more specifically here $N_{0_{Des}}$. Through careful field calibration, normally by collection and moisture analysis of field soil samples, the value of $N_{0_{Des}}$ is determined. This provides the crucial link or traceability of the measured neutron counts to soil moisture content – the quantity which hydrologists are actually interested in knowing.***

***However, in this paper, the field calibration value of $N_{0_{Des}}$ is not used. Instead as part of the model calibration period (line 292) the $N_{0_{Des}}$ is optimised – presumably by minimising the neutron count rate or soil moisture error of the model (it is not stated what objective function was used). This model calibrated $N_{0_{Des}}$ will be different to the field calibration, giving different soil moisture content for a given neutron count – thus the true site-specific calibration of neutron counts to soil moisture has been lost. Whilst the calibrated model may have better agreement with the observed neutron counts, the model output calibrated soil moisture does not necessarily have a similar improvement i.e. the soil moisture could be biased high or low, and that bias accounted for in terms of neutron counts by the model calibrated $N_{0_{Des}}$.***

***Seeing Fig. S5, I actually do not think this flaw really exists – but the detail of the $N_0$ model calibration versus soil moisture calibration needs to be clearer to explain how this potential issue has been dealt with. The authors should justify their approach of a model calibrated $N_0$ versus using the value already known from site specific field calibration of the CRNS. And it may be of value to compare these.***

AR: *Thank you for your detailed observations concerning the calibration of $N_{0_{Des}}$ in our study. Based on the reviewer's suggestion, we conducted an experiment by fixing the $N_0$ values from the measurement sites for each location and run the model by taking snow, soil moisture, and neutron parameters for 100 000 simulation we took the best 10 parameterset based on the objective function $KGE_{\alpha\beta}$. We found that by fixing the $N_0$ values, the observed neutron counts matched well for the agriculture sites* Cunnerdrof, Hordorf *and grassland site* Grosses Bruch. *However, a larger discrepancy was noted at* Hohes Holz, *a dense forest site. This difference could be attributed to the Leaf Area Index (LAI), biomass and*

*vegetation dynamics, which are not currently integrated into mHM. Recent efforts by Bahrami et al. (2022) aim to address vegetation dynamics in mHM, but this integration is still incomplete. Our study suggests that future research should focus on regionalizing these parameters i.e., $N_0$, particularly in ungauged locations, to enhance model accuracy and applicability.*

[Figure]

Figure 1: Daily $SWC_{Des,U}$ time series at four sites, comparing observed SWC (gray dots) with mHM-derived SWC at 5 cm (green), 25 cm (blue), and 60 cm (orange) and the average (red). The left panels use default mHM parameters, while the right panels use parameters calibrated with $N_{Des,U}$ method. Selection of $N_0$ values was based on COSMOS Europe data.

**RC:**4. *Line 303 "Estimated values of $N_{0_{Des}}$ and $N_{0_{COSMIC}}$ obtained in our study are close to optimal values" – how do you know that? What are the optimal values? And the inference drawn is not sound – model simulation of dry conditions is not a prerequisite to obtaining accurate $N_0$ values.*

AR: *Thank you for your comment. Upon reviewing the evidence and addressing the concerns highlighted in your comment, we acknowledge the issue regarding the closeness of the estimated values of $N_{0_{COSMIC}}$ to the observed values in our study. This discrepancy arises because $N_0$ functions as a free parameter, varying significantly depending on the chosen method COSMIC or Desilets as also different $N_0$ values depends upon the method by Iwema et al. (2015); Baatz et al. (2014). However, since we do not have field measurements for $N_{0_{COSMIC}}$, we calculated using the Shuttleworth et al. (2013) method where the*

*observed neutron counts and soil moisture from different profile depth are known the only unknown value was $N_0$.*

To address your concern, we have removed this sentence from the manuscript to maintain accuracy and avoid unsupported claims.

>

**RC:**5. *Results – Fig. 5 Also show plots of SWC (as per field calibration – observations) and calibrated modelled SWC.*

AR: *Thank you for your feedback and for suggesting the inclusion of plots showing the traceability of the measured neutron counts to soil moisture content. Also inclusion of SWC estimated from mHM neutron counts. We have showed the result of the time series of SWC in Figure 1. This figure now includes the results where the $N_0$ value was fixed based on field measurements, and the analysis was conducted accordingly. The new plot provides a comparison between observed SWC (from field calibration) and the SWC derived from the calibrated mHM.*

**RC:**6. *Discussion – I would question the soundness of discussing model performance, when it appears that neutron count comparisons rather than SWC have been calibrated. As the authors have chosen to present neutron count data, then they need to be careful as to what is claimed with regard to soil moisture modelling, or to provide evidence to support those claims.*

AR: *Thank you for your feedback. We understand that our explanation of model performance in relation to SWC calibration has to be clarified. Accordingly, we will revise the discussion section to focus on the calibration of neutron counts and their implications for soil moisture estimation. As mentioned in response to above question 3, we have already addressed the estimation of SWC.*

**RC:**7. *Line 440 " …improved not only soil moisture estimation" – NO improvement in soil moisture estimation is shown in the main paper!*

AR: *Thank you for highlighting the need for clearer representation of soil moisture estimation improvements. To address this, we will include Figure 1 about SWC into the Discussion section. This figure shows the soil water content of different layer along with the total average soil water content of mHM based on the calibration of neutron counts.*

**RC:**8. *Conclusion – Line 507 …evaluation with soil moisture observations has not been shown.*

*Several statements in the conclusion are not supported by the paper (at least not without digging into Supplementary material) e.g. Line 525 "improved the soil moisture performance of the model.*

AR: *It is noted in the conclusion that the evaluation of soil moisture performance was not adequately illustrated in the main paper. We have therefore chosen to include Figure 1, which presents the soil moisture performance of the mHM, in the paper's main discussion section in order to correct this omission. This inclusion of Figure 1 along with a performance matrix table from all sites will make sure that the conclusions are easily verified by readers .*

**References**

Baatz, R., Bogena, H., Franssen, H.-J. H., Huisman, J., Qu, W., Montzka, C., and Vereecken, H. (2014). Calibration of a catchment scale cosmic-ray probe network: A comparison of three parameterization methods. *Journal of Hydrology*, 516:231–244.

Bahrami, B., Hildebrandt, A., Thober, S., Rebmann, C., Fischer, R., Samaniego, L., Rakovec, O., and Kumar, R. (2022). Developing a parsimonious canopy model (pcm v1. 0) to predict forest gross primary productivity and leaf area index of deciduous broad-leaved forest. *Geoscientific Model Development*, 15(18):6957–6984.

Iwema, J., Rosolem, R., Baatz, R., Wagener, T., and Bogena, H. (2015). Investigating temporal field sampling strategies for site-specific calibration of three soil moisture–neutron intensity parameterisation methods. *Hydrology and Earth System Sciences*, 19(7):3203–3216.

Shuttleworth, J., Rosolem, R., Zreda, M., and Franz, T. (2013). The cosmic-ray soil moisture interaction code (cosmic) for use in data assimilation. *Hydrology and Earth System Sciences*, 17(8):3205–3217.